# Improving the Knowledge Gradient Algorithm

**Le Yang**
Department of Systems Engineering
City University of Hong Kong
lyang272-c@my.cityu.edu.hk

**Siyang Gao**
Department of Systems Engineering
City University of Hong Kong
siyangao@cityu.edu.hk

**Chin Pang Ho**
School of Data Science
City University of Hong Kong
clint.ho@cityu.edu.hk

## Abstract

The knowledge gradient (KG) algorithm is a popular policy for the best arm identification (BAI) problem. It is built on the simple idea of always choosing the measurement that yields the greatest expected one-step improvement in the estimate of the best mean of the arms. In this research, we show that this policy has limitations, causing the algorithm not asymptotically optimal. We next provide a remedy for it, by following the manner of one-step look ahead of KG, but instead choosing the measurement that yields the greatest one-step improvement in the probability of selecting the best arm. The new policy is called improved knowledge gradient (iKG). iKG can be shown to be asymptotically optimal. In addition, we show that compared to KG, it is easier to extend iKG to variant problems of BAI, with the $\epsilon$-good arm identification and feasible arm identification as two examples. The superior performances of iKG on these problems are further demonstrated using numerical examples.

## 1 Introduction

The best arm identification (BAI) is a sequential decision problem where in each stage, the agent pulls one out of $k$ given arms and observes a noisy sample of the chosen arm. At the end of the sampling stage, the agent needs to select the arm that is believed to be the best according to the samples. In this research, we let the best arm be the one with the largest mean. BAI is a useful abstraction of issues faced in many practical settings [1, 2] and has been widely studied in the machine learning community [3, 4]. Since in practical problems, the target arm(s) (to be identified) is not necessarily the best arm, some variant models of BAI have also been proposed in the literature, e.g., top-$m$ arm identification [5, 6], Pareto front identification [7], $\epsilon$-good arm identification [8], feasible arm identification [9, 10], etc.

In this research, we focus on the fixed-budget BAI, in which the total number of samples (budget) is fixed and known by the agent. The goal is to correctly identify the best arm when the budget is used up. To solve this problem, many methods have been proposed, e.g., successive rejects (SR) [4], expected improvements (EI) [11], top-two sampling [12, 13], knowledge gradient (KG) [14, 15], optimal computing budget allocation (OCBA)[16, 17, 18], etc. Among these methods, KG has been prevailing. It was first proposed in [19] and further analyzed in [20, 21]. It is built on the simple idea of always pulling the arm that yields the greatest expected one-step improvement in the estimate of the best mean of the arms. This improvement measure is analytical, making the algorithm easily implementable. KG often offers reasonable empirical performances and has been successfully applied in a number of real applications [22, 23].

37th Conference on Neural Information Processing Systems (NeurIPS 2023).

However, we observe that this definition of KG has limitations, causing the algorithm not asymptotically optimal. Here by not being asymptotically optimal, we mean that the KG algorithm is not rate optimal, in the sense that the probability of the best arm being falsely selected based on the posterior means of the $k$ arms does not converge to zero at the fastest possible rate. This is resulted from KG allocating too few samples to the best arm and excessive samples to the remaining arms. Note that Frazier et al. [20] claimed that KG is "asymptotically optimal", but in their context, "asymptotically optimal" is consistent, i.e., all the arms will be infinitely sampled as the round $n \to \infty$, so that the best arm will be correctly selected eventually. This is a relatively weak result for BAI algorithms (the simple equal allocation is also consistent). In this paper, asymptotically optimal refers to rate optimal.

**Contributions.** We propose a new policy that can overcome this limitation of KG. The new policy follows the manner of one-step look ahead of KG, but pulls the arm that yields the greatest one-step improvement in the probability of selecting the best arm. We call it improved knowledge gradient (iKG) and show that it is asymptotically optimal. This policy is originated from the thought of looking at whether the best arm has been selected at the end of sampling, instead of looking at the extent that the mean of the selected arm has been maximized. Although both ways can identify the best arm, it turns out that the algorithms developed from them are significantly different in the rates of posterior convergence. Another advantage of iKG over KG is that iKG is more general and can be more easily extended to variant problems of BAI. We use $\epsilon$-good arm identification and feasible arm identification as examples, develop algorithms for them using the idea of iKG and establish asymptotic optimality for the algorithms.

This paper is conceptually similar to [12] which improves the EI algorithm for BAI. However, for EI, sampling ratios of any two arms in the non-best set are already asymptotically optimal. One only needs to introduce a parameter $\beta$ to balance the probabilities of sampling the best arm and the non-best set without changing the sampling policy within the non-best set to further improve EI. For KG, sampling ratios are not asymptotically optimal for any two out of the $k$ arms. It requires a fundamental change on the sampling policy that influences the sampling rates of all the arms to improve KG. Moreover, the improved rate of posterior convergence of EI in [12] still depends on $\beta$ which is not necessarily optimal, while we can show that this rate of iKG is optimal.

## 2 Knowledge Gradient and its Limitations

In this section, we review KG and discuss its limitations. Suppose there are $k$ arms in BAI. In each round $t$, the agent chooses any arm $i$ to pull and obtains a noisy sample $X_{t+1,i}$. After $n$ rounds, the agent needs to select an arm that he/she believes to be the best. Under the framework of the KG algorithm, $X_{t+1,i}$'s are assumed to be independent across different rounds $t$ and arms $i$ and following the normal distribution $\mathcal{N}(\mu_i, \sigma_i^2)$ with unknown means $\mu_i$ and known variances $\sigma_i^2$. The best arm is assumed to be unique. Without loss of generality, let $\mu_{\langle 1 \rangle} > \mu_{\langle 2 \rangle} \geq \ldots \geq \mu_{\langle k \rangle}$, where $\langle i \rangle$ indicates the arm with $i$-th largest mean.

The KG algorithm can be derived from a dynamic programming (DP) formulation of BAI. The state space $\mathbb{S}$ consists of all the possible posterior means and variances of the arms, denoted as $\mathbb{S} \triangleq \mathbb{R}^k \times (0, \infty)^k$. State $S_t$ in round $t$ can be written as $S_t = (\mu_{t,1}, \mu_{t,2}, \ldots, \mu_{t,k}, \sigma_{t,1}^2, \sigma_{t,2}^2, \ldots, \sigma_{t,k}^2)^\top$. In the Bayesian model, the unknown mean $\mu_i$ is treated as random and let $\theta_i$ be the random variable following its posterior distribution. We adopt normal distribution priors $\mathcal{N}(\mu_{0,i}, \sigma_{0,i}^2)$. With samples of the arms, we can compute their posterior distributions, which are still normal $\mathcal{N}(\mu_{t,i}, \sigma_{t,i}^2)$ in round $t$ by conjugacy. The posterior mean and variance of arm $i$ are

$$
\mu_{t+1,i} = \begin{cases} \dfrac{\sigma_{t,i}^{-2} \mu_{t,i} + \sigma_i^{-2} X_{t+1,i}}{\sigma_{t,i}^{-2} + \sigma_i^{-2}} & \text{if } I_t = i, \\ \mu_{t,i} & \text{if } I_t \neq i, \end{cases} \quad \text{and} \quad \sigma_{t+1,i}^2 = \begin{cases} \dfrac{1}{\sigma_{t,i}^{-2} + \sigma_i^{-2}} & \text{if } I_t = i, \\ \sigma_{t,i}^2 & \text{if } I_t \neq i. \end{cases} \tag{1}
$$

In this paper, we adopt a non-informative prior for each arm $i \in \mathbb{A}$, i.e., $\mu_{0,i} = 0$ and $\sigma_{0,i} = \infty$. Denote the action space as $\mathbb{A} \triangleq \{1, 2, \ldots, k\}$ and transition function as $\mathcal{T} \triangleq \mathbb{S} \times \mathbb{A} \times \mathbb{S} \to \mathbb{S}$. Suppose $\theta_{t,i}$ is a random variable following the posterior distribution $\mathcal{N}(\mu_{t,i}, \sigma_i^2)$ of arm $i$. Then, the state transition can be written as $S_{t+1} = \mathcal{T}(S_t, i, \theta_{t,i})$. Let $\pi$ be the sampling policy that guides the agent to pull arm $I_t$ in round $t$ and $\Pi$ be the set of sampling policies $\pi = (I_0, I_1, \ldots, I_{n-1})$ adapted to the filtration $I_0, X_{1,I_0}, \ldots, I_{t-1}, X_{t,I_{t-1}}$. After $n$ rounds, the estimated best arm $I_n^*$ is selected

and a terminal reward $v_n(S_n)$ is received. We can write our objective as

$$\sup_{\pi \in \Pi} \mathbb{E}_\pi v_n(S_n). \tag{2}$$

The DP principle implies that the value function in round $0 \le t < n$ can be computed recursively by

$$v_t(S) \triangleq \max_{i \in \mathbb{A}} \mathbb{E}[v_{t+1}(\mathcal{T}(S, i, \theta_{t,i}))], \quad S \in \mathbb{S}.$$

We define the Q-factors as

$$Q_t(S, i) \triangleq \mathbb{E}[v_{t+1}(\mathcal{T}(S, i, \theta_{t,i}))], \quad S \in \mathbb{S},$$

and the DP principle tells us that any policy satisfying

$$I_t(S) \in \operatorname*{argmax}_{i \in \mathbb{A}} Q_t(S, i), \quad S \in \mathbb{S}$$

is optimal. However, the optimal policy is basically intractable unless for problems with very small scales, known as the "curse of dimensionality".

On the other hand, note that except the terminal reward $v_n(S_n)$, this problem has no rewards in the other rounds, so we can restructure $v_n(S_n)$ as a telescoping sequence

$$v_n(S_n) = [v_n(S_n) - v_n(S_{n-1})] + \ldots + [v_n(S_{t+1}) - v_n(S_t)] + v_n(S_t).$$

Thus, $v_n(S_n)$ can be treated as the cumulation of multiple one-step improvements $v_n(S_l) - v_n(S_{l-1})$, $l = t + 1, \ldots, n$. A class of one-step look ahead algorithms iteratively pull the arm that maximizes the expectation of the one-step improvement on the value function

$$\mathbb{E}[v_n(\mathcal{T}(S_t, i, \theta_{t,i})) - v_n(S_t)]. \tag{3}$$

These algorithms are not optimal in general unless there is only one round left, i.e., $n = t + 1$.

The KG algorithm falls in this class. It sets the terminal reward as $v_n(S_n) = \mu_{I_n^*}$. With this reward, the one-step improvement in (3) becomes

$$\mathrm{KG}_{t,i} = \mathbb{E}[\max\{\mathcal{T}(\mu_{t,i}, i, \theta_{t,i}), \max_{i' \neq i} \mu_{t,i'}\} - \max_{i \in \mathbb{A}} \mu_{t,i}],$$

and in each round, the KG algorithm pulls the arm $I_t(S_t) \in \operatorname{argmax}_{i \in \mathbb{A}} \mathrm{KG}_{t,i}$.

---

**Algorithm 1:** KG Algorithm

**Input:** $k \ge 2, n$
1 Collect $n_0$ samples for each arm $i$;
2 **while** $t < n$ **do**
3      Compute $\mathrm{KG}_{t,i}$ and set $I_t = \operatorname{argmax}_{i \in \mathbb{A}} \mathrm{KG}_{t,i}$;
4      Play $I_t$;
5      Update $\mu_{t+1,i}$ and $\sigma_{t+1,i}$;
**Output:** $I_n^*$

---

We next characterize for the KG algorithm the rate of posterior convergence of $1 - \mathbb{P}\{I_n^* = I^*\}$, the probability that the best arm is falsely selected.

**Proposition 1.** *Let $c_{\langle i \rangle} = \frac{(\mu_{\langle 1 \rangle} - \mu_{\langle i \rangle})/\sigma_{\langle i \rangle}}{(\mu_{\langle 1 \rangle} - \mu_{\langle 2 \rangle})/\sigma_{\langle 2 \rangle}}$, $i = 2, ..., k$. For the KG algorithm,*

$$\lim_{n \to \infty} -\frac{1}{n} \log(1 - \mathbb{P}\{I_n^* = I^*\}) = \Gamma^{KG},$$

*where*

$$\Gamma^{KG} = \min_{i \neq 1} \left( \frac{(\mu_{\langle i \rangle} - \mu_{\langle 1 \rangle})^2}{2((\sum_{i \neq 1} \sigma_{\langle 2 \rangle}/c_{\langle i \rangle} + \sigma_{\langle 1 \rangle})\sigma_{\langle 1 \rangle} + c_{\langle i \rangle}\sigma_{\langle i \rangle}^2(\sum_{i \neq 1} 1/c_{\langle i \rangle} + \sigma_{\langle 1 \rangle}/\sigma_{\langle 2 \rangle}))} \right).$$

We observe that $\Gamma^{KG}$ is not optimal. To make this point, Proposition 2 gives an example that $\Gamma^{KG}$ is no better than this rate of the TTEI algorithm [12] when the parameter $\beta$ (probability of sampling the best arm) of TTEI is set to some suboptimal value.

**Proposition 2.** *For the TTEI algorithm [12], the rate of posterior convergence of $1 - \mathbb{P}\{I_n^* = I^*\}$ exists and is denoted as $\Gamma^{TTEI}$. Let its probability of sampling the best arm $\beta = (\sigma_{\langle 2 \rangle}/\sigma_{\langle 1 \rangle} \sum_{i \neq 1} 1/c_{\langle i \rangle} + 1)^{-1}$. We have $\Gamma^{KG} \leq \Gamma^{TTEI}$.*

According to the proof of Proposition 2, there are configurations of the BAI problem leading to $\Gamma^{KG} < \Gamma^{TTEI}$, i.e., $\Gamma^{KG}$ is not optimal. In fact, with $\beta = (\sigma_{\langle 2 \rangle}/\sigma_{\langle 1 \rangle} \sum_{i \neq 1} 1/c_{\langle i \rangle} + 1)^{-1}$, $\Gamma^{KG} = \Gamma^{TTEI}$ is achieved only in some special cases, e.g., when $k = 2$.

## 3 Improved Knowledge Gradient

In this section, we propose an improved knowledge gradient (iKG) algorithm. We still follow the manner of one-step look ahead of KG, but set the terminal reward of problem (2) as $v_n(S_n) = \mathbf{1}\{I_n^* = I^*\}$. That is, for the goal of identifying the best arm, we reward the selected arm by a 0-1 quantity showing whether this arm is the best arm, instead of the mean of this arm (as in KG).

In this case, $\mathbb{E}[v_n(S_n)] = \mathbb{P}\{I_n^* = I^*\}$, where

$$\mathbb{P}\{I_n^* = I^*\} = \mathbb{P}\left\{ \bigcap_{i \neq I_n^*} (\theta_{I_n^*} > \theta_i) \right\} = 1 - \mathbb{P}\left\{ \bigcup_{i \neq I_n^*} (\theta_i > \theta_{I_n^*}) \right\}. \tag{4}$$

However, the probability $\mathbb{P}\left\{ \bigcup_{i \neq I_n^*} (\theta_i > \theta_{I_n^*}) \right\}$ in (4) does not have an analytical expression. To facilitate the algorithm implementation and analysis, we adopt an approximation to it using the Bonferroni inequality [24]:

$$\mathbb{P}\left\{ \bigcup_{i \neq I_n^*} (\theta_i > \theta_{I_n^*}) \right\} \leq \sum_{i \neq I_n^*} \mathbb{P}(\theta_i > \theta_{I_n^*}),$$

and $\mathbb{E}[v_n(S_n)]$ can be approximately computed as

$$\mathbb{E}[v_n(S_n)] \approx 1 - \sum_{i \neq I_n^*} \mathbb{P}(\theta_i > \theta_{I_n^*}) = 1 - \sum_{i \neq I_n^*} \exp\left( -\frac{(\mu_{n,i} - \mu_{n,I_n^*})^2}{2(\sigma_{n,i}^2 + \sigma_{n,I_n^*}^2)} \right). \tag{5}$$

Note that the Bonferroni inequality has been adopted as an approximation of the probability of correct selection in the literature for development of BAI algorithms [16]. For our purpose, we can show that the use of this approximation still makes the resulting algorithm asymptotically optimal and empirically superior.

---

**Algorithm 2:** iKG Algorithm

---

**Input:** $k \geq 2$, $n$
1 Collect $n_0$ samples for each arm $i$;
2 **while** $t < n$ **do**
3     Compute $\text{iKG}_{t,i}$ and set $I_t = \text{argmax}_{i \in \mathbb{A}} \text{iKG}_{t,i}$;
4     Play $I_t$;
5     Update $\mu_{t+1,i}$, $\sigma_{t+1,i}$ and $I_{t+1}^*$;
**Output:** $I_n^*$

---

Let $\text{iKG}_{t,i}$ be the one-step improvement in (3) with $I_t^*$ treated as unchanged after one more sample and $\mathbb{E}[v_n(S_n)]$ approximated by (5). We have the following proposition to compute $\text{iKG}_{t,i}$. The iKG algorithm pulls the arm with the largest $\text{iKG}_{t,i}$ in each round.

**Proposition 3.** *With the definition of $\text{iKG}_{t,i}$ above, we have*

$$\text{iKG}_{t,i} = \begin{cases} \exp\left( -\frac{(\mu_{t,i} - \mu_{t,I_t^*})^2}{2(\sigma_{t,i}^2 + \sigma_{t,I_t^*}^2)} \right) - \exp\left( -\frac{(\mu_{t,i} - \mu_{t,I_t^*})^2}{2(\sigma_{t+1,i}^2 + \sigma_{t,I_t^*}^2 + \sigma_i^2(\sigma_{t+1,i}^2/\sigma_i^2)^2)} \right), & \text{if } i \neq I_t^*, \\[4mm] \sum_{i' \neq I_t^*} \exp\left( -\frac{(\mu_{t,i'} - \mu_{t,I_t^*})^2}{2(\sigma_{t,i'}^2 + \sigma_{t,I_t^*}^2)} \right) - \sum_{i' \neq I_t^*} \exp\left( -\frac{(\mu_{t,i'} - \mu_{t,I_t^*})^2}{2(\sigma_{t,i'}^2 + \sigma_{t+1,I_t^*}^2 + \sigma_{I_t^*}^2(\sigma_{t+1,I_t^*}^2/\sigma_{I_t^*}^2)^2)} \right), & \text{if } i = I_t^*. \end{cases} \tag{6}$$

Both KG and iKG are greedy algorithms that look at the improvement only one-step ahead. The essential difference between them is on the reward they use for the event of best arm identification. For KG, it is the mean of the arm selected, while for iKG, it is a 0-1 quantity showing whether the best arm is selected. It is interesting to note that the choice between these two rewards has been discussed in the control community for optimization of complex systems, known as cardinal optimization (similar to KG) vs. ordinal optimization (similar to iKG) [25], with the discussion result in line with this research, indicating that ordinal optimization has advantages over cardinal optimization in the convergence rates of the optimization algorithms [26].

**Theorem 1.** *For the iKG algorithm,* $\lim_{n\to\infty} -\frac{1}{n}\log(1 - \mathbb{P}\{I_n^* = I^*\}) = \Gamma^{iKG}$, *where*

$$\Gamma^{iKG} = \frac{(\mu_{\langle i\rangle} - \mu_{\langle 1\rangle})^2}{2(\sigma_{\langle i\rangle}^2/w_{\langle i\rangle} + \sigma_{\langle 1\rangle}^2/w_{\langle 1\rangle})}, \tag{7}$$

*and $w_i$ is the sampling rate of arm $i$ satisfying*

$$\sum_{i=1}^{k} w_i = 1, \quad \frac{w_{\langle 1\rangle}^2}{\sigma_{\langle 1\rangle}^2} = \sum_{i=2}^{k} \frac{w_{\langle i\rangle}^2}{\sigma_{\langle i\rangle}^2} \quad and \quad \frac{(\mu_{\langle i\rangle} - \mu_{\langle 1\rangle})^2}{2(\sigma_{\langle i\rangle}^2/w_{\langle i\rangle} + \sigma_{\langle 1\rangle}^2/w_{\langle 1\rangle})} = \frac{(\mu_{\langle i'\rangle} - \mu_{\langle 1\rangle})^2}{2(\sigma_{\langle i'\rangle}^2/w_{\langle i'\rangle} + \sigma_{\langle 1\rangle}^2/w_{\langle 1\rangle})}, \quad i \neq i' \neq 1. \tag{8}$$

*In addition, for any BAI algorithms,*

$$\limsup_{n\to\infty} -\frac{1}{n}\log(1 - \mathbb{P}\{I_n^* = I^*\}) \leq \Gamma^{iKG}.$$

Theorem 1 shows that the rate of posterior convergence $\Gamma^{iKG}$ of the iKG algorithm is the fastest possible. We still use TTEI as an example. This theorem indicates that $\Gamma^{TTEI} \leq \Gamma^{iKG}$ for any $\beta \in (0, 1)$ and the equality holds only when $\beta$ is set to $\beta^*$, where $\beta^*$ is the optimal value of $\beta$ and is typically unknown.

## 4 Variant Problems of BAI

Another advantage of iKG over KG is that iKG is more general, in the sense that it can be easily extended to solve variant problems of BAI. In the variants, the target arms to be identified are not the single best arm, but no matter how the target arms are defined, one can always look at the event that whether these arms are correctly identified at the end of sampling and investigate the probability of this event to develop iKG and the algorithm. In contrast, it is difficult to extend KG to identify arms that cannot be found through optimizing means of these (and/or other) arms. In this section, we extend iKG to two BAI variants: $\epsilon$-good arm identification [8] and feasible arm identification [10]. We develop algorithms for them and establish their asymptotic optimality. Note that in these two variant problems, the target arms need to be found by comparing their means with some fixed values. In such cases, the idea of KG is not straightforward.

### 4.1 $\epsilon$-Good Arm Identification

We follow the notation in Sections 2 and 3. For the $k$ arms, suppose $\mu_{\langle 1\rangle} \geq \mu_{\langle 2\rangle} \geq \ldots \geq \mu_{\langle k\rangle}$. Given $\epsilon > 0$, the $\epsilon$-good arm identification problem aims to find all the arms $i$ with $\mu_{\langle i\rangle} > \mu_{\langle 1\rangle} - \epsilon$, i.e., all the arms whose means are close enough to the best ($\epsilon$-good). Assume that no arms have means lying on $\mu_{\langle 1\rangle} - \epsilon$. Denote the set of $\epsilon$-good arms as $G^\epsilon$ and the estimated set of $\epsilon$-good arms after $n$ rounds as $G_n^\epsilon$. We set the terminal reward $v_n(S_n) = \mathbf{1}\{G_n^\epsilon = G^\epsilon\}$, i.e., whether the set $G^\epsilon$ is correctly selected. Then, $\mathbb{E}[v_n(S_n)] = \mathbb{P}\{G_n^\epsilon = G^\epsilon\}$, where

$$\mathbb{P}\{G_n^\epsilon = G^\epsilon\} = \mathbb{P}\left\{ \bigcap_{i\in G_n^\epsilon} (\theta_i > \max_{i'\in\mathbb{A}}\theta_{i'} - \epsilon) \cap \bigcap_{i\in\mathbb{A}\backslash G_n^\epsilon} (\theta_i < \max_{i'\in\mathbb{A}}\theta_{i'} - \epsilon) \right\}$$

$$= 1 - \mathbb{P}\left\{ \bigcup_{i\in G_n^\epsilon} (\theta_i < \max_{i'\in\mathbb{A}}\theta_{i'} - \epsilon) \cup \bigcup_{i\in\mathbb{A}\backslash G_n^\epsilon} (\theta_i > \max_{i'\in\mathbb{A}}\theta_{i'} - \epsilon) \right\}.$$

Again, applying the Bonferroni inequality,

$$\mathbb{P}\{G_n^\epsilon = G^\epsilon\} \geq 1 - \sum_{i\in G_n^\epsilon} \mathbb{P}(\theta_i < \max_{i'\in\mathbb{A}}\theta_{i'} - \epsilon) - \sum_{i\in\mathbb{A}\backslash G_n^\epsilon} \mathbb{P}(\theta_i > \max_{i'\in\mathbb{A}}\theta_{i'} - \epsilon). \tag{9}$$

Let iKG$_{t,i}^\epsilon$ be the one-step improvement in (3) with $I_t^*$ treated as unchanged after one more sample and $\mathbb{E}[v_n(S_n)]$ approximated by the right-hand side of (9). We have the following proposition to compute iKG$_{t,i}^\epsilon$ .

**Proposition 4.** *With the definition of iKG$_{t,i}^\epsilon$ above, we have*

$$
iKG_{t,i}^\epsilon = \begin{cases}
\exp\left(-\dfrac{(\mu_{t,i} - \mu_{t,I_t^*} + \epsilon)^2}{2(\sigma_{t,i}^2 + \sigma_{t,I_t^*}^2)}\right) - \exp\left(-\dfrac{(\mu_{t,i} - \mu_{t,I_t^*} + \epsilon)^2}{2(\sigma_{t+1,i}^2 + \sigma_{t,I_t^*}^2 + \sigma_i^2(\sigma_{t+1,i}^2/\sigma_i^2)^2)}\right), & \text{if } i \neq I_t^*, \\[3mm]
\displaystyle\sum_{i' \neq I_t^*} \exp\left(-\dfrac{(\mu_{t,i'} - \mu_{t,I_t^*} + \epsilon)^2}{2(\sigma_{t,i'}^2 + \sigma_{t,I_t^*}^2)}\right) - \sum_{i' \neq I_t^*} \exp\left(-\dfrac{(\mu_{t,i'} - \mu_{t,I_t^*} + \epsilon)^2}{2(\sigma_{t,i'}^2 + \sigma_{t+1,I_t^*}^2 + \sigma_{I_t^*}^2(\sigma_{t+1,I_t^*}^2/\sigma_{I_t^*}^2)^2)}\right), & \text{if } i = I_t^*.
\end{cases}
\tag{10}
$$

---

**Algorithm 3:** iKG-$\epsilon$ Algorithm ($\epsilon$-good Arm Identification)

**Input:** $k \geq 2, n$
1 Collect $n_0$ samples for each arm $i$;
2 **while** $t < n$ **do**
3     Compute iKG$_{t,i}^\epsilon$ and set $I_t = \text{argmax}_{i \in \mathbb{A}} \text{iKG}_{t,i}^\epsilon$;
4     Play $I_t$;
5     Update $\mu_{t+1,i}, \sigma_{t+1,i}$ and $I_{t+1}^*$;
**Output:** $G_n^\epsilon$

---

To identify the $\epsilon$-good arms, the iKG-$\epsilon$ algorithm pulls the arm with the largest iKG$_{t,i}^\epsilon$ in each round. For this algorithm, we can show that the rate of posterior convergence of $1 - \mathbb{P}\{G_n^\epsilon = G^\epsilon\}$ is the fastest possible.

**Theorem 2.** *For the iKG-$\epsilon$ algorithm, $\lim_{n \to \infty} -\frac{1}{n} \log(1 - \mathbb{P}\{G_n^\epsilon = G^\epsilon\}) = \Gamma^\epsilon$, where*

$$
\Gamma^\epsilon = \frac{(\mu_{\langle i \rangle} - \mu_{\langle 1 \rangle} + \epsilon)^2}{2(\sigma_{\langle i \rangle}^2/w_{\langle i \rangle} + \sigma_{\langle 1 \rangle}^2/w_{\langle 1 \rangle})},
\tag{11}
$$

*and $w_i$ is the sampling rate of arm $i$ satisfying*

$$
\sum_{i=1}^k w_i = 1, \quad \frac{w_{\langle 1 \rangle}^2}{\sigma_{\langle 1 \rangle}^2} = \sum_{i=2}^k \frac{w_{\langle i \rangle}^2}{\sigma_{\langle i \rangle}^2} \quad \text{and} \quad \frac{(\mu_{\langle i \rangle} - \mu_{\langle 1 \rangle} + \epsilon)^2}{2(\sigma_{\langle i \rangle}^2/w_{\langle i \rangle} + \sigma_{\langle 1 \rangle}^2/w_{\langle 1 \rangle})} = \frac{(\mu_{\langle i' \rangle} - \mu_{\langle 1 \rangle} + \epsilon)^2}{2(\sigma_{\langle i' \rangle}^2/w_{\langle i' \rangle} + \sigma_{\langle 1 \rangle}^2/w_{\langle 1 \rangle})}, \quad i \neq i' \neq 1.
\tag{12}
$$

*In addition, for any $\epsilon$-good arm identification algorithms,*

$$
\limsup_{n \to \infty} -\frac{1}{n} \log(1 - \mathbb{P}\{G_n^\epsilon = G^\epsilon\}) \leq \Gamma^\epsilon.
$$

### 4.2 Feasible Arm Identification

In the feasible arm identification, samples from pulling arms $i$ are $m$-dimensional vectors $\boldsymbol{X}_{t+1,i} = [X_{t+1,i1}, \ldots, X_{t+1,im}]$ instead of scalars, where each dimension of the vector corresponds to some measure of the system performance and $X_{t+1,ij}$ is the observation associated with arm $i$ and measure $j$. Suppose $X_{t+1,ij}$'s follow the normal distribution with unknown means $\mu_{ij}$ and known variances $\sigma_{ij}^2$. We impose constraints $\mu_{ij} \leq \gamma_j$ on arms $i = 1, 2, \ldots, k$ and measures $j = 1, 2, \ldots, m$. The goal of this problem is to find the set of feasible arms $\mathcal{S}^1$. Let the estimated set of feasible arms after $n$ rounds be $\mathcal{S}_n^1$ and $\mathcal{S}^2 = \mathbb{A} \setminus \mathcal{S}^1$. We assume that $X_{t+1,ij}$'s are independent across different rounds $t$ and measures $j$, and $\mu_{ij}$'s do not lie on the constraint limits $\gamma_j$. To facilitate the analysis, we also define for round $t$ the set of measures $\mathcal{E}_{t,i}^1 \triangleq \{j : \mu_{t,ij} \leq \gamma_j\}$ satisfied by arm $i$ and the set of measures $\mathcal{E}_{t,i}^2 \triangleq \{j : \mu_{t,ij} > \gamma_j\}$ violated by arm $i$.

Set the terminal reward $v_n(S_n) = \mathbf{1}\{S_n^1 = S^1\}$, i.e., whether the set $S^1$ is correctly selected. Then, $\mathbb{E}[v_n(S_n)] = \mathbb{P}\{S_n^1 = S^1\}$, where

$$\mathbb{P}\{S_n^1 = S^1\} = \mathbb{P}\bigg\{ \bigcap_{i \in S_n^1} \bigg( \bigcap_{j=1}^m (\theta_{ij} \le \gamma_j) \bigg) \cap \bigcap_{i \in S_n^2} \bigg( \bigcup_{j=1}^m (\theta_{ij} > \gamma_j) \bigg) \bigg\}$$

$$= 1 - \mathbb{P}\bigg\{ \bigcup_{i \in S_n^1} \bigg( \bigcup_{j=1}^m (\theta_{ij} > \gamma_j) \bigg) \cup \bigcup_{i \in S_n^2} \bigg( \bigcap_{j=1}^m (\theta_{ij} \le \gamma_j) \bigg) \bigg\}.$$

Applying the Bonferroni inequality,

$$\mathbb{P}\{S_n^1 = S^1\} \ge 1 - \sum_{i \in S_n^1} \sum_{j=1}^m \mathbb{P}(\theta_{ij} > \gamma_j) - \sum_{i \in S_n^2} \prod_{j \in \mathcal{E}_{t,i}^2} \mathbb{P}(\theta_{ij} \le \gamma_j). \tag{13}$$

The inequality holds because $0 < \prod_{j \in \mathcal{E}_{n,i}^1} \mathbb{P}(\theta_{ij} \le \gamma_j) \le 1$.

Let $\text{iKG}_{t,i}^{\text{F}}$ be the one-step improvement in (3) with $S_t^1$, $S_t^2$ and $\mathcal{E}_{t,i}^2$ treated as unchanged after one more sample and $\mathbb{E}[v_n(S_n)]$ approximated by the right-hand side of (13). We have the following proposition to compute $\text{iKG}_{t,i}^{\text{F}}$.

**Proposition 5.** *With the definition of $\text{iKG}_{t,i}^F$ above, we have*

$$\text{iKG}_{t,i}^F = \sum_{j=1}^m \bigg( \exp\bigg( -\frac{(\gamma_j - \mu_{t,ij})^2}{2\sigma_{t,ij}^2} \mathbf{1}\{i \in S_t^1\} \bigg) - \exp\bigg( -\frac{(\gamma_j - \mu_{t,ij})^2}{2(\sigma_{t+1,ij}^2 + \sigma_{ij}^2(\sigma_{t+1,ij}^2/\sigma_{ij}^2)^2)} \mathbf{1}\{i \in S_t^1\} \bigg) \bigg)$$

$$+ \exp\bigg( -\sum_{j \in \mathcal{E}_{t,i}^2} \frac{(\gamma_j - \mu_{t,ij})^2}{2\sigma_{t,ij}^2} \mathbf{1}\{i \in S_t^2\} \bigg) - \exp\bigg( -\sum_{j \in \mathcal{E}_{t,i}^2} \frac{(\gamma_j - \mu_{t,ij})^2}{2(\sigma_{t+1,ij}^2 + \sigma_{ij}^2(\sigma_{t+1,ij}^2/\sigma_{ij}^2)^2)} \mathbf{1}\{i \in S_t^2\} \bigg). \tag{14}$$

---

**Algorithm 4:** iKG-F Algorithm (Feasible Arm Identification)

**Input:** $k \ge 2$, $n$
1  Collect $n_0$ samples for each arm $i$;
2  **while** $t < n$ **do**
3      Compute $\text{iKG}_{t,i}^{\text{F}}$ and set $I_t = \text{argmax}_{i \in \mathbb{A}} \text{iKG}_{t,i}^{\text{F}}$;
4      Play $I_t$;
5      Update $\mu_{t+1,i}$, $\sigma_{t+1,i}$, $S_{t+1}^1$, $S_{t+1}^2$, $\mathcal{E}_{t+1,i}^1$ and $\mathcal{E}_{t+1,i}^2$;

**Output:** $S_n^1$

---

To identify the feasible arms, the iKG-F algorithm pulls the arm with the largest $\text{iKG}_{t,i}^{\text{F}}$ in each round. For this algorithm, we can show that the rate of posterior convergence of $1 - \mathbb{P}\{S_n^1 = S^1\}$ is also the fastest possible.

**Theorem 3.** *For the iKG-F algorithm, $\lim_{n \to \infty} -\frac{1}{n} \log(1 - \mathbb{P}\{S_n^1 = S^1\}) = \Gamma^F$, where*

$$\Gamma^F = w_i \min_{j \in \mathcal{E}_i^1} \frac{(\gamma_j - \mu_{ij})^2}{2\sigma_{ij}^2} \mathbf{1}\{i \in S^1\} + w_i \sum_{j \in \mathcal{E}_i^2} \frac{(\gamma_j - \mu_{ij})^2}{2\sigma_{ij}^2} \mathbf{1}\{i \in S^2\}, \tag{15}$$

*and $w_i$ is the sampling rate of arm $i$ satisfying*

$$\sum_{i=1}^k w_i = 1,$$

$$w_i \min_{j \in \mathcal{E}_i^1} \frac{(\gamma_j - \mu_{ij})^2}{2\sigma_{ij}^2} \mathbf{1}\{i \in S^1\} + w_i \sum_{j \in \mathcal{E}_i^2} \frac{(\gamma_j - \mu_{ij})^2}{2\sigma_{ij}^2} \mathbf{1}\{i \in S^2\} \tag{16}$$

$$= w_{i'} \min_{j \in \mathcal{E}_{i'}^1} \frac{(\gamma_j - \mu_{i'j})^2}{2\sigma_{i'j}^2} \mathbf{1}\{i' \in S^1\} + w_{i'} \sum_{j \in \mathcal{E}_{i'}^2} \frac{(\gamma_j - \mu_{i'j})^2}{2\sigma_{i'j}^2} \mathbf{1}\{i' \in S^2\}, \ \ i \ne i'.$$

*In addition, for any feasible arm identification algorithms*

$$\limsup_{n \to \infty} -\frac{1}{n} \log(1 - \mathbb{P}\{\mathcal{S}_n^1 = \mathcal{S}^1\}) \leq \Gamma^F.$$

## 5 Numerical Experiments

In this section, we show empirical performances of the iKG, iKG-$\epsilon$ and iKG-F algorithms on synthetic and real-world examples. For the best arm identification problem, we compare iKG with the following algorithms.

- Expected Improvement (EI) [11]. This is another common strategy for BAI. In each round, it pulls the arm offering the maximal expected improvement over the current estimate of the best mean of the arms.
- Top-Two Expected Improvement (TTEI) [12]. This is a modification of the EI algorithm by introducing a parameter $\beta$ to control the probabilities of sampling the best arm and the non-best set. We set the parameter $\beta$ in TTEI as its default value $1/2$.
- Knowledge Gradient. This is the algorithm under study in this research.

For the $\epsilon$-good arm identification problem, we compare iKG-$\epsilon$ with the following algorithms.

- APT Algorithm [27]. It is a fixed-budget algorithm for identifying the arms whose means are above a given threshold. We set the input tolerance parameter as $0.0001$ and the threshold as the posterior mean of the estimated best arm minus $\epsilon$.
- $(\text{ST})^2$ Algorithm [8]. It is a fixed-confidence algorithm for $\epsilon$-good arm identification. It pulls three arms in each round, the estimated best arm, one arm above the threshold and one arm below the threshold. We set the input tolerance parameter as $0.0001$ and $\gamma = 0$.

For the feasible arm identification problem, we compare iKG-F with the following algorithms.

- MD-UCBE Algorithm [10]. This is a fixed-budget algorithm for feasible arm identification based on the upper confidence bound. We set the input tolerance parameter as $0.0001$ and hyperparameter $a = \frac{25}{36} \frac{n-k}{H}$, where $H$ is a constant that can be computed. Katz-Samuels and Scott [10] showed that with $a = \frac{25}{36} \frac{n-k}{H}$, the performance of MD-UCBE is nearly optimal.
- MD-SAR Algorithm [10]. This is a fixed-budget algorithm for feasible arm identification based on successive accepts and rejects. We set the input tolerance parameter as $0.0001$.

In addition, iKG, iKG-$\epsilon$ and iKG-F will be compared with the equal allocation, where each arm is simply played with the same number of rounds. It is a naive method and is often used as a benchmark against which improvements might be measured.

The examples for testing include three synthetic examples, called Examples 1-3, and three real examples, namely the Dose-Finding Problem, Drug Selection Problem, and Caption Selection Problem. For Example 1-3 and the Dose-Finding problem, samples of the arms are two-dimensional. We call the measures of them measures 1 and 2. When the examples are tested for the best arm identification and $\epsilon$-good identification, only measure 1 will be used for identifying good/best arms. When the examples are tested for the feasible arm identification, both measures will be used for feasibility detection. For the Drug Selection and Caption Selection problems, samples of the arms are one-dimensional. They are tested for the best arm identification, $\epsilon$-good identification and feasible arm identification.

**Synthetic Datasets**. We consider three examples, all containing ten arms.

Example 1. The means in measure 1 of the ten arms are 0.1927, 0.6438, 3.0594, 3.0220, 1.3753, 1.4215, 0.9108, 1.0126, 0.1119 and 1.8808, and the means in measure 2 of the ten arms are 0.4350, 0.7240, 1.1566, 0.8560, 3.4712, 0.8248, 3.8797,1.9819, 3.2431 and 1.4315, all of which are uniformly generated in $(0, 4)$. Samples of the arms are corrupted by normal noises $\mathcal{N}(0, 1)$. The best arm is arm 3 and 0.1-good arms are arms 3 and 4. For the feasible arm identification, we choose arms with means in both measures less than 2. Then the feasible arms are arms 1, 2, 6, 8 and 10.

Example 2. We keep the setting of Example 1. Distributions of the noises for arms 1-5 are changed to $\mathcal{N}(0, 4)$.

Example 3. Consider functions $y_1(x) = -0.05x^2$, $y_2(x) = -0.06(7 - x)$ and $y_3(x) = 0.06(x - 6)$. The means in measure 1 of the ten arms are $y_1(x)$ with $x = 1, 2, \ldots, 10$. The means in measure 2 of the ten arms are $y_2(x)$ with $x = 1, \ldots, 6$ and $y_3(x)$ with $x = 7, \ldots, 10$. Noises follow the normal distribution $\mathcal{N}(0, 1)$. The best arm is arm 1 and 0.5-good arms are arms 1-3. For the feasible arm identification, we choose arms with means in measure 1 greater than $-0.5$ and means in measure 2 less than 0. The feasible arms are arms 1-3.

**Dose-Finding Problem**. We use the data in [28] (see ACR50 in week 16) for treating rheumatoid arthritis by the drug secukinumab. There are four dosage levels, 25mg, 75mg, 150mg, and 300mg, and a placebo, which are treated as five arms. We develop a simulation model based on the dataset. Each arm is associated with two performance measures: the probability of the drug being effective and the probability of the drug causing infections. The means of the five arms are $\boldsymbol{\mu_1} = (0.151, 0.259)$, $\boldsymbol{\mu_2} = (0.184, 0.184)$, $\boldsymbol{\mu_3} = (0.209, 0.209)$, $\boldsymbol{\mu_4} = (0.171, 0.293)$ and $\boldsymbol{\mu_5} = (0.06, 0.16)$. Samples of each arm are corrupted by normal noises $\mathcal{N}(0, 0.25)$. The best arm is arm 3 and the 0.03-good arms are arms 2 and 3. For the feasible arm identification, we find the arms whose probability of being effective is larger than 0.18 and the probability of causing infections is less than 0.25. The feasible arms are arms 2 and 3.

**Drug Selection Problem**. We consider five contraceptive alternatives based on the Drug Review Dataset (https://doi.org/10.24432/C5SK5S): Ethinyl estradiol / levonorgest, Ethinyl estradiol / norethindro, Ethinyl estradiol / norgestimat, Etonogestrel and Nexplanon, which can be treated as five arms. The dataset provides user reviews on the five drugs along with related conditions and ratings reflecting overall user satisfaction. We set the means of the five arms as $\mu_1 = 5.8676$, $\mu_2 = 5.6469$, $\mu_3 = 5.8765$, $\mu_4 = 5.8298$ and $\mu_5 = 5.6332$, and the variances of the five arms as $\sigma_1^2 = 3.2756$, $\sigma_2^2 = 3.4171$, $\sigma_3^2 = 3.2727$, $\sigma_4^2 = 3.3198$ and $\sigma_5^2 = 3.3251$, all calculated by the data. When this example is used for the best arm identification and $\epsilon$-good arm identification, the best arm (with the highest user satisfaction) and 0.003-good arm are both arm 3 (Ethinyl estradiol / norgestimat). When this example is used for feasible arm identification, we will select the drugs whose ratings are over 5.6, and the feasible arms are arm 1 (Ethinyl estradiol / levonorgest), arm 2 (Ethinyl estradiol / norethindro), arm 3 (Ethinyl estradiol / norgestimat), arm 4 (Etonogestrel) and arm 5 (Nexplanon).

**Caption Selection Problem**. We aim to select good captions based on the New Yorker Cartoon Caption Contest Dataset (https://nextml.github.io/caption-contest-data/). In the contests, each caption can be treated as an arm. The dataset provides the mean and variance of each arm, which can be used to set up our experiments. We will test contests 853 (Caption 853) and 854 (Caption 854).

In Caption 853, we randomly select ten captions as arms. We set the means of the ten arms as $\mu_1 = 1.1400$, $\mu_2 = 1.0779$, $\mu_3 = 1.4160$, $\mu_4 = 1.0779$, $\mu_5 = 1.1081$, $\mu_6 = 1.1467$, $\mu_7 = 1.1333$, $\mu_8 = 1.1075$, $\mu_9 = 1.1026$ and $\mu_{10} = 1.4900$, and the variances of the arms as $\sigma_1^2 = 0.1418$, $\sigma_2^2 = 0.0991$, $\sigma_3^2 = 0.4871$, $\sigma_4^2 = 0.0728$, $\sigma_5^2 = 0.0977$, $\sigma_6^2 = 0.1809$, $\sigma_7^2 = 0.1843$, $\sigma_8^2 = 0.0970$, $\sigma_9^2 = 0.0932$ and $\sigma_{10}^2 = 0.4843$, which are all calculated by the data. When this example is used for the best arm identification, the best arm (with the highest funniness score) is arm 10. When this example is used for $\epsilon$-good arm identification, the 0.1-good arms are arms 3 and 10. When this example is used for feasible arm identification, we will select the captions whose funniness scores are over 1.4, and the feasible arms are arms 3 and 10.

In Caption 854, we also randomly select ten captions as arms. We set the means of the ten arms as $\mu_1 = 1.1986$, $\mu_2 = 1.1890$, $\mu_3 = 1.1400$, $\mu_4 = 1.2621$, $\mu_5 = 1.1544$, $\mu_6 = 1.0339$, $\mu_7 = 1.1349$, $\mu_8 = 1.2786$, $\mu_9 = 1.1765$ and $\mu_{10} = 1.1367$, and the variances of the arms as $\sigma_1^2 = 0.1879$,

Table 1: Probabilities of false selection for the tested algorithms in best arm identification problem.

| | Example | Example 1 | | Example 2 | | Example 3 | | Dose-finding | | Drug Selection | | Caption 853 | | Caption 854 | |
|---|---|---|---|---|---|---|---|---|---|---|---|---|---|---|---|
| Algorithms | Sample size | 1000 | 5000 | 4400 | 18000 | 400 | 1000 | 1200 | 13000 | 2400 | 98000 | 1600 | 3000 | 12000 | 18000 |
| | Equal Allocation | 0.38 | 0.22 | 0.44 | 0.31 | 0.25 | 0.13 | 0.35 | 0.05 | 0.43 | 0.27 | 0.17 | 0.11 | 0.26 | 0.18 |
| | EI | 0.36 | 0.21 | 0.40 | 0.28 | 0.28 | 0.22 | 0.46 | 0.21 | 0.46 | 0.37 | 0.14 | 0.12 | 0.26 | 0.23 |
| BAI | TTEI | 0.25 | 0.07 | 0.32 | 0.09 | 0.13 | 0.02 | 0.31 | 0.03 | 0.55 | 0.28 | 0.04 | 0.01 | 0.10 | 0.06 |
| | KG | 0.29 | 0.14 | 0.32 | 0.13 | 0.14 | 0.03 | 0.40 | 0.03 | 0.44 | 0.28 | 0.04 | 0.01 | 0.11 | 0.05 |
| | iKG | 0.21 | 0.03 | 0.23 | 0.03 | 0.09 | 0.01 | 0.29 | 0.01 | 0.38 | 0.23 | 0.02 | 0.00 | 0.07 | 0.04 |

Table 2: Probabilities of false selection for the tested algorithms in $\epsilon$-good arm identification problem.

| Example | | Example 1 | | Example 2 | | Example 3 | | Dose-finding | | Drug Selection | | Caption 853 | | Caption 854 | |
|---|---|---|---|---|---|---|---|---|---|---|---|---|---|---|---|
| Sample size / Algorithms | | 1000 | 4000 | 2400 | 12000 | 400 | 4000 | 1600 | 6000 | 2600 | 90000 | 4000 | 10000 | 9400 | 15000 |
| $\epsilon$-good | Equal Allocation | 0.54 | 0.20 | 0.65 | 0.28 | 0.61 | 0.26 | 0.46 | 0.18 | 0.62 | 0.37 | 0.28 | 0.19 | 0.14 | 0.05 |
| | APT | 0.28 | 0.17 | 0.52 | 0.25 | 0.72 | 0.49 | 0.56 | 0.53 | 0.74 | 0.70 | 0.41 | 0.35 | 0.48 | 0.49 |
| | $(ST)^2$ | 0.29 | 0.07 | 0.35 | 0.11 | 0.51 | 0.06 | 0.38 | 0.17 | 0.64 | 0.34 | 0.21 | 0.10 | 0.12 | 0.04 |
| | iKG-$\epsilon$ | 0.17 | 0.03 | 0.29 | 0.00 | 0.48 | 0.03 | 0.34 | 0.06 | 0.60 | 0.27 | 0.10 | 0.02 | 0.11 | 0.03 |

Table 3: Probabilities of false selection for the tested algorithms in feasible arm identification problem.

| Example | | Example 1 | | Example 2 | | Example 3 | | Dose-finding | | Drug Selection | | Caption 853 | | Caption 854 | |
|---|---|---|---|---|---|---|---|---|---|---|---|---|---|---|---|
| Sample size / Algorithms | | 3400 | 11000 | 4800 | 14000 | 2200 | 4800 | 2000 | 4000 | 100000 | 140000 | 4000 | 10000 | 30600 | 44000 |
| feasible arm | Equal Allocation | 0.34 | 0.26 | 0.33 | 0.23 | 0.22 | 0.14 | 0.22 | 0.18 | 0.03 | 0.03 | 0.36 | 0.29 | 0.18 | 0.07 |
| | MD-UCBE | 0.27 | 0.16 | 0.33 | 0.26 | 0.05 | 0.01 | 0.20 | 0.17 | 0.06 | 0.06 | 0.32 | 0.15 | 0.06 | 0.04 |
| | MD-SAR | 0.74 | 0.33 | 0.68 | 0.22 | 0.30 | 0.03 | 0.79 | 0.55 | 0.06 | 0.02 | 0.58 | 0.19 | 0.08 | 0.05 |
| | iKG-F | 0.23 | 0.02 | 0.24 | 0.01 | 0.04 | 0.00 | 0.14 | 0.01 | 0.01 | 0.01 | 0.20 | 0.07 | 0.05 | 0.00 |

$\sigma_2^2 = 0.2279$, $\sigma_3^2 = 0.1346$, $\sigma_4^2 = 0.3186$, $\sigma_5^2 = 0.1314$, $\sigma_6^2 = 0.0330$, $\sigma_7^2 = 0.1337$, $\sigma_8^2 = 0.3167$, $\sigma_9^2 = 0.1858$ and $\sigma_{10}^2 = 0.1478$, all calculated by the data. When this example is used for the best arm identification, the best arm is arm 8. When this example is used for $\epsilon$-good arm identification, the 0.05-good arms are arms 4 and 8. When this example is used for feasible arm identification, we will select the captions whose funniness scores are over 1.25, and the feasible arms are arms 4 and 8.

For the tested algorithms, probabilities of false selection (PFS) are obtained based on the average of 100 macro-replications. Tables 1-3 show the PFS of the algorithms under some fixed sample sizes (additional numerical results about the PFS and sampling rates of the tested algorithms are provided in the Supplement). The proposed iKG, iKG-$\epsilon$ and iKG-F perform the best. For the best arm identification, EI tends to allocate too many samples to the estimated best arm, leading to insufficient exploration in the remaining arms, while KG tends to allocate too few samples to the estimated best arm, leading to excessive exploration in the remaining arms. TTEI always allocates approximately one-half budget to the estimated best arm when $\beta = 1/2$, leading to the budget not being the best utilized. For the $\epsilon$-good identification, APT and $(ST)^2$ are inferior because the former insufficiently pulls the estimate best arm, leading to inaccurate estimates of the threshold, while the latter falls in the fixed-confidence regime that focuses on making guarantees on the probability of false selection instead of minimizing it. For the feasible arm identification, both MD-UCBE and MD-SAR allocate too many samples to the arms near the constraint limits. For the three problems, equal allocation performs the worst in general, because it does not have any efficient sampling mechanisms for identifying the target arms in these problems.

# 6   Conclusion

This paper studies the knowledge gradient (KG), a popular policy for the best arm identification (BAI). We observe that the KG algorithm is not asymptotically optimal, and then propose a remedy for it. The new policy follows KG's manner of one-step look ahead, but utilizes different evidence to identify the best arm. We call it improved knowledge gradient (iKG) and show that it is asymptotically optimal. Another advantage of iKG is that it can be easily extended to variant problems of BAI. We use $\epsilon$-good arm identification and feasible arm identification as two examples for algorithm development and analysis. The superior performances of iKG on BAI and the two variants are further demonstrated using numerical examples.

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
