# Supplement to "Improving the Knowledge Gradient Algorithm"

**Le Yang**
Department of Systems Engineering
City University of Hong Kong
lyang272-c@my.cityu.edu.hk

**Siyang Gao**
Department of Systems Engineering
City University of Hong Kong
siyangao@cityu.edu.hk

**Chin Pang Ho**
School of Data Science
City University of Hong Kong
clint.ho@cityu.edu.hk

## A  Proof of Proposition 1

To facilitate the analysis, we make the following definition. For two real-valued sequences $\{a_n\}$ and $\{b_n\}$, if $\lim_{n\to\infty} 1/n \log(a_n/b_n) = 0$, we call them logarithmically equivalent, denoted by $a_n \dot{=} b_n$. We first analyze $1 - \mathbb{P}\{I_n^* = I^*\}$. Note that $1 - \mathbb{P}\{I_n^* = I^*\} = \mathbb{P}\left\{ \bigcup_{i \neq I_n^*} (\theta_i > \theta_{I_n^*}) \right\}$ and we have

$$\max_{i \neq I_n^*} \mathbb{P}(\theta_i > \theta_{I_n^*}) \leq \mathbb{P}\left\{ \bigcup_{i \neq I_n^*} (\theta_i > \theta_{I_n^*}) \right\} \leq (k-1) \max_{i \neq I_n^*} \mathbb{P}(\theta_i > \theta_{I_n^*}).$$

Then $1 - \mathbb{P}\{I_n^* = I^*\} \dot{=} \max_{i \neq I_n^*} \mathbb{P}(\theta_i > \theta_{I_n^*})$. In round $n$, $\theta_i - \theta_{I_n^*}$ follows $\mathcal{N}(\mu_{n,i} - \mu_{n,I_n^*}, \sigma_{n,i}^2 + \sigma_{n,I_n^*}^2)$. Let $\Phi(\cdot)$ and $\phi(\cdot)$ be the cumulative density function and probability density function of the standard normal distribution, respectively. We have

$$\mathbb{P}(\theta_i > \theta_{I_n^*}) = 1 - \Phi\left( \frac{\mu_{n,I_n^*} - \mu_{n,i}}{\sqrt{\sigma_{n,i}^2 + \sigma_{n,I_n^*}^2}} \right).$$

Let $z = \frac{\mu_{n,I_n^*} - \mu_{n,i}}{\sqrt{\sigma_{n,i}^2 + \sigma_{n,I_n^*}^2}}$ and $z > 0$. By the following property of the cumulative probability function of the standard normal distribution

$$\frac{z}{(z^2 + 1)} < 1 - \Phi(z) < \frac{1}{z}\phi(z)$$

and $\mu_{n,I_n^*} - \mu_{n,i} > 0$, we have

$$\mathbb{P}(\theta_i > \theta_{I_n^*}) \dot{=} \phi\left( \frac{\mu_{n,i} - \mu_{n,I_n^*}}{\sqrt{\sigma_{n,i}^2 + \sigma_{n,I_n^*}^2}} \right) \dot{=} \exp\left( -\frac{(\mu_{n,i} - \mu_{n,I_n^*})^2}{2(\sigma_{n,i}^2 + \sigma_{n,I_n^*}^2)} \right). \tag{A.1}$$

Denote $T_{n,i}$ as the number of samples for arm $i$ before round $t$, i.e., $T_{t,i} \triangleq \sum_{l=0}^{t-1} \mathbf{1}\{I_l = i\}$. By (1) in the main text, we have

$$\sigma_{n,i}^2 = \frac{1}{(\sigma_i^2/T_{n,i})^{-1} + \sigma_{0,i}^{-2}}.$$

37th Conference on Neural Information Processing Systems (NeurIPS 2023).

Then

$$1 - \mathbb{P}\{I_n^* = I^*\} \doteq \max_{i \neq I_n^*} \left( \exp \left( -\frac{(\mu_{n,i} - \mu_{n,I_n^*})^2}{2(\sigma_{n,i}^2 + \sigma_{t,I_n^*}^2)} \right) \right)$$

$$\doteq \exp \left( -n \min_{i \neq I_n^*} \frac{(\mu_{n,i} - \mu_{n,I_n^*})^2}{2(\sigma_{n,i}^2 + \sigma_{t,I_n^*}^2)} \right).$$

Hence

$$\Gamma^{\text{KG}} = \lim_{n \to \infty} -\frac{1}{n} \log(1 - \mathbb{P}\{I_n^* = I^*\}) = \min_{i \neq I^*} \frac{(\mu_i - \mu_{I^*})^2}{2(\sigma_i^2/w_i + \sigma_{I^*}^2/w_{I^*})}. \tag{A.2}$$

Notice that the sampling rate $w_i$ of each arm $i$ of the KG algorithm has been characterized in [1], with

$$\frac{w_{\langle 1 \rangle}}{w_{\langle 2 \rangle}} = \frac{\sigma_{\langle 1 \rangle}}{\sigma_{\langle 2 \rangle}} \quad \text{and} \quad \frac{w_{\langle i \rangle}}{w_{\langle i' \rangle}} = \frac{(\mu_{\langle 1 \rangle} - \mu_{\langle i' \rangle})/\sigma_{\langle i' \rangle}}{(\mu_{\langle 1 \rangle} - \mu_{\langle i \rangle})/\sigma_{\langle i \rangle}}, \quad i, i' = 2, 3, \dots, k \text{ and } i \neq i'.$$

Together with $\sum_{i=1}^k w_i = 1$, we have

$$w_{\langle 1 \rangle} = \left( \frac{\sigma_{\langle 2 \rangle}}{\sigma_{\langle 1 \rangle}} \sum_{i \neq 1} \frac{1}{c_{\langle i \rangle}} + 1 \right)^{-1} \tag{A.3}$$

and

$$w_{\langle i \rangle} = \left( c_{\langle i \rangle} \left( \sum_{i \neq 1} \frac{1}{c_{\langle i \rangle}} + \frac{\sigma_{\langle 1 \rangle}}{\sigma_{\langle 2 \rangle}} \right) \right)^{-1}, \quad i = 2, 3, \dots, k. \tag{A.4}$$

Plugging into (A.2),

$$\Gamma^{\text{KG}} = \min_{i \neq 1} \left( \frac{(\mu_{\langle i \rangle} - \mu_{\langle 1 \rangle})^2}{2((\sum_{i \neq 1} \sigma_{\langle 2 \rangle}/c_{\langle i \rangle} + \sigma_{\langle 1 \rangle})\sigma_{\langle 1 \rangle} + c_{\langle i \rangle}\sigma_{\langle i \rangle}^2(\sum_{i \neq 1} 1/c_{\langle i \rangle} + \sigma_{\langle 1 \rangle}/\sigma_{\langle 2 \rangle}))} \right).$$

## B   Proof of Proposition 2

Similar to the proof of Proposition 1, we have

$$\Gamma^{\text{TTEI}} = \lim_{n \to \infty} -\frac{1}{n} \log(1 - \mathbb{P}\{I_n^* = I^*\}) = \min_{i \neq I^*} \frac{(\mu_i - \mu_{I^*})^2}{2(\sigma_i^2/w_i + \sigma_{I^*}^2/w_{I^*})}.$$

Since for the TTEI algorithm,

$$\frac{(\mu_i - \mu_{I^*})^2}{2(\sigma_i^2/w_i + \sigma_{I^*}^2/w_{I^*})} = \frac{(\mu_{i'} - \mu_{I^*})^2}{2(\sigma_{i'}^2/w_{i'} + \sigma_{I^*}^2/w_{I^*})}, \quad \forall i \neq i' \neq I^*,$$

we have

$$\Gamma^{\text{TTEI}} = \frac{(\mu_i - \mu_{I^*})^2}{2(\sigma_i^2/w_i + \sigma_{I^*}^2/w_{I^*})} \quad \forall i \neq I^*.$$

According to (A.3), for the KG algorithm, $w_{\langle 1 \rangle} = (\sigma_{\langle 2 \rangle}/\sigma_{\langle 1 \rangle} \sum_{i \neq I^*} 1/c_{\langle i \rangle} + 1)^{-1}$. Now by setting $\beta$ of the TTEI algorithm to the same value, the sampling rates of the best arm from these two algorithms will be the same. According to Theorem 2 of [2], among algorithms allocating the same proportion of the samples to the best arm, $\Gamma^{\text{TTEI}}$ of the TTEI algorithm is optimal, i.e., $\Gamma^{\text{KG}} \leq \Gamma^{\text{TTEI}}$.

## C   Proof of Propositions 3, 4 and 5

Propositions 3, 4 and 5 give the expressions of $\text{iKG}_{t,i}$, $\text{iKG}_{t,i}^\epsilon$ and $\text{iKG}_{t,i}^{\text{F}}$. Below we introduce a lemma first, which will be used in the proofs of the three propositions.

**Lemma C.1.** *If arm $i$ is sampled from $\mathcal{N}(\mu_{t,i}, \sigma_i^2)$ in round $t$, $\theta_i$ and $\theta_{i'}$ follow $\mathcal{N}(\mu_{t,i}, \sigma_{t,i}^2)$ and $\mathcal{N}(\mu_{t,i'}, \sigma_{t,i'}^2)$ respectively. Then,*

$$\mathbb{E}[\mathbb{P}(\theta_i > \theta_{i'})] = \exp \left( -\frac{(\mu_{t,i} - \mu_{t,i'})^2}{2(\sigma_{t+1,i}^2 + \sigma_{t,i'}^2 + \sigma_i^2(\sigma_{t+1,i}^2/\sigma_i^2)^2)} \right).$$

Proof of Lemma C.1:

We know that $\theta_{t,i}$ follows $\mathcal{N}(\mu_{t,i}, \sigma_i^2)$. Then by (1) of the main text, we have

$$\mu_{t+1,i} = \begin{cases} \dfrac{\sigma_{t,i}^{-2}\mu_{t,i} + \sigma_i^{-2}\theta_{t,i}}{\sigma_{t,i}^{-2} + \sigma_i^{-2}} & \text{if } I_t = i, \\ \mu_{t,i} & \text{if } I_t \neq i, \end{cases} \quad \text{and} \quad \sigma_{t+1,i}^2 = \begin{cases} \dfrac{1}{\sigma_{t,i}^{-2} + \sigma_i^{-2}} & \text{if } I_t = i, \\ \sigma_{t,i}^2 & \text{if } I_t \neq i. \end{cases}$$

Recall that

$$\mathbb{P}(\theta_i > \theta_{I_t^*}) \doteq \exp\left(-\frac{(\mu_{t,i} - \mu_{t,I_t^*})^2}{2(\sigma_{t,i}^2 + \sigma_{t,I_t^*}^2)}\right).$$

Then

$$\mathbb{E}[\mathbb{P}(\theta_i > \theta_{i'})] = \mathbb{E}\left[\exp\left(-\frac{(\mu_{t+1,i} - \mu_{t,i'})^2}{2(\sigma_{t+1,i}^2 + \sigma_{t,i'}^2)}\right)\right]$$

$$= \frac{1}{\sqrt{2\pi}\sigma_i} \int_{-\infty}^{\infty} \exp\left(-\frac{(\frac{\sigma_{t,i}^{-2}\mu_{t,i} + \sigma_i^{-2}\theta_{t,i}}{\sigma_{t+1,i}^{-2}} - \mu_{t,i'})^2}{2(\sigma_{t+1,i}^2 + \sigma_{t,i'}^2)}\right) \exp\left(-\frac{(\theta_{t,i} - \mu_{t,i'})^2}{2\sigma_i^2}\right) d\theta_{t,i}$$

$$\doteq \exp\left(-\frac{(\mu_{t,i} - \mu_{t,i'})^2}{2(\sigma_{t+1,i}^2 + \sigma_{t,i'}^2 + \sigma_i^2(\sigma_{t+1,i}^2/\sigma_i^2)^2)}\right).$$

Proof of Proposition 3:

For the best arm identification problem, if $i \neq I_t^*$,

$$\text{iKG}_{t,i} = \mathbb{E}[v_n(\mathcal{T}(S_t, i, \theta_{t,i})) - v_n(S_t)]$$

$$= 1 - \sum_{i' \neq i \neq I_t^*} \exp\left(-\frac{(\mu_{t,i'} - \mu_{t,I_t^*})^2}{2(\sigma_{t,i'}^2 + \sigma_{t,I_t^*}^2)}\right) - \exp\left(-\frac{(\mu_{t,i} - \mu_{t,I_t^*})^2}{2(\sigma_{t+1,i}^2 + \sigma_{t,I_t^*}^2 + \sigma_i^2(\sigma_{t+1,i}^2/\sigma_i^2)^2)}\right)$$

$$- \left(1 - \sum_{i' \neq I_t^*} \exp\left(-\frac{(\mu_{t,i'} - \mu_{t,I_t^*})^2}{2(\sigma_{t,i'}^2 + \sigma_{t,I_t^*}^2)}\right)\right)$$

$$= \exp\left(-\frac{(\mu_{t,i} - \mu_{t,I_t^*})^2}{2(\sigma_{t,i}^2 + \sigma_{t,I_t^*}^2)}\right) - \exp\left(-\frac{(\mu_{t,i} - \mu_{t,I_t^*})^2}{2(\sigma_{t+1,i}^2 + \sigma_{t,I_t^*}^2 + \sigma_i^2(\sigma_{t+1,i}^2/\sigma_i^2)^2)}\right).$$

If $i = I_t^*$,

$$\text{iKG}_{t,i} = \mathbb{E}[v_n(\mathcal{T}(S_t, i, \theta_{t,i})) - v_n(S_t)]$$

$$= 1 - \sum_{i' \neq I_t^*} \exp\left(-\frac{(\mu_{t,i'} - \mu_{t,I_t^*})^2}{2(\sigma_{t,i'}^2 + \sigma_{t+1,I_t^*}^2 + \sigma_{I_t^*}^2(\sigma_{t+1,I_t^*}^2/\sigma_{I_t^*}^2)^2)}\right) - \left(1 - \sum_{i' \neq I_t^*} \exp\left(-\frac{(\mu_{t,i'} - \mu_{t,I_t^*})^2}{2(\sigma_{t,i'}^2 + \sigma_{t,I_t^*}^2)}\right)\right)$$

$$= \sum_{i' \neq I_t^*} \exp\left(-\frac{(\mu_{t,i'} - \mu_{t,I_t^*})^2}{2(\sigma_{t,i'}^2 + \sigma_{t,I_t^*}^2)}\right) - \sum_{i' \neq I_t^*} \exp\left(-\frac{(\mu_{t,i'} - \mu_{t,I_t^*})^2}{2(\sigma_{t,i'}^2 + \sigma_{t+1,I_t^*}^2 + \sigma_{I_t^*}^2(\sigma_{t+1,I_t^*}^2/\sigma_{I_t^*}^2)^2)}\right).$$

Proof of Proposition 4:

We explore the expression of $\mathbb{E}[v_n(S_n)]$ in the $\epsilon$-good arm identification problem first. We know that

$$\mathbb{E}[v_n(S_n)] = 1 - \sum_{i \in G_n^\epsilon} \mathbb{P}(\theta_i < \max_{i' \in \mathbb{A}} \theta_{i'} - \epsilon) - \sum_{i \in \mathbb{A} \setminus G_n^\epsilon} \mathbb{P}(\theta_i > \max_{i' \in \mathbb{A}} \theta_{i'} - \epsilon).$$

Note that in round $n$, $\theta_i - \theta_{I_n^*} + \epsilon$ follows $\mathcal{N}(\mu_{n,i} - \mu_{n,I_n^*} + \epsilon, \sigma_{n,i}^2 + \sigma_{n,I_n^*}^2)$. Similarly as in the proof of Proposition 1, we can know that if $i \in G_n^\epsilon$

$$\mathbb{P}(\theta_i < \max_{i' \in \mathbb{A}} \theta_{i'} - \epsilon) \doteq \exp\left(-\frac{(\mu_{n,i} - \mu_{n,I_n^*} + \epsilon)^2}{2(\sigma_{n,i}^2 + \sigma_{n,I_n^*}^2)}\right),$$

and if $i \in \mathbb{A} \setminus G_n^\epsilon$

$$\mathbb{P}(\theta_i > \max_{i' \in \mathbb{A}} \theta_{i'} - \epsilon) \doteq \exp\left(-\frac{(\mu_{n,i} - \mu_{n,I_n^*} + \epsilon)^2}{2(\sigma_{n,i}^2 + \sigma_{n,I_n^*}^2)}\right).$$

Then

$$\mathbb{E}[v_n(S_n)] = 1 - \sum_{i \neq I_n^*} \exp\left(-\frac{(\mu_{n,i} - \mu_{n,I_n^*} + \epsilon)^2}{2(\sigma_{n,i}^2 + \sigma_{n,I_n^*}^2)}\right).$$

For the $\epsilon$-good arm identification problem, if $i \neq I_t^*$,

$$\text{iKG}_{t,i}^\epsilon = \mathbb{E}[v_n(\mathcal{T}(S_t, i, \theta_{t,i})) - v_n(S_t)]$$

$$= 1 - \sum_{i' \neq i \neq I_t^*} \exp\left(-\frac{(\mu_{t,i'} - \mu_{t,I_t^*} + \epsilon)^2}{2(\sigma_{t,i'}^2 + \sigma_{t,I_t^*}^2)}\right) - \exp\left(-\frac{(\mu_{t,i} - \mu_{t,I_t^*} + \epsilon)^2}{2(\sigma_{t+1,i}^2 + \sigma_{t,I_t^*}^2 + \sigma_i^2(\sigma_{t+1,i}^2/\sigma_i^2)^2)}\right)$$

$$- \left(1 - \sum_{i' \neq I_t^*} \exp\left(-\frac{(\mu_{t,i'} - \mu_{t,I_t^*} + \epsilon)^2}{2(\sigma_{t,i'}^2 + \sigma_{t,I_t^*}^2)}\right)\right)$$

$$= \exp\left(-\frac{(\mu_{t,i} - \mu_{t,I_t^*} + \epsilon)^2}{2(\sigma_{t,i}^2 + \sigma_{t,I_t^*}^2)}\right) - \exp\left(-\frac{(\mu_{t,i} - \mu_{t,I_t^*} + \epsilon)^2}{2(\sigma_{t+1,i}^2 + \sigma_{t,I_t^*}^2 + \sigma_i^2(\sigma_{t+1,i}^2/\sigma_i^2)^2)}\right).$$

If $i = I_t^*$,

$$\text{iKG}_{t,i}^\epsilon = \mathbb{E}[v_n(\mathcal{T}(S_t, i, \theta_{t,i})) - v_n(S_t)]$$

$$= 1 - \sum_{i' \neq I_t^*} \exp\left(-\frac{(\mu_{t,i'} - \mu_{t,I_t^*} + \epsilon)^2}{2(\sigma_{t,i'}^2 + \sigma_{t+1,I_t^*}^2 + \sigma_{I_t^*}^2(\sigma_{t+1,I_t^*}^2/\sigma_{I_t^*}^2)^2)}\right) - \left(1 - \sum_{i' \neq I_t^*} \exp\left(-\frac{(\mu_{t,i'} - \mu_{t,I_t^*} + \epsilon)^2}{2(\sigma_{t,i'}^2 + \sigma_{t,I_t^*}^2)}\right)\right)$$

$$= \sum_{i' \neq I_t^*} \exp\left(-\frac{(\mu_{t,i'} - \mu_{t,I_t^*} + \epsilon)^2}{2(\sigma_{t,i'}^2 + \sigma_{t,I_t^*}^2)}\right) - \sum_{i' \neq I_t^*} \exp\left(-\frac{(\mu_{t,i'} - \mu_{t,I_t^*} + \epsilon)^2}{2(\sigma_{t,i'}^2 + \sigma_{t+1,I_t^*}^2 + \sigma_{I_t^*}^2(\sigma_{t+1,I_t^*}^2/\sigma_{I_t^*}^2)^2)}\right).$$

Proof of Proposition 5:
We explore the expression of $\mathbb{E}[v_n(S_n)]$ in the feasible arm identification problem first. We know that

$$\mathbb{E}[v_n(S_n)] = 1 - \sum_{i \in \mathcal{S}_n^1} \sum_{j=1}^m \mathbb{P}(\theta_{ij} > \gamma_j) - \sum_{i \in \mathcal{S}_n^2} \prod_{j \in \mathcal{E}_{t,i}^2} \mathbb{P}(\theta_{ij} \leq \gamma_j).$$

Note that in round $n$, $\theta_{ij} - \gamma_j$ follows $\mathcal{N}(\mu_{n,i} - \gamma_j, \sigma_{n,ij}^2)$. Similarly as in the proof of Proposition 1, we can know that if $i \in \mathcal{S}_n^1$ and measure $j \in \{1, 2, \ldots, m\}$,

$$\mathbb{P}(\theta_{ij} > \gamma_j) \doteq \exp\left(-\frac{(\gamma_j - \mu_{n,ij})^2}{2\sigma_{n,ij}^2}\right),$$

and if $i \in \mathcal{S}_n^2$ and measure $j \in \mathcal{E}_{n,i}^2$,

$$\mathbb{P}(\theta_{ij} \leq \gamma_j) \doteq \exp\left(-\frac{(\gamma_j - \mu_{n,ij})^2}{2\sigma_{n,ij}^2}\right).$$

Then

$$\mathbb{E}[v_n(S_n)] = 1 - \sum_{i \in \mathcal{S}_n^1} \sum_{j=1}^m \exp\left(-\frac{(\gamma_j - \mu_{n,ij})^2}{2\sigma_{n,ij}^2}\right) - \sum_{i \in \mathcal{S}_n^2} \exp\left(-\sum_{j \in \mathcal{E}_{n,i}^2} \frac{(\gamma_j - \mu_{n,ij})^2}{2\sigma_{n,ij}^2}\right).$$

For the feasible arm identification problem,

$$
\mathrm{iKG}^{\mathrm{F}}_{t,i} = \mathbb{E}[v_n(\mathcal{T}(S_t, i, \theta_{t,i})) - v_n(S_t)]
$$

$$
\begin{aligned}
=1 &- \sum_{i' \neq i \in \mathcal{S}^1_t} \sum_{j=1}^m \exp\left(-\frac{(\gamma_j - \mu_{t,i'j})^2}{2\sigma^2_{t,i'j}}\right) - \sum_{i' \neq i \in \mathcal{S}^2_t} \exp\left(-\sum_{j \in \mathcal{E}^2_{t,i'}} \frac{(\gamma_j - \mu_{t,i'j})^2}{2\sigma^2_{t,i'j}}\right) \\
&- \sum_{j=1}^m \exp\left(-\frac{(\gamma_j - \mu_{t,ij})^2}{2(\sigma^2_{t+1,ij} + \sigma^2_{ij}(\sigma^2_{t+1,ij}/\sigma^2_{ij})^2)}\right)\mathbf{1}\{i \in \mathcal{S}^1_t\} - \exp\left(-\sum_{j \in \mathcal{E}^2_{t,i}} \frac{(\gamma_j - \mu_{t,ij})^2}{2(\sigma^2_{t+1,ij} + \sigma^2_{ij}(\sigma^2_{t+1,ij}/\sigma^2_{ij})^2)}\right)\mathbf{1}\{i \in \mathcal{S}^2_t\} \\
&- \left(1 - \sum_{i \in \mathcal{S}^1_t} \sum_{j=1}^m \exp\left(-\frac{(\gamma_j - \mu_{t,ij})^2}{2\sigma^2_{t,ij}}\right) - \sum_{i \in \mathcal{S}^2_t} \exp\left(-\sum_{j \in \mathcal{E}^2_{t,i}} \frac{(\gamma_j - \mu_{t,ij})^2}{2\sigma^2_{t,ij}}\right)\right)
\end{aligned}
$$

$$
\begin{aligned}
=\sum_{j=1}^m &\left(\exp\left(-\frac{(\gamma_j - \mu_{t,ij})^2}{2\sigma^2_{t,ij}}\mathbf{1}\{i \in \mathcal{S}^1_t\}\right) - \exp\left(-\frac{(\gamma_j - \mu_{t,ij})^2}{2(\sigma^2_{t+1,ij} + \sigma^2_{ij}(\sigma^2_{t+1,ij}/\sigma^2_{ij})^2)}\mathbf{1}\{i \in \mathcal{S}^1_t\}\right)\right) \\
&+ \exp\left(-\sum_{j \in \mathcal{E}^2_{t,i}} \frac{(\gamma_j - \mu_{t,ij})^2}{2\sigma^2_{t,ij}}\mathbf{1}\{i \in \mathcal{S}^2_t\}\right) - \exp\left(-\sum_{j \in \mathcal{E}^2_{t,i}} \frac{(\gamma_j - \mu_{t,ij})^2}{2(\sigma^2_{t+1,ij} + \sigma^2_{ij}(\sigma^2_{t+1,ij}/\sigma^2_{ij})^2)}\mathbf{1}\{i \in \mathcal{S}^2_t\}\right).
\end{aligned}
$$

## D   Proof of Theorem 1

Our proof of Theorem 1 will be divided into the analysis of the consistency, sampling rates and asymptotic optimality of the iKG algorithm.

We first show the consistency, i.e., each arm will be pulled infinitely by the algorithm as the round $n$ goes to infinity. Since

$$
\mathrm{iKG}_{t,i} = \begin{cases}
\exp\left(-\frac{(\mu_{t,i} - \mu_{t,I^*_t})^2}{2(\sigma^2_i/T_{t,i} + \sigma^2_{I^*_t}/T_{t,I^*_t})}\right) - \exp\left(-\frac{(\mu_{t,i} - \mu_{t,I^*_t})^2}{2((T_{t,i}+2)\sigma^2_i/(T_{t,i}+1)^2 + \sigma^2_{I^*_t}/T_{t,I^*_t})}\right), & \text{if } i \neq I^*_t, \\
\sum_{i' \neq I^*_t} \exp\left(-\frac{(\mu_{t,i'} - \mu_{t,I^*_t})^2}{2(\sigma^2_{i'}/T_{t,i'} + \sigma^2_{I^*_t}/T_{t,I^*_t})}\right) - \sum_{i' \neq I^*_t} \exp\left(-\frac{(\mu_{t,i'} - \mu_{t,I^*_t})^2}{2(\sigma^2_{i'}/T_{t,i'} + (T_{t,I^*_t}+2)\sigma^2_{I^*_t}/(T_{t,I^*_t}+1)^2)}\right), & \text{if } i = I^*_t,
\end{cases}
$$

$$(\text{D.1})$$

it is obvious that $\mathrm{iKG}_{t,i} > 0$ for $t > 0$. To prove the consistency, we define a set $V \triangleq \{i \in \mathbb{A} : \sum_{l \geq 0} \mathbf{1}\{I_l = i\} < \infty\}$. It suffices to prove that $V = \emptyset$, and then the claim is straightforward based on the Strong Law of Large Numbers. For any $\delta_1 > 0$ and arm $i \notin V$, there exists $N_1$ such that when $n > N_1$, $|\mu_{n,i} - \mu_i| < \delta_1$, because arms not in $V$ will be infinitely pulled. Since the $\exp(\cdot)$ is a continuous function and $\sigma^2_i/T_{t,i} - \sigma^2_i(T_{t,i}+2)/(T_{t,i}+1)^2 = \sigma^2_i/((T_{t,i}+1)^2 T_{t,i}) \to 0$ holds for arm $i \notin V$, then for any $\delta_2 > 0$, there exists $N_2$ such that when $n > N_2$, $\mathrm{iKG}_{t,i} < \delta_2$.

Arms $i' \in V$ are pulled for only a finite number of rounds. Then $\max_{i' \in V} T_{t,i'}$ exists and we have $\sigma^2_{i'}/((T_{t,i'}+1)^2 T_{t,i'}) > \min_{i' \neq I^*_t} \sigma^2_{i'}/\max_{i' \in V}(T_{t,i'}+2)/(T_{t,i'}+1)^2$. According to the continuity of the function $\exp(\cdot)$, there exists $\delta_3 > 0$ such that $\mathrm{iKG}_{t,i'} > \delta_3$. Since $\delta_2$ is arbitrary, let $\delta_2 < \delta_3$, and then $\mathrm{iKG}_{t,i'} > \mathrm{iKG}_{t,i}$ holds, which implies $I_t \in V$. As the total number of rounds tend to infinity, $V$ will become an empty set eventually. In other words, all the arms will be pulled infinitely and $I^*_n = I^* = \langle 1 \rangle$ holds with probability 1.

We next analyze the sampling rate of each arm by the iKG algorithm. Let $\delta_4 = 2\delta_2 > 0$, we know that when $n$ is large, $\mathrm{iKG}_{n,i} < \delta_2 = \delta_4/2$ for all $i \in \mathbb{A}$. Then $|\mathrm{iKG}_{n,i} - \mathrm{iKG}_{n,i'}| < \mathrm{iKG}_{n,i} + \mathrm{iKG}_{n,i'} <$

$\delta_4/2 + \delta_4/2 = \delta_4$, where $i \neq i'$. For any $i, i' \in \mathbb{A}$ and $i \neq i' \neq \langle 1 \rangle$,

$$
\left| \text{iKG}_{n,i} - \text{iKG}_{n,i'} \right|
$$

$$
= \left| \exp\left( -\frac{(\mu_{n,i} - \mu_{n,\langle 1 \rangle})^2}{2(\sigma_i^2/T_{n,i} + \sigma_{\langle 1 \rangle}^2/T_{n,\langle 1 \rangle})} \right) - \exp\left( -\frac{(\mu_{n,i'} - \mu_{n,\langle 1 \rangle})^2}{2(\sigma_{i'}^2/T_{n,i'} + \sigma_{\langle 1 \rangle}^2/T_{n,\langle 1 \rangle})} \right) \right.
$$

$$
\left. + \exp\left( -\frac{(\mu_{n,i'} - \mu_{n,\langle 1 \rangle})^2}{2((T_{n,i'}+2)\sigma_{i'}^2/(T_{n,i'}+1)^2 + \sigma_{\langle 1 \rangle}^2/T_{n,\langle 1 \rangle})} \right) - \exp\left( -\frac{(\mu_{n,i} - \mu_{n,\langle 1 \rangle})^2}{2((T_{n,i}+2)\sigma_i^2/(T_{n,i}+1)^2 + \sigma_{\langle 1 \rangle}^2/T_{n,\langle 1 \rangle})} \right) \right|
$$

$$
\leq 2 \left| \exp\left( -\frac{(\mu_{n,i} - \mu_{n,\langle 1 \rangle})^2}{2(\sigma_i^2/T_{n,i} + \sigma_{\langle 1 \rangle}^2/T_{n,\langle 1 \rangle})} \right) - \exp\left( -\frac{(\mu_{n,i'} - \mu_{n,\langle 1 \rangle})^2}{2(\sigma_{i'}^2/T_{n,i'} + \sigma_{\langle 1 \rangle}^2/T_{n,\langle 1 \rangle})} \right) \right|
$$

$$
= 2 \left| \exp\left( -n\frac{(\mu_{n,i} - \mu_{n,\langle 1 \rangle})^2}{2(\sigma_i^2/w_i + \sigma_{\langle 1 \rangle}^2/w_{\langle 1 \rangle})} \right) - \exp\left( -n\frac{(\mu_{n,i'} - \mu_{n,\langle 1 \rangle})^2}{2(\sigma_{i'}^2/w_{i'} + \sigma_{\langle 1 \rangle}^2/w_{\langle 1 \rangle})} \right) \right|,
$$

where $w_i = T_{n,i}/n$ is the sampling rate of arm $i$. For any $\delta_5 = \delta_1^2 > 0$, we have $|\text{iKG}_{n,i} - \text{iKG}_{n,i'}| < \delta_4$ if and only if

$$
\left| \frac{(\mu_i - \mu_{\langle 1 \rangle})^2}{2(\sigma_i^2/w_i + \sigma_{\langle 1 \rangle}^2/w_{\langle 1 \rangle})} - \frac{(\mu_{i'} - \mu_{\langle 1 \rangle})^2}{2(\sigma_{i'}^2/w_{i'} + \sigma_{\langle 1 \rangle}^2/w_{\langle 1 \rangle})} \right| < \delta_5 \tag{D.2}
$$

by the continuity of the function $\exp(\cdot)$ and $|\mu_{n,i} - \mu_i| < \delta_1$. For arms $i \neq \langle 1 \rangle$,

$$
\left| \text{iKG}_{n,i} - \text{iKG}_{n,\langle 1 \rangle} \right|
$$

$$
= \left| \exp\left( -\frac{(\mu_{n,i} - \mu_{n,\langle 1 \rangle})^2}{2(\sigma_i^2/T_{n,i} + \sigma_{\langle 1 \rangle}^2/T_{n,\langle 1 \rangle})} \right) - \sum_{i' \neq 1} \exp\left( -\frac{(\mu_{n,i'} - \mu_{n,\langle 1 \rangle})^2}{2(\sigma_{i'}^2/T_{n,i'} + \sigma_{\langle 1 \rangle}^2/T_{n,\langle 1 \rangle})} \right) \right.
$$

$$
\left. + \sum_{i' \neq \langle 1 \rangle} \exp\left( -\frac{(\mu_{n,i'} - \mu_{n,\langle 1 \rangle})^2}{2(\sigma_{i'}^2/T_{n,i'} + (T_{n,\langle 1 \rangle}+2)\sigma_{\langle 1 \rangle}^2/(T_{n,\langle 1 \rangle}+1)^2)} \right) - \exp\left( -\frac{(\mu_{n,i} - \mu_{n,\langle 1 \rangle})^2}{2((T_{n,i}+2)\sigma_i^2/(T_{n,i}+1)^2 + \sigma_{\langle 1 \rangle}^2/T_{n,\langle 1 \rangle})} \right) \right|.
$$

Notice that $(T_{n,i}+2)\sigma_i^2/(T_{n,i}+1)^2 = \sigma_i^2/(T_{n,i}+1/(T_{n,i}+2))$. When $n$ is large enough, $1/(T_{n,i}+2)$ is sufficiently small according to the consistency of the algorithm. Then

$$
\lim_{n \to \infty} \exp\left( -\frac{(\mu_{n,i} - \mu_{n,\langle 1 \rangle})^2}{2(\sigma_i^2/T_{n,i} + \sigma_{\langle 1 \rangle}^2/T_{n,\langle 1 \rangle})} \right) - \exp\left( -\frac{(\mu_{n,i} - \mu_{n,\langle 1 \rangle})^2}{2((T_{n,i}+2)\sigma_i^2/(T_{n,i}+1)^2 + \sigma_{\langle 1 \rangle}^2/T_{n,\langle 1 \rangle})} \right)
$$

$$
= \frac{\partial\left( \exp\left( -\frac{(\mu_{n,i}-\mu_{n,\langle 1 \rangle})^2}{2(\sigma_i^2/T_{n,i}+\sigma_{\langle 1 \rangle}^2/T_{n,\langle 1 \rangle})} \right) \right)}{\partial T_{n,i}} = \frac{\partial\left( \exp\left( -n\frac{(\mu_{n,i}-\mu_{n,\langle 1 \rangle})^2}{2(\sigma_i^2/w_i+\sigma_{\langle 1 \rangle}^2/w_{\langle 1 \rangle})} \right) \right)}{\partial w_i}.
$$

Since $|\text{iKG}_{n,i} - \text{iKG}_{n,\langle 1 \rangle}| < \delta_4$ for $i \neq \langle 1 \rangle$, given $\delta_6 > 0$, we have

$$
1 - \delta_6 < \left| \sum_{i \neq \langle 1 \rangle} \frac{\partial\left( \exp\left( -n\frac{(\mu_{n,i}-\mu_{n,\langle 1 \rangle})^2}{2(\sigma_i^2/w_i+\sigma_{\langle 1 \rangle}^2/w_{\langle 1 \rangle})} \right) \right)}{\partial w_{\langle 1 \rangle}} \middle/ \frac{\partial\left( \exp\left( -n\frac{(\mu_{n,i}-\mu_{n,\langle 1 \rangle})^2}{2(\sigma_i^2/w_i+\sigma_{\langle 1 \rangle}^2/w_{\langle 1 \rangle})} \right) \right)}{\partial w_i} \right| < 1 + \delta_6.
$$

By (D.2), we have

$$
\frac{\partial\left( \exp\left( -n\frac{(\mu_{n,i}-\mu_{n,\langle 1 \rangle})^2}{2(\sigma_i^2/w_i+\sigma_{\langle 1 \rangle}^2/w_{\langle 1 \rangle})} \right) \right)}{\partial w_i} = \frac{\partial\left( \exp\left( -n\frac{(\mu_{n,i'}-\mu_{n,\langle 1 \rangle})^2}{2(\sigma_{i'}^2/w_{i'}+\sigma_{\langle 1 \rangle}^2/w_{\langle 1 \rangle})} \right) \right)}{\partial w_{i'}}.
$$

Then

$$
1 - \delta_6 < \sum_{i \neq \langle 1 \rangle} \left| \frac{\partial\left( \exp\left( -n\frac{(\mu_{n,i}-\mu_{n,\langle 1 \rangle})^2}{2(\sigma_i^2/w_i+\sigma_{\langle 1 \rangle}^2/w_{\langle 1 \rangle})} \right) \right)}{\partial w_{\langle 1 \rangle}} \middle/ \frac{\partial\left( \exp\left( -n\frac{(\mu_{n,i}-\mu_{n,\langle 1 \rangle})^2}{2(\sigma_i^2/w_i+\sigma_{\langle 1 \rangle}^2/w_{\langle 1 \rangle})} \right) \right)}{\partial w_i} \right| < 1 + \delta_6.
$$

Hence

$$\left| \frac{w_{\langle 1 \rangle}^2}{\sigma_{\langle 1 \rangle}^2} - \sum_{i \neq \langle 1 \rangle} \frac{w_i^2}{\sigma_i^2} \right| < \delta_6.$$

Since $\delta_6$ can be arbitrarily small, $\frac{w_{\langle 1 \rangle}^2}{\sigma_{\langle 1 \rangle}^2} \to \sum_{i \neq \langle 1 \rangle} \frac{w_i^2}{\sigma_i^2}$.

We have shown that

$$1 - \mathbb{P}\{I_n^* = \langle 1 \rangle\} \doteq \exp\left( -n \min_{i \neq \langle 1 \rangle} \frac{(\mu_{n,i} - \mu_{n,\langle 1 \rangle})^2}{2(\sigma_i^2 n/T_{n,i} + \sigma_{\langle 1 \rangle}^2 n/T_{n,\langle 1 \rangle})} \right).$$

Then

$$\Gamma^{\text{iKG}} = \lim_{n \to \infty} -\frac{1}{n} \log(1 - \mathbb{P}\{I_n^* = \langle 1 \rangle\}) = \min_{i \neq \langle 1 \rangle} \frac{(\mu_i - \mu_{\langle 1 \rangle})^2}{2(\sigma_i^2/w_i + \sigma_{\langle 1 \rangle}^2/w_{\langle 1 \rangle})}. \qquad \text{(D.3)}$$

By (D.2),

$$\Gamma^{\text{iKG}} = \frac{(\mu_i - \mu_{\langle 1 \rangle})^2}{2(\sigma_i^2/w_i + \sigma_{\langle 1 \rangle}^2/w_{\langle 1 \rangle})}, \quad \forall i \neq \langle 1 \rangle, \qquad \text{(D.4)}$$

where $w_i$ in (D.3) and (D.4) is the solution of (8) in the main text.

Next, we will show that for any BAI algorithms, $\lim_{n \to \infty} -\frac{1}{n} \log(1 - \mathbb{P}\{I_n^* = \langle 1 \rangle\}) \leq \Gamma^{\text{iKG}}$. Let $W \triangleq \{\boldsymbol{w} = (w_1, \ldots, w_k) : \sum_{i=1}^k w_i = 1 \text{ and } w_i \geq 0, \forall i \in \mathbb{A}\}$ be set of the feasible sampling rates of the $k$ arms. The proof of this claim is divided into two stages. First, suppose that $w_{\langle 1 \rangle} = \alpha$ is fixed for some $0 < \alpha < 1$. We will show that $\max_{\boldsymbol{w} \in W, w_{\langle 1 \rangle} = \alpha} \min_{i \neq \langle 1 \rangle} \frac{(\mu_i - \mu_{\langle 1 \rangle})^2}{2(\sigma_i^2/w_i + \sigma_{\langle 1 \rangle}^2/\alpha)}$ is achieved when

$$\sum_{i \neq \langle 1 \rangle} w_i = 1 - \alpha, \quad \text{and} \quad \frac{(\mu_i - \mu_{\langle 1 \rangle})^2}{2(\sigma_i^2/w_i + \sigma_{\langle 1 \rangle}^2/\alpha)} = \frac{(\mu_{i'} - \mu_{\langle 1 \rangle})^2}{2(\sigma_{i'}^2/w_{i'} + \sigma_{\langle 1 \rangle}^2/\alpha)}, \quad i \neq i' \neq \langle 1 \rangle. \qquad \text{(D.5)}$$

In other words, in this stage, we will prove the first and third equations in (8) of the main text. We prove it by contradiction. Suppose there exists a policy with sampling rates $\boldsymbol{w}' = (w_1', w_2', \ldots, w_k')$ of the $k$ arms such that $\min_{i \neq \langle 1 \rangle} \frac{(\mu_i - \mu_{\langle 1 \rangle})^2}{2(\sigma_i^2/w_i' + \sigma_{\langle 1 \rangle}^2/\alpha)} = \max_{\boldsymbol{w} \in W, w_{\langle 1 \rangle} = \alpha} \min_{i \neq \langle 1 \rangle} \frac{(\mu_i - \mu_{\langle 1 \rangle})^2}{2(\sigma_i^2/w_i + \sigma_{\langle 1 \rangle}^2/\alpha)}$. Since the solution of (D.5) is unique, there exists an arm $i'$ satisfying $\frac{(\mu_{i'} - \mu_{\langle 1 \rangle})^2}{2(\sigma_i^2/w_{i'}' + \sigma_{\langle 1 \rangle}^2/\alpha)} > \min_{i \neq \langle 1 \rangle} \frac{(\mu_i - \mu_{\langle 1 \rangle})^2}{2(\sigma_i^2/w_i' + \sigma_{\langle 1 \rangle}^2/\alpha)}$. We consider a new policy. There exists $\delta_7 > 0$ such that $\tilde{w}_{i'} = w_{i'}' - \delta_7 \in (0, 1)$ and $\tilde{w}_i = w_i' + \delta_7/(k-2) \in (0, 1)$ for $i \neq i' \neq \langle 1 \rangle$. Then

$$\min_{i \neq \langle 1 \rangle} \frac{(\mu_i - \mu_{\langle 1 \rangle})^2}{2(\sigma_i^2/\tilde{w}_i + \sigma_{\langle 1 \rangle}^2/\alpha)} > \min_{i \neq \langle 1 \rangle} \frac{(\mu_i - \mu_{\langle 1 \rangle})^2}{2(\sigma_i^2/w_i' + \sigma_{\langle 1 \rangle}^2/\alpha)} = \max_{\boldsymbol{w} \in W, w_{\langle 1 \rangle} = \alpha} \min_{i \neq \langle 1 \rangle} \frac{(\mu_i - \mu_{\langle 1 \rangle})^2}{2(\sigma_i^2/w_i + \sigma_{\langle 1 \rangle}^2/\alpha)},$$

which yields a contradiction. Therefore, the first and third equations in (8) of the main text hold.

In the second stage, we will prove the second equation in (8) of the main text. Consider the following optimization problem

$$\max_{\alpha \in (0,1)} \quad z$$

$$\text{s.t.} \quad \frac{(\mu_i - \mu_{\langle 1 \rangle})^2}{2(\sigma_i^2/w_i + \sigma_{\langle 1 \rangle}^2/\alpha)} = \frac{(\mu_{i'} - \mu_{\langle 1 \rangle})^2}{2(\sigma_{i'}^2/w_{i'} + \sigma_{\langle 1 \rangle}^2/\alpha)} \quad i, i' \neq \langle 1 \rangle \text{ and } i \neq i',$$

$$\frac{(\mu_i - \mu_{\langle 1 \rangle})^2}{2(\sigma_i^2/w_i + \sigma_{\langle 1 \rangle}^2/\alpha)} \geq z, \quad i \neq \langle 1 \rangle, \qquad \text{(D.6)}$$

$$\sum_{i \neq \langle 1 \rangle} w_i = 1 - \alpha.$$

The Lagrangian function of (D.6) is

$$L(\alpha, \lambda_i) = z + \sum_{i \neq \langle 1 \rangle} \lambda_i \left( \frac{(\mu_i - \mu_{\langle 1 \rangle})^2}{2(\sigma_i^2/w_i + \sigma_{\langle 1 \rangle}^2/\alpha)} - z \right) + \lambda_1 \left( \sum_{i \neq \langle 1 \rangle} w_i - 1 + \alpha \right),$$

where $\lambda_i$'s are the Lagrange multipliers. By the KKT conditions, we have $\lambda_i \partial(\frac{(\mu_i - \mu_{\langle 1 \rangle})^2}{2(\sigma_i^2/w_i + \sigma_{\langle 1 \rangle}^2/\alpha)})/\partial w_i + \lambda_1 = 0$ for all $i \neq \langle 1 \rangle$ and $\sum_{i \neq \langle 1 \rangle} \lambda_i \partial(\frac{(\mu_i - \mu_{\langle 1 \rangle})^2}{2(\sigma_i^2/w_i + \sigma_{\langle 1 \rangle}^2/\alpha)})/\partial w_{\langle 1 \rangle} + \lambda_1 = 0$. Then

$$\sum_{i \neq \langle 1 \rangle} \frac{\partial(\frac{(\mu_i - \mu_{\langle 1 \rangle})^2}{2(\sigma_i^2/w_i + \sigma_{\langle 1 \rangle}^2/\alpha)})/\partial w_{\langle 1 \rangle}}{\partial(\frac{(\mu_i - \mu_{\langle 1 \rangle})^2}{2(\sigma_i^2/w_i + \sigma_{\langle 1 \rangle}^2/\alpha)})/\partial w_i} = 1,$$

i.e., $\frac{w_{\langle 1 \rangle}^2}{\sigma_{\langle 1 \rangle}^2} = \sum_{i \neq \langle 1 \rangle} \frac{w_i^2}{\sigma_i^2}$.

*Remark*: The conditions in (8) of the main text coincide with the optimality conditions developed in [3] using the OCBA method under normal sampling distributions.

## E   Proof of Theorem 2

Our proof of Theorem 2 will be divided into the analysis of the consistency, sampling rates and asymptotic optimality of the iKG-$\epsilon$ algorithm.

We first show consistency, i.e., each arm will be pulled infinitely by the algorithm as the round $n$ goes to infinity. Since

$$\text{iKG}_{t,i}^\epsilon = \begin{cases} \exp\left(-\frac{(\mu_{t,i} - \mu_{t,I_t^*} + \epsilon)^2}{2(\sigma_i^2/T_{t,i} + \sigma_{I_t^*}^2/T_{t,I_t^*})}\right) - \exp\left(-\frac{(\mu_{t,i} - \mu_{t,I_t^*} + \epsilon)^2}{2((T_{t,i} + 2)\sigma_i^2/(T_{t,i} + 1)^2 + \sigma_{I_t^*}^2/T_{t,I_t^*})}\right), & \text{if } i \neq I_t^*, \\ \sum_{i' \neq I_t^*} \exp\left(-\frac{(\mu_{t,i'} - \mu_{t,I_t^*} + \epsilon)^2}{2(\sigma_{i'}^2/T_{t,i'} + \sigma_{I_t^*}^2/T_{t,I_t^*})}\right) - \sum_{i' \neq I_t^*} \exp\left(-\frac{(\mu_{t,i'} - \mu_{t,I_t^*} + \epsilon)^2}{2(\sigma_{i'}^2/T_{t,i'} + (T_{t,I_t^*} + 2)\sigma_{I_t^*}^2/(T_{t,I_t^*} + 1)^2)}\right), & \text{if } i = I_t^*, \end{cases}$$
(E.1)

it is obvious that $\text{iKG}_{t,i}^\epsilon > 0$ for $t > 0$. To prove the consistency, it suffices to prove that $V = \emptyset$, and then the claim is straightforward based on the Strong Law of Large Numbers. For any $\delta_8 > 0$ and arm $i \notin V$, there exists $N_3$ such that when $n > N_3$, $|\mu_{n,i} - \mu_i| < \delta_8$, because arms not in $V$ will be infinitely pulled. Since the $\exp(\cdot)$ is a continuous function and $\sigma_i^2/T_{t,i} - \sigma_i^2(T_{t,i} + 2)/(T_{t,i} + 1)^2 = \sigma_i^2/((T_{t,i} + 1)^2 T_{t,i}) \to 0$ holds for arm $i \notin V$, then for any $\delta_9 > 0$, there exists $N_4$ such that when $n > N_4$, $\text{iKG}_{t,i}^\epsilon < \delta_9$.

Arms $i' \in V$ are pulled for only a finite number of rounds. Then $\max_{i' \in V} T_{t,i'}$ exists and we have $\sigma_{i'}^2/((T_{t,i'} + 1)^2 T_{t,i'}) > \min_{i' \neq I_t^*} \sigma_{i'}^2/\max_{i' \in V}(T_{t,i'} + 2)/(T_{t,i'} + 1)^2$. According to the continuity of the function $\exp(\cdot)$, there exists $\delta_{10} > 0$ such that $\text{iKG}_{t,i'}^\epsilon > \delta_{10}$. Since $\delta_9$ is arbitrary, let $\delta_9 < \delta_{10}$, and then $\text{iKG}_{t,i'}^\epsilon > \text{iKG}_{t,i}^\epsilon$ holds, which implies $I_t \in V$. As the total number of rounds tend to infinity, $V$ will become an empty set eventually. In other words, all the arms will be pulled infinitely and $I_n^* = I^* = \langle 1 \rangle$ holds with probability 1.

We next analyze the sampling rate each arm by the iKG-$\epsilon$ algorithm. Let $\delta_{11} = 2\delta_9 > 0$, we know that when $n$ is large, $\text{iKG}_{n,i}^\epsilon < \delta_9 = \delta_{11}/2$ holds for $i \in \mathbb{A}$. Then $|\text{iKG}_{n,i}^\epsilon - \text{iKG}_{n,i'}^\epsilon| < \text{iKG}_{n,i}^\epsilon + \text{iKG}_{n,i'}^\epsilon < \delta_{11}/2 + \delta_{11}/2 = \delta_{11}$, where $i \neq i'$. For any $i, i' \in \mathbb{A}$ and $i \neq i' \neq \langle 1 \rangle$,

$$|\text{iKG}_{n,i}^\epsilon - \text{iKG}_{n,i'}^\epsilon|$$
$$= \left| \exp\left(-\frac{(\mu_{n,i} - \mu_{n,\langle 1 \rangle} + \epsilon)^2}{2(\sigma_i^2/T_{n,i} + \sigma_{\langle 1 \rangle}^2/T_{n,\langle 1 \rangle})}\right) - \exp\left(-\frac{(\mu_{n,i'} - \mu_{n,\langle 1 \rangle} + \epsilon)^2}{2(\sigma_{i'}^2/T_{n,i'} + \sigma_{\langle 1 \rangle}^2/T_{n,\langle 1 \rangle})}\right) \right.$$
$$\left. + \exp\left(-\frac{(\mu_{n,i'} - \mu_{n,\langle 1 \rangle} + \epsilon)^2}{2((T_{n,i'} + 2)\sigma_{i'}^2/(T_{n,i'} + 1)^2 + \sigma_{\langle 1 \rangle}^2/T_{n,\langle 1 \rangle})}\right) - \exp\left(-\frac{(\mu_{n,i} - \mu_{n,\langle 1 \rangle} + \epsilon)^2}{2((T_{n,i} + 2)\sigma_i^2/(T_{n,i} + 1)^2 + \sigma_{\langle 1 \rangle}^2/T_{n,\langle 1 \rangle})}\right) \right|$$
$$\leq 2 \left| \exp\left(-\frac{(\mu_{n,i} - \mu_{n,\langle 1 \rangle} + \epsilon)^2}{2(\sigma_i^2/T_{n,i} + \sigma_{\langle 1 \rangle}^2/T_{n,\langle 1 \rangle})}\right) - \exp\left(-\frac{(\mu_{n,i'} - \mu_{n,\langle 1 \rangle} + \epsilon)^2}{2(\sigma_{i'}^2/T_{n,i'} + \sigma_{\langle 1 \rangle}^2/T_{n,\langle 1 \rangle})}\right) \right|$$
$$= 2 \left| \exp\left(-n\frac{(\mu_{n,i} - \mu_{n,\langle 1 \rangle} + \epsilon)^2}{2(\sigma_i^2/w_i + \sigma_{\langle 1 \rangle}^2/w_{\langle 1 \rangle})}\right) - \exp\left(-n\frac{(\mu_{n,i'} - \mu_{n,\langle 1 \rangle} + \epsilon)^2}{2(\sigma_{i'}^2/w_{i'} + \sigma_{\langle 1 \rangle}^2/w_{\langle 1 \rangle})}\right) \right|,$$

where $w_i = T_{n,i}/n$ is the sampling rate of arm $i$. For any $\delta_{12} = \delta_8^2 > 0$, we have $|\text{iKG}_{n,i}^\epsilon - \text{iKG}_{n,i'}^\epsilon| < \delta_{11}$ if and only if

$$\left| \frac{(\mu_i - \mu_{\langle 1 \rangle} + \epsilon)^2}{2(\sigma_i^2/w_i + \sigma_{\langle 1 \rangle}^2/w_{\langle 1 \rangle})} - \frac{(\mu_{i'} - \mu_{\langle 1 \rangle} + \epsilon)^2}{2(\sigma_{i'}^2/w_{i'} + \sigma_{\langle 1 \rangle}^2/w_{\langle 1 \rangle})} \right| < \delta_{12} \tag{E.2}$$

by the continuity of the function $\exp(\cdot)$ and $|\mu_{n,i} - \mu_i| < \delta_8$. For arms $i \neq \langle 1 \rangle$,

$$\begin{aligned}
&\left| \text{iKG}_{n,i}^\epsilon - \text{iKG}_{n,\langle 1 \rangle}^\epsilon \right| \\
&= \left| \exp\left( -\frac{(\mu_{n,i} - \mu_{n,\langle 1 \rangle} + \epsilon)^2}{2(\sigma_i^2/T_{n,i} + \sigma_{\langle 1 \rangle}^2/T_{n,\langle 1 \rangle})} \right) - \sum_{i' \neq \langle 1 \rangle} \exp\left( -\frac{(\mu_{n,i'} - \mu_{n,\langle 1 \rangle} + \epsilon)^2}{2(\sigma_{i'}^2/T_{n,i'} + \sigma_{\langle 1 \rangle}^2/T_{n,\langle 1 \rangle})} \right) \right. \\
&\quad \left. + \sum_{i' \neq \langle 1 \rangle} \exp\left( -\frac{(\mu_{n,i'} - \mu_{n,\langle 1 \rangle} + \epsilon)^2}{2(\sigma_{i'}^2/T_{n,i'} + (T_{n,\langle 1 \rangle} + 2)\sigma_{\langle 1 \rangle}^2/(T_{n,\langle 1 \rangle} + 1)^2)} \right) - \exp\left( -\frac{(\mu_{n,i} - \mu_{n,\langle 1 \rangle} + \epsilon)^2}{2((T_{n,i} + 2)\sigma_i^2/(T_{n,i} + 1)^2 + \sigma_{\langle 1 \rangle}^2/T_{n,\langle 1 \rangle})} \right) \right|.
\end{aligned}$$

Notice that $(T_{n,i} + 2)\sigma_i^2/(T_{n,i} + 1)^2 = \sigma_i^2/(T_{n,i} + 1/(T_{n,i} + 2))$. When $n$ is large enough, $1/(T_{n,i} + 2)$ is sufficiently small according to the consistency of the algorithm. Then

$$\begin{aligned}
&\lim_{n \to \infty} \exp\left( -\frac{(\mu_{n,i} - \mu_{n,\langle 1 \rangle} + \epsilon)^2}{2(\sigma_i^2/T_{n,i} + \sigma_{\langle 1 \rangle}^2/T_{n,\langle 1 \rangle})} \right) - \exp\left( -\frac{(\mu_{n,i} - \mu_{n,\langle 1 \rangle} + \epsilon)^2}{2((T_{n,i} + 2)\sigma_i^2/(T_{n,i} + 1)^2 + \sigma_{\langle 1 \rangle}^2/T_{n,\langle 1 \rangle})} \right) \\
&= \frac{\partial \left( \exp\left( -\frac{(\mu_{n,i} - \mu_{n,\langle 1 \rangle} + \epsilon)^2}{2(\sigma_i^2/T_{n,i} + \sigma_{\langle 1 \rangle}^2/T_{n,\langle 1 \rangle})} \right) \right)}{\partial T_{n,i}} = \frac{\partial \left( \exp\left( -n\frac{(\mu_{n,i} - \mu_{n,\langle 1 \rangle} + \epsilon)^2}{2(\sigma_i^2/w_i + \sigma_{\langle 1 \rangle}^2/w_{\langle 1 \rangle})} \right) \right)}{\partial w_i}.
\end{aligned}$$

Since $|\text{iKG}_{n,i}^\epsilon - \text{iKG}_{n,\langle 1 \rangle}^\epsilon| < \delta_{11}$ for $i \neq \langle 1 \rangle$, given $\delta_{13} > 0$, we have

$$1 - \delta_{13} < \left| \sum_{i \neq \langle 1 \rangle} \frac{\partial \left( \exp\left( -n\frac{(\mu_{n,i} - \mu_{n,\langle 1 \rangle} + \epsilon)^2}{2(\sigma_i^2/w_i + \sigma_{\langle 1 \rangle}^2/w_{\langle 1 \rangle})} \right) \right)}{\partial w_{\langle 1 \rangle}} \Bigg/ \frac{\partial \left( \exp\left( -n\frac{(\mu_{n,i} - \mu_{n,\langle 1 \rangle} + \epsilon)^2}{2(\sigma_i^2/w_i + \sigma_{\langle 1 \rangle}^2/w_{\langle 1 \rangle})} \right) \right)}{\partial w_i} \right| < 1 + \delta_{13}.$$

By (E.2), we have

$$\frac{\partial \left( \exp\left( -n\frac{(\mu_{n,i} - \mu_{n,\langle 1 \rangle} + \epsilon)^2}{2(\sigma_i^2/w_i + \sigma_{\langle 1 \rangle}^2/w_{\langle 1 \rangle})} \right) \right)}{\partial w_i} = \frac{\partial \left( \exp\left( -n\frac{(\mu_{n,i'} - \mu_{n,\langle 1 \rangle} + \epsilon)^2}{2(\sigma_{i'}^2/w_{i'} + \sigma_{\langle 1 \rangle}^2/w_{\langle 1 \rangle})} \right) \right)}{\partial w_{i'}}.$$

Then

$$1 - \delta_{13} < \sum_{i \neq \langle 1 \rangle} \left| \frac{\partial \left( \exp\left( -n\frac{(\mu_{n,i} - \mu_{n,\langle 1 \rangle} + \epsilon)^2}{2(\sigma_i^2/w_i + \sigma_{\langle 1 \rangle}^2/w_{\langle 1 \rangle})} \right) \right)}{\partial w_{\langle 1 \rangle}} \Bigg/ \frac{\partial \left( \exp\left( -n\frac{(\mu_{n,i} - \mu_{n,\langle 1 \rangle} + \epsilon)^2}{2(\sigma_i^2/w_i + \sigma_{\langle 1 \rangle}^2/w_{\langle 1 \rangle})} \right) \right)}{\partial w_i} \right| < 1 + \delta_{13}.$$

Hence

$$\left| \frac{w_{\langle 1 \rangle}^2}{\sigma_{\langle 1 \rangle}^2} - \sum_{i \neq \langle 1 \rangle} \frac{w_i^2}{\sigma_i^2} \right| < \delta_{13}.$$

Since $\delta_{13}$ can be arbitrarily small, $\frac{w_{\langle 1 \rangle}^2}{\sigma_{\langle 1 \rangle}^2} \to \sum_{i \neq \langle 1 \rangle} \frac{w_i^2}{\sigma_i^2}$.

We know that

$$1 - \mathbb{P}\{G_n^\epsilon = G^\epsilon\} = \mathbb{P}\left\{ \bigcup_{i \in G_n^\epsilon} (\theta_i < \theta_{\langle 1 \rangle} - \epsilon) \cup \bigcup_{i \in \mathbb{A} \setminus G_n^\epsilon} (\theta_i > \theta_{\langle 1 \rangle} - \epsilon) \right\},$$

and

$$\max(\max_{i \in G_n^\epsilon} \mathbb{P}(\theta_i < \theta_{\langle 1 \rangle} - \epsilon), \max_{i \in \mathbb{A} \backslash G_n^\epsilon} \mathbb{P}(\theta_i > \theta_{\langle 1 \rangle} - \epsilon))$$

$$\leq \mathbb{P}\left\{ \bigcup_{i \in G_n^\epsilon} (\theta_i < \theta_{\langle 1 \rangle} - \epsilon) \cup \bigcup_{i \in \mathbb{A} \backslash G_n^\epsilon} (\theta_i > \theta_{\langle 1 \rangle} - \epsilon) \right\}$$

$$\leq k \max(\max_{i \in G_n^\epsilon} \mathbb{P}(\theta_i < \theta_{\langle 1 \rangle} - \epsilon), \max_{i \in \mathbb{A} \backslash G_n^\epsilon} \mathbb{P}(\theta_i > \theta_{\langle 1 \rangle} - \epsilon)).$$

Then

$$1 - \mathbb{P}\{G_n^\epsilon = G^\epsilon\} \doteq \exp\left( -\frac{(\mu_{n,i} - \mu_{n,\langle 1 \rangle} + \epsilon)^2}{2(\sigma_{n,i}^2 + \sigma_{n,\langle 1 \rangle}^2)} \right).$$

We have

$$\Gamma^\epsilon = \lim_{n \to \infty} -\frac{1}{n} \log(1 - \mathbb{P}\{G_n^\epsilon = G^\epsilon\}) = \min_{i \neq \langle 1 \rangle} \frac{(\mu_i - \mu_{\langle 1 \rangle} + \epsilon)^2}{2(\sigma_i^2/w_i + \sigma_{\langle 1 \rangle}^2/w_{\langle 1 \rangle})}. \tag{E.3}$$

By (E.2),

$$\Gamma^\epsilon = \frac{(\mu_i - \mu_{\langle 1 \rangle} + \epsilon)^2}{2(\sigma_i^2/w_i + \sigma_{\langle 1 \rangle}^2/w_{\langle 1 \rangle})}, \quad \forall i \neq \langle 1 \rangle, \tag{E.4}$$

where $w_i$ in (E.3) and (E.4) is the solution of (12) in the main text.

Next, we will show that for any $\epsilon$-good arm identification algorithms, $\lim_{n \to \infty} -\frac{1}{n} \log(1 - \mathbb{P}\{G_n^\epsilon = G^\epsilon\}) \leq \Gamma^\epsilon$. Let $W \triangleq \{\boldsymbol{w} = (w_1, \ldots, w_k) : \sum_{i=1}^k w_i = 1 \text{ and } w_i \geq 0, \forall i \in \mathbb{A}\}$ be set of the feasible sampling rates of the $k$ arms. The proof of this claim is divided into two stages. First, suppose that $w_{\langle 1 \rangle} = \alpha$ is fixed for some $0 < \alpha < 1$. We will show that $\max_{\boldsymbol{w} \in W, w_{\langle 1 \rangle} = \alpha} \min_{i \neq \langle 1 \rangle} \frac{(\mu_i - \mu_{\langle 1 \rangle} + \epsilon)^2}{2(\sigma_i^2/w_i + \sigma_{\langle 1 \rangle}^2/\alpha)}$ is achieved when

$$\sum_{i \neq \langle 1 \rangle} w_i = 1 - \alpha, \quad \text{and} \quad \frac{(\mu_i - \mu_{\langle 1 \rangle} + \epsilon)^2}{2(\sigma_i^2/w_i + \sigma_{\langle 1 \rangle}^2/\alpha)} = \frac{(\mu_{i'} - \mu_{\langle 1 \rangle} + \epsilon)^2}{2(\sigma_{i'}^2/w_{i'} + \sigma_{\langle 1 \rangle}^2/\alpha)}, \quad i \neq i' \neq \langle 1 \rangle. \tag{E.5}$$

In other words, in this stage, we will prove the first and third equations in (12) of the main text. We prove it by contradiction. Suppose there exists a policy with sampling rates $\boldsymbol{w}' = (w_1', w_2', \ldots, w_k')$ of the $k$ arms such that $\min_{i \neq \langle 1 \rangle} \frac{(\mu_i - \mu_{\langle 1 \rangle} + \epsilon)^2}{2(\sigma_i^2/w_i' + \sigma_{\langle 1 \rangle}^2/\alpha)} = \max_{\boldsymbol{w} \in W, w_{\langle 1 \rangle} = \alpha} \min_{i \neq \langle 1 \rangle} \frac{(\mu_i - \mu_{\langle 1 \rangle} + \epsilon)^2}{2(\sigma_i^2/w_i + \sigma_{\langle 1 \rangle}^2/\alpha)}$. Since the solution of (E.5) is unique, there exists an arm $i'$ satisfying $\frac{(\mu_{i'} - \mu_{\langle 1 \rangle} + \epsilon)^2}{2(\sigma_{i'}^2/w_{i'}' + \sigma_{\langle 1 \rangle}^2/\alpha)} > \min_{i \neq \langle 1 \rangle} \frac{(\mu_i - \mu_{\langle 1 \rangle} + \epsilon)^2}{2(\sigma_i^2/w_i' + \sigma_{\langle 1 \rangle}^2/\alpha)}$. We consider a new policy. There exists $\delta_{14} > 0$ such that $\tilde{w}_{i'} = w_{i'}' - \delta_{14} \in (0, 1)$ and $\tilde{w}_i = w_i' + \delta_{14}/(k-2) \in (0, 1)$ for $i \neq i' \neq \langle 1 \rangle$. Then

$$\min_{i \neq \langle 1 \rangle} \frac{(\mu_i - \mu_{\langle 1 \rangle} + \epsilon)^2}{2(\sigma_i^2/\tilde{w}_i + \sigma_{\langle 1 \rangle}^2/\alpha)} > \min_{i \neq \langle 1 \rangle} \frac{(\mu_i - \mu_{\langle 1 \rangle} + \epsilon)^2}{2(\sigma_i^2/w_i' + \sigma_{\langle 1 \rangle}^2/\alpha)} = \max_{\boldsymbol{w} \in W, w_{\langle 1 \rangle} = \alpha} \min_{i \neq \langle 1 \rangle} \frac{(\mu_i - \mu_{\langle 1 \rangle} + \epsilon)^2}{2(\sigma_i^2/w_i + \sigma_{\langle 1 \rangle}^2/\alpha)},$$

which yields a contradiction. Therefore, the first and third equations in (12) of the main text hold.

In the second stage, we will prove the second equation in (12) of the main text. Consider the following optimization problem

$$\begin{aligned}
\max_{\alpha \in (0,1)} \quad & z \\
\text{s.t.} \quad & \frac{(\mu_i - \mu_{\langle 1 \rangle} + \epsilon)^2}{2(\sigma_i^2/w_i + \sigma_{\langle 1 \rangle}^2/\alpha)} = \frac{(\mu_{i'} - \mu_{\langle 1 \rangle} + \epsilon)^2}{2(\sigma_{i'}^2/w_{i'} + \sigma_{\langle 1 \rangle}^2/\alpha)}, \quad i, i' \neq \langle 1 \rangle \text{ and } i \neq i', \\
& \frac{(\mu_i - \mu_{\langle 1 \rangle} + \epsilon)^2}{2(\sigma_i^2/w_i + \sigma_{\langle 1 \rangle}^2/\alpha)} \geq z, \quad i \neq \langle 1 \rangle, \\
& \sum_{i \neq \langle 1 \rangle} w_i = 1 - \alpha.
\end{aligned} \tag{E.6}$$

The Lagrangian function of (E.6) is

$$L(\alpha, \lambda_i) = z + \sum_{i \neq \langle 1 \rangle} \lambda_i \left( \frac{(\mu_i - \mu_{\langle 1 \rangle} + \epsilon)^2}{2(\sigma_i^2/w_i + \sigma_{\langle 1 \rangle}^2/\alpha)} - z \right) + \lambda_1 \left( \sum_{i \neq \langle 1 \rangle} w_i - 1 + \alpha \right),$$

where $\lambda_i$'s are the Lagrange multipliers. By the KKT conditions, we have $\lambda_i \partial \left( \frac{(\mu_i - \mu_{\langle 1 \rangle} + \epsilon)^2}{2(\sigma_i^2/w_i + \sigma_{\langle 1 \rangle}^2/\alpha)} \right)/\partial w_i + \lambda_1 = 0$ for all $i \neq \langle 1 \rangle$ and $\sum_{i \neq \langle 1 \rangle} \lambda_i \partial \left( \frac{(\mu_i - \mu_{\langle 1 \rangle} + \epsilon)^2}{2(\sigma_i^2/w_i + \sigma_{\langle 1 \rangle}^2/\alpha)} \right)/\partial w_{\langle 1 \rangle} + \lambda_1 = 0$. Then

$$\sum_{i \neq \langle 1 \rangle} \frac{\partial \left( \frac{(\mu_i - \mu_{\langle 1 \rangle} + \epsilon)^2}{2(\sigma_i^2/w_i + \sigma_{\langle 1 \rangle}^2/\alpha)} \right)/\partial w_{\langle 1 \rangle}}{\partial \left( \frac{(\mu_i - \mu_{\langle 1 \rangle} + \epsilon)^2}{2(\sigma_i^2/w_i + \sigma_{\langle 1 \rangle}^2/\alpha)} \right)/\partial w_i} = 1,$$

i.e., $\frac{w_{\langle 1 \rangle}^2}{\sigma_{\langle 1 \rangle}^2} = \sum_{i \neq \langle 1 \rangle} \frac{w_i^2}{\sigma_i^2}$.

# F  Proof of Theorem 3

Our proof of Theorem 3 will be divided into the analysis of the consistency, sampling rates and asymptotic optimality of the iKG-F algorithm.

We first show consistency, i.e., each arm will be pulled infinitely by the algorithm as the round $n$ goes to infinity. Since

$$\text{iKG}_{t,i}^{F} = \sum_{j=1}^{m} \left( \exp \left( -\frac{(\gamma_j - \mu_{t,ij})^2}{2\sigma_{ij}^2/T_{t,i}} \mathbf{1}\{i \in \mathcal{S}_t^1\} \right) - \exp \left( -\frac{(\gamma_j - \mu_{t,ij})^2}{2(T_{t,i} + 2)\sigma_{ij}^2/(T_{t,i} + 1)^2} \mathbf{1}\{i \in \mathcal{S}_t^1\} \right) \right)$$
$$+ \exp \left( -\sum_{j \in \mathcal{E}_{t,i}^2} \frac{(\gamma_j - \mu_{t,ij})^2}{2\sigma_{ij}^2/T_{t,i}} \mathbf{1}\{i \in \mathcal{S}_t^2\} \right) - \exp \left( -\sum_{j \in \mathcal{E}_{t,i}^2} \frac{(\gamma_j - \mu_{t,ij})^2}{2(T_{t,i} + 2)\sigma_{ij}^2/(T_{t,i} + 1)^2} \mathbf{1}\{i \in \mathcal{S}_t^2\} \right).$$

(F.1)

It is obvious that $\text{iKG}_{t,i}^{F} > 0$ for $t > 0$. To prove the consistency, it suffices to prove that $V = \emptyset$ and then the claim is straightforward based on the Strong Law of Large Numbers. For any $\delta_{15} > 0$ and $i \notin V$, there exists $N_5$ such that when $n > N_5$, $|\mu_{n,i} - \mu_i| < \delta_{15}$, because arms not in $V$ will be infinitely pulled. Since the $\exp(\cdot)$ is a continuous function and $\sigma_i^2/T_{t,i} - \sigma_i^2(T_{t,i} + 2)/(T_{t,i} + 1)^2 = \sigma_i^2/((T_{t,i} + 1)^2 T_{t,i}) \to 0$ holds for arm $i \notin V$, then for any $\delta_{16} > 0$, there exists $N_6$ such that when $n > N_6$, $\text{iKG}_{t,i}^{F} < \delta_{16}$.

Arms $i' \in V$ are pulled for only a finite number of rounds. Then $\max_{i' \in V} T_{t,i'}$ exists and we have $\sigma_{i'}^2/((T_{t,i'} + 1)^2 T_{t,i'}) > \min_{i' \in \mathbb{A}} \sigma_{i'}^2/\max_{i' \in V}(T_{t,i'} + 2)/(T_{t,i'} + 1)^2$. According to the continuity of the function $\exp(\cdot)$, there exists $\delta_{17} > 0$ such that $\text{iKG}_{t,i'}^{F} > \delta_{17}$. Since $\delta_{16}$ is arbitrary, let $\delta_{16} < \delta_{17}$ and then $\text{iKG}_{t,i'}^{F} > \text{iKG}_{t,i}^{F}$ holds, which implies $I_t \in V$. As the total number of rounds tend to infinity, $V$ will become an empty set eventually. In other words, all the arms will be pulled infinitely.

We next analyze the sampling rate each arm by the iKG-F algorithm. Let $\delta_{18} = 2\delta_{16} > 0$, we know that when $n$ is large, $\text{iKG}_{n,i}^{F} < \delta_{16} = \delta_{18}/2$ holds for $i \in \mathbb{A}$. Then

$|\text{iKG}^F_{n,i} - \text{iKG}^F_{n,i'}| < \text{iKG}^F_{n,i} + \text{iKG}^F_{n,i'} < \delta_{18}/2 + \delta_{18}/2 = \delta_{18}$, where $i \neq i'$. For any $i, i' \in \mathbb{A}$,

$$|\text{iKG}^F_{n,i} - \text{iKG}^F_{n,i'}|$$

$$= \left| \sum_{j=1}^m \exp\left( -\frac{(\gamma_j - \mu_{n,ij})^2}{2\sigma_{ij}^2/T_{n,i}} \mathbf{1}\{i \in \mathcal{S}_n^1\} \right) - \sum_{j=1}^m \exp\left( -\frac{(\gamma_j - \mu_{n,i'j})^2}{2\sigma_{i'j}^2/T_{n,i'}} \mathbf{1}\{i' \in \mathcal{S}_n^1\} \right) \right.$$

$$+ \exp\left( -\sum_{j \in \mathcal{E}_{n,i}^2} \frac{(\gamma_j - \mu_{n,ij})^2}{2\sigma_{ij}^2/T_{n,i}} \mathbf{1}\{i \in \mathcal{S}_n^2\} \right) - \exp\left( -\sum_{j \in \mathcal{E}_{n,i'}^2} \frac{(\gamma_j - \mu_{n,i'j})^2}{2\sigma_{i'j}^2/T_{n,i'}} \mathbf{1}\{i' \in \mathcal{S}_n^2\} \right)$$

$$+ \exp\left( -\frac{(\gamma_j - \mu_{n,i'j})^2}{2(T_{n,i'} + 2)\sigma_{i'j}^2/(T_{n,i'} + 1)^2} \mathbf{1}\{i' \in \mathcal{S}_n^1\} \right) - \exp\left( -\frac{(\gamma_j - \mu_{n,ij})^2}{2(T_{n,i} + 2)\sigma_{ij}^2/(T_{n,i} + 1)^2} \mathbf{1}\{i \in \mathcal{S}_n^1\} \right)$$

$$\left. + \exp\left( -\sum_{j \in \mathcal{E}_{n,i'}^2} \frac{(\gamma_j - \mu_{n,i'j})^2}{2(T_{n,i'} + 2)\sigma_{i'j}^2/(T_{n,i'} + 1)^2} \mathbf{1}\{i' \in \mathcal{S}_n^2\} \right) - \exp\left( -\sum_{j \in \mathcal{E}_{n,i}^2} \frac{(\gamma_j - \mu_{n,ij})^2}{2(T_{n,i} + 2)\sigma_{ij}^2/(T_{n,i} + 1)^2} \mathbf{1}\{i \in \mathcal{S}_n^2\} \right) \right|$$

$$\leq 2 \left| \sum_{j=1}^m \exp\left( -\frac{(\gamma_j - \mu_{n,ij})^2}{2\sigma_{ij}^2/T_{n,i}} \mathbf{1}\{i \in \mathcal{S}_n^1\} \right) - \sum_{j=1}^m \exp\left( -\frac{(\gamma_j - \mu_{n,i'j})^2}{2\sigma_{i'j}^2/T_{n,i'}} \mathbf{1}\{i' \in \mathcal{S}_n^1\} \right) \right.$$

$$\left. + \exp\left( -\sum_{j \in \mathcal{E}_{n,i}^2} \frac{(\gamma_j - \mu_{n,ij})^2}{2\sigma_{ij}^2/T_{n,i}} \mathbf{1}\{i \in \mathcal{S}_n^2\} \right) - \exp\left( -\sum_{j \in \mathcal{E}_{n,i'}^2} \frac{(\gamma_j - \mu_{n,i'j})^2}{2\sigma_{i'j}^2/T_{n,i'}} \mathbf{1}\{i' \in \mathcal{S}_n^2\} \right) \right|$$

$$\leq 2 \left| m \max_{j \in \mathcal{E}_{n,i}^1} \exp\left( -\frac{(\gamma_j - \mu_{n,ij})^2}{2\sigma_{ij}^2/T_{n,i}} \mathbf{1}\{i \in \mathcal{S}_n^1\} \right) - m \max_{j \in \mathcal{E}_{n,i}^1} \exp\left( -\frac{(\gamma_j - \mu_{n,i'j})^2}{2\sigma_{i'j}^2/T_{n,i'}} \mathbf{1}\{i' \in \mathcal{S}_n^1\} \right) \right.$$

$$\left. + \exp\left( -\sum_{j \in \mathcal{E}_{n,i}^2} \frac{(\gamma_j - \mu_{n,ij})^2}{2\sigma_{ij}^2/T_{n,i}} \mathbf{1}\{i \in \mathcal{S}_n^2\} \right) - \exp\left( -\sum_{j \in \mathcal{E}_{n,i'}^2} \frac{(\gamma_j - \mu_{n,i'j})^2}{2\sigma_{i'j}^2/T_{n,i'}} \mathbf{1}\{i' \in \mathcal{S}_n^2\} \right) \right|$$

$$= 2 \left| m \exp\left( -w_i \min_{j \in \mathcal{E}_{n,i}^1} \frac{(\gamma_j - \mu_{n,ij})^2}{2\sigma_{ij}^2} \mathbf{1}\{i \in \mathcal{S}_n^1\} \right) - m \exp\left( -w_{i'} \min_{j \in \mathcal{E}_{n,i}^1} \frac{(\gamma_j - \mu_{n,i'j})^2}{2\sigma_{i'j}^2} \mathbf{1}\{i' \in \mathcal{S}_n^1\} \right) \right.$$

$$\left. + \exp\left( -w_i \sum_{j \in \mathcal{E}_{n,i}^2} \frac{(\gamma_j - \mu_{n,ij})^2}{2\sigma_{ij}^2} \mathbf{1}\{i \in \mathcal{S}_n^2\} \right) - \exp\left( -w_{i'} \sum_{j \in \mathcal{E}_{n,i'}^2} \frac{(\gamma_j - \mu_{n,i'j})^2}{2\sigma_{i'j}^2} \mathbf{1}\{i' \in \mathcal{S}_n^2\} \right) \right|,$$

where $w_i = T_{n,i}/n$ is the sampling rate of arm $i$. We have shown that $|\mu_{n,i} - \mu_i| < \delta_{15}$ for any $\delta_{15} > 0$ and $i = 1, 2, \ldots, k$. We can find a sufficiently large positive integer $n'$ such that when $n > n'$, $S_n^1 = S^1$, $S_n^2 = S^2$, $\mathcal{E}_{n,i}^1 = \mathcal{E}_i^1$ and $\mathcal{E}_{n,i}^2 = \mathcal{E}_i^2$, where $i = 1, 2, \ldots, k$ and $j = 1, 2, \ldots, m$. Note that $S_n^1 \cap S_2^2 = \emptyset$ and $S^1 \cap S^2 = \emptyset$. If $i \in S_n^1 = S^1$, $|\text{iKG}^F_{n,i} - \text{iKG}^F_{n,i'}| \leq 2m \left| \exp\left( -w_i \min_{j \in \mathcal{E}_{n,i}^1} \frac{(\gamma_j - \mu_{n,ij})^2}{2\sigma_{ij}^2} \right\} \right) - \exp\left( -w_{i'} \min_{j \in \mathcal{E}_{n,i'}^1} \frac{(\gamma_j - \mu_{n,i'j})^2}{2\sigma_{i'j}^2} \right) \right| \leq 2m \left| \exp\left( -w_i \min_{j \in \mathcal{E}_i^1} \frac{(\gamma_j - \mu_{ij} - \delta_{15})^2}{2\sigma_{ij}^2} \right\} \right) - \exp\left( -w_{i'} \min_{j \in \mathcal{E}_{i'}^1} \frac{(\gamma_j - \mu_{i'j} + \delta_{15})^2}{2\sigma_{i'j}^2} \right) \right|$. We have shown that $|\text{iKG}^F_{n,i} - \text{iKG}^F_{n,i'}| < \delta_{18}$. Hence $\left| w_i \min_{j \in \mathcal{E}_i^1} \frac{(\gamma_j - \mu_{ij})^2}{2\sigma_{ij}^2} - w_{i'} \min_{j \in \mathcal{E}_{i'}^1} \frac{(\gamma_j - \mu_{i'j})^2}{2\sigma_{i'j}^2} \right| < \delta_{19}$ for any $\delta_{19} = \delta_{15}^2 > 0$ by the continuity of the function $\exp(\cdot)$. We can get similar result when $i \in S_n^2 = S^2$. Hence, $|\text{iKG}^F_{n,i} - \text{iKG}^F_{n,i'}| < \delta_{18}$ if and only if

$$\left| w_i \min_{j \in \mathcal{E}_i^1} \frac{(\gamma_j - \mu_{ij})^2}{2\sigma_{ij}^2} \mathbf{1}\{i \in \mathcal{S}^1\} + w_i \sum_{j \in \mathcal{E}_i^2} \frac{(\gamma_j - \mu_{ij})^2}{2\sigma_{ij}^2} \mathbf{1}\{i \in \mathcal{S}^2\} \right.$$

$$\left. - w_{i'} \min_{j \in \mathcal{E}_{i'}^1} \frac{(\gamma_j - \mu_{i'j})^2}{2\sigma_{i'j}^2} \mathbf{1}\{i' \in \mathcal{S}^1\} + w_{i'} \sum_{j \in \mathcal{E}_{i'}^2} \frac{(\gamma_j - \mu_{i'j})^2}{2\sigma_{i'j}^2} \mathbf{1}\{i' \in \mathcal{S}^2\} \right| < \delta_{19}$$

(F.2)

by the continuity of the function $\exp(\cdot)$ and $|\mu_{n,i} - \mu_i| < \delta_{15}$.

We have known that

$$1 - \mathbb{P}\{\mathcal{S}_n^1 = \mathcal{S}^1\} = \mathbb{P}\left\{ \bigcup_{i \in \mathcal{S}_n^1} \left( \bigcup_{j=1}^m (\theta_{ij} > \gamma_j) \right) \cup \bigcup_{i \in \mathcal{S}_n^2} \left( \bigcap_{j=1}^m (\theta_{ij} \leq \gamma_j) \right) \right\}.$$

We have

$$\max \left( \max_{i \in \mathcal{S}_n^1} \mathbb{P}\left( \bigcup_{j=1}^m (\theta_{ij} > \gamma_j) \right), \max_{i \in \mathcal{S}_n^2} \mathbb{P}\left( \bigcap_{j=1}^m (\theta_{ij} \leq \gamma_j) \right) \right)$$

$$\leq \mathbb{P}\left\{ \bigcup_{i \in \mathcal{S}_n^1} \left( \bigcup_{j=1}^m (\theta_{ij} > \gamma_j) \right) \cup \bigcup_{i \in \mathcal{S}_n^2} \left( \bigcap_{j=1}^m (\theta_{ij} \leq \gamma_j) \right) \right\}$$

$$\leq k \max \left( \max_{i \in \mathcal{S}_n^1} \mathbb{P}\left( \bigcup_{j=1}^m (\theta_{ij} > \gamma_j) \right), \max_{i \in \mathcal{S}_n^2} \mathbb{P}\left( \bigcap_{j=1}^m (\theta_{ij} \leq \gamma_j) \right) \right).$$

Then

$$1 - \mathbb{P}\{\mathcal{S}_n^1 = \mathcal{S}^1\} \doteq \max \left( \max_{i \in \mathcal{S}_n^1} \mathbb{P}\left( \bigcup_{j=1}^m (\theta_{ij} > \gamma_j) \right), \max_{i \in \mathcal{S}_n^2} \mathbb{P}\left( \bigcap_{j=1}^m (\theta_{ij} \leq \gamma_j) \right) \right).$$

For arm $i \in \mathcal{S}_n^1$,

$$\max_{j \in \mathcal{E}_{n,i}^1} \mathbb{P}(\theta_{ij} > \gamma_j) \leq \mathbb{P}\left( \bigcup_{j=1}^m (\theta_{ij} > \gamma_j) \right) \leq m \max_{j \in \mathcal{E}_{n,i}^1} \mathbb{P}(\theta_{ij} > \gamma_j).$$

For arm $i \in \mathcal{S}_n^2$,

$$\mathbb{P}\left( \bigcap_{j=1}^m (\theta_{ij} \leq \gamma_j) \right) \to \mathbb{P}\left( \bigcap_{j \in \mathcal{E}_{n,i}^2} (\theta_{ij} \leq \gamma_j) \right),$$

because $\lim_{n \to \infty} \mathbb{P}\left( \bigcap_{j \in \mathcal{E}_{n,i}^1} (\theta_{ij} \leq \gamma_j) \right) \to 1$. Hence

$$1 - \mathbb{P}\{\mathcal{S}_n^1 = \mathcal{S}^1\} \doteq \exp\left( -\min_{i \in \mathcal{S}_n^1} \frac{(\gamma_j - \mu_{n,ij})^2}{2\sigma_{ij}^2/T_{n,i}} \mathbf{1}\{i \in \mathcal{S}_n^1\} \right) + \exp\left( -\sum_{j \in \mathcal{E}_{n,i}^2} \frac{(\gamma_j - \mu_{n,ij})^2}{2\sigma_{ij}^2/T_{n,i}} \mathbf{1}\{i \in \mathcal{S}_n^2\} \right).$$

We have

$$\Gamma^{\mathrm{F}} = \lim_{n \to \infty} -\frac{1}{n} \log(1 - \mathbb{P}\{\mathcal{S}_n^1 = \mathcal{S}^1\}) = \min_{i \in \mathbb{A}} w_i \min_{j \in \mathcal{E}_i^1} \frac{(\gamma_j - \mu_{ij})^2}{2\sigma_{ij}^2} \mathbf{1}\{i \in \mathcal{S}^1\} + w_i \sum_{j \in \mathcal{E}_i^2} \frac{(\gamma_j - \mu_{ij})^2}{2\sigma_{ij}^2} \mathbf{1}\{i \in \mathcal{S}^2\}.$$

$$(\mathrm{F.3})$$

By (F.2),

$$\Gamma^{\mathrm{F}} = w_i \min_{j \in \mathcal{E}_i^1} \frac{(\gamma_j - \mu_{ij})^2}{2\sigma_{ij}^2} \mathbf{1}\{i \in \mathcal{S}^1\} + w_i \sum_{j \in \mathcal{E}_i^2} \frac{(\gamma_j - \mu_{ij})^2}{2\sigma_{ij}^2} \mathbf{1}\{i \in \mathcal{S}^2\}, \quad \forall i \in \mathbb{A}, \qquad (\mathrm{F.4})$$

where $w_i$ in (F.3) and (F.4) is the solution of (16) in the main text.

Next, we will show that for any feasible arm identification algorithms, $\lim_{n \to \infty} -\frac{1}{n} \log(1 - \mathbb{P}\{\mathcal{S}_n^1 = \mathcal{S}^1\}) \leq \Gamma^{\mathrm{F}}$. Let $W \triangleq \{\boldsymbol{w} = (w_1, \ldots, w_k) : \sum_{i=1}^k w_i = 1 \text{ and } w_i \geq 0, \forall i \in \mathbb{A}\}$ be set of the feasible sampling rates of the $k$ arms. We prove it by contradiction. Suppose there exists a policy with sampling rates $\boldsymbol{w}' = (w_1', w_2', \ldots, w_k')$ of the $k$ arms such that

$$w_i' \min_{j \in \mathcal{E}_i^1} \frac{(\gamma_j - \mu_{ij})^2}{2\sigma_{ij}^2} \mathbf{1}\{i \in \mathcal{S}^1\} + w_i' \sum_{j \in \mathcal{E}_i^2} \frac{(\gamma_j - \mu_{ij})^2}{2\sigma_{ij}^2} \mathbf{1}\{i \in \mathcal{S}^2\}$$

$$= \max_{\boldsymbol{w} \in W} \min_{i \in \mathbb{A}} w_i \min_{j \in \mathcal{E}_i^1} \frac{(\gamma_j - \mu_{ij})^2}{2\sigma_{ij}^2} \mathbf{1}\{i \in \mathcal{S}^1\} + w_i \sum_{j \in \mathcal{E}_i^2} \frac{(\gamma_j - \mu_{ij})^2}{2\sigma_{ij}^2} \mathbf{1}\{i \in \mathcal{S}^2\}.$$

We will show that $\max_{\boldsymbol{w} \in W} \min_{i \in \mathbb{A}} w_i \min_{j \in \mathcal{E}_i^1} \frac{(\gamma_j - \mu_{ij})^2}{2\sigma_{ij}^2} \mathbf{1}\{i \in \mathcal{S}^1\} + w_i \sum_{j \in \mathcal{E}_i^2} \frac{(\gamma_j - \mu_{ij})^2}{2\sigma_{ij}^2} \mathbf{1}\{i \in \mathcal{S}^2\}$
is achieved when

$$
\begin{aligned}
& w_i \min_{j \in \mathcal{E}_i^1} \frac{(\gamma_j - \mu_{ij})^2}{2\sigma_{ij}^2} \mathbf{1}\{i \in \mathcal{S}^1\} + w_i \sum_{j \in \mathcal{E}_i^2} \frac{(\gamma_j - \mu_{ij})^2}{2\sigma_{ij}^2} \mathbf{1}\{i \in \mathcal{S}^2\} \\
& = w_{i'} \min_{j \in \mathcal{E}_{i'}^1} \frac{(\gamma_j - \mu_{i'j})^2}{2\sigma_{i'j}^2} \mathbf{1}\{i' \in \mathcal{S}^1\} + w_{i'} \sum_{j \in \mathcal{E}_{i'}^2} \frac{(\gamma_j - \mu_{i'j})^2}{2\sigma_{i'j}^2} \mathbf{1}\{i' \in \mathcal{S}^2\}, \quad i \neq i'.
\end{aligned}
\tag{F.5}
$$

Since the solution of (F.5) is unique, there exists an arm $i'$ satisfying

$$
\begin{aligned}
& w_{i'}' \min_{j \in \mathcal{E}_{i'}^1} \frac{(\gamma_j - \mu_{i'j})^2}{2\sigma_{i'j}^2} \mathbf{1}\{i' \in \mathcal{S}^1\} + w_{i'}' \sum_{j \in \mathcal{E}_{i'}^2} \frac{(\gamma_j - \mu_{i'j})^2}{2\sigma_{i'j}^2} \mathbf{1}\{i' \in \mathcal{S}^2\} \\
& \geq \min_{i \in \mathbb{A}} w_i' \min_{j \in \mathcal{E}_i^1} \frac{(\gamma_j - \mu_{ij})^2}{2\sigma_{ij}^2} \mathbf{1}\{i \in \mathcal{S}^1\} + w_i' \sum_{j \in \mathcal{E}_i^2} \frac{(\gamma_j - \mu_{ij})^2}{2\sigma_{ij}^2} \mathbf{1}\{i \in \mathcal{S}^2\}.
\end{aligned}
$$

We consider a new policy. There exists $\delta_{20} > 0$ such that $\tilde{w}_{i'} = w_{i'}' - \delta_{20} \in (0, 1)$ and $\tilde{w}_i = w_i' + \delta_{20}/(k-2) \in (0, 1)$ for $i \neq i'$. Then

$$
\begin{aligned}
& \min_{i \in \mathbb{A}} \tilde{w}_i \min_{j \in \mathcal{E}_i^1} \frac{(\gamma_j - \mu_{ij})^2}{2\sigma_{ij}^2} \mathbf{1}\{i \in \mathcal{S}^1\} + \tilde{w}_i \sum_{j \in \mathcal{E}_i^2} \frac{(\gamma_j - \mu_{ij})^2}{2\sigma_{ij}^2} \mathbf{1}\{i \in \mathcal{S}^2\} \\
& > \min_{i \in \mathbb{A}} w_i' \min_{j \in \mathcal{E}_i^1} \frac{(\gamma_j - \mu_{ij})^2}{2\sigma_{ij}^2} \mathbf{1}\{i \in \mathcal{S}^1\} + w_i' \sum_{j \in \mathcal{E}_i^2} \frac{(\gamma_j - \mu_{ij})^2}{2\sigma_{ij}^2} \mathbf{1}\{i \in \mathcal{S}^2\} \\
& = \max_{\boldsymbol{w} \in W} \min_{i \in \mathbb{A}} w_i \min_{j \in \mathcal{E}_i^1} \frac{(\gamma_j - \mu_{ij})^2}{2\sigma_{ij}^2} \mathbf{1}\{i \in \mathcal{S}^1\} + w_i \sum_{j \in \mathcal{E}_i^2} \frac{(\gamma_j - \mu_{ij})^2}{2\sigma_{ij}^2} \mathbf{1}\{i \in \mathcal{S}^2\},
\end{aligned}
$$

which yields a contradiction. Therefore, the equations in (16) of the main text hold.

## G    Additional Numerical Results

In this section, we provide additional numerical results for the experiments conducted in Section 5 of the main text. Figures 1(a)-7(a) show how the probabilities of false selection of the compared algorithms change with the sample sizes for the best arm identification problem, and Figures 1(b)-7(b) show the sampling rates of the algorithms on some selected arms. It can be observed in Figures 1(a)-7(a) that the proposed iKG algorithm performs the best, followed by TTEI, KG, EI and the equal allocation. On the log scale, the probability of false selection (PFS) values of the iKG algorithm demonstrate linear patterns, indicating the potentially exponential convergence rates. For both EI and KG, the rates of the posterior convergence are not optimal, which might influence their empirical performances. The equal allocation performs the worst in general. In Figures 1(b)-7(b), we can see that TTEI always allocates half samples to the best arm when $\beta = 0.5$, EI allocates too many samples to the best arm while KG allocates too few samples to the best arm.

Figures 8(a)-14(a) show how the probabilities of false selection of the compared algorithms change with the sample sizes for the $\epsilon$-good arm identification, and Figures 8(b)-14(b) show the sampling rates of the algorithms on some selected arms. It can be observed in Figures 8(a)-14(a) the proposed iKG-$\epsilon$ algorithm performs the best and demonstrates a linear pattern on the log scale. $(\text{ST})^2$ and APT are inferior, and the equal allocation performs the worst. In Figures 8(b)-14(b), we can see that APT allocates too few samples to the best arm and too many samples to the arms near the threshold. $\text{ST}^2$ allocates too few samples to the best arm, which influences the accuracy of the threshold.

Figures 15(a)-21(a) show how the probabilities of false selection of the compared algorithms change with the sample sizes for the feasible arm identification, and Figures 15(b)-21(b) show the sampling rates of the algorithms on some selected arms. The results in Figures 15(a)-21(a) are similar to those in Figures 1(a)-14(a). The proposed iKG-F algorithm has the best performance, followed by the compared MD-UCBE, equal allocation and MD-SAR. In Figures 15(b)-21(b), we can see that MD-UCBE and MD-SAR allocate too many samples to the arms near the constraint limits.

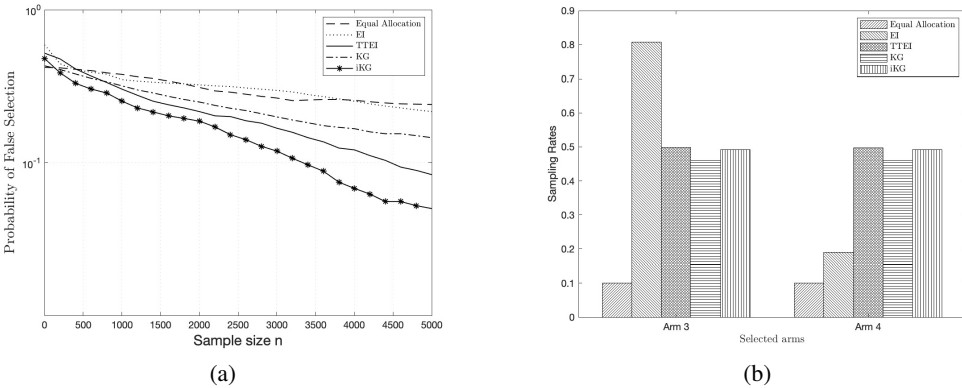

(a)                                         (b)

Figure 1: PFS and sampling rates of selected arms for the best arm identification (Example 1)

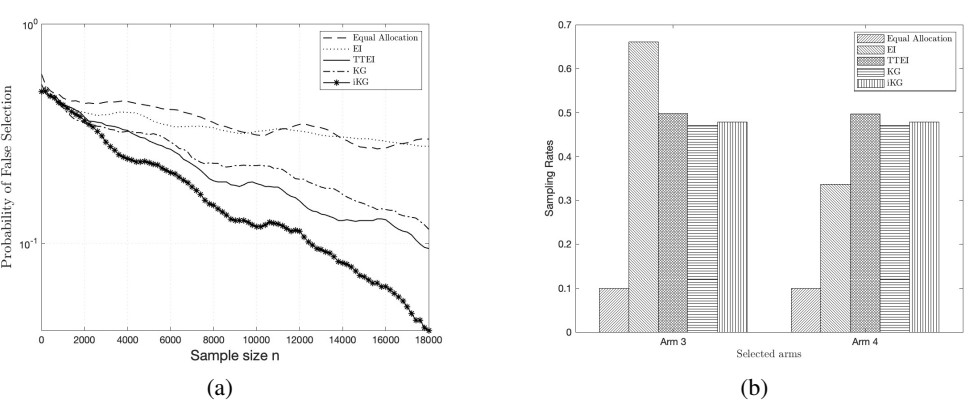

(a)                                         (b)

Figure 2: PFS and sampling rates of selected arms for the best arm identification (Example 2)

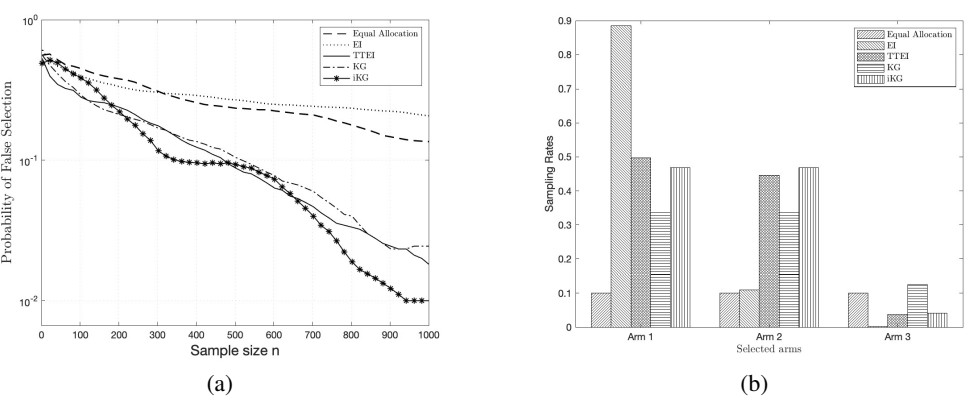

(a)                                         (b)

Figure 3: PFS and sampling rates of selected arms for the best arm identification (Example 3)

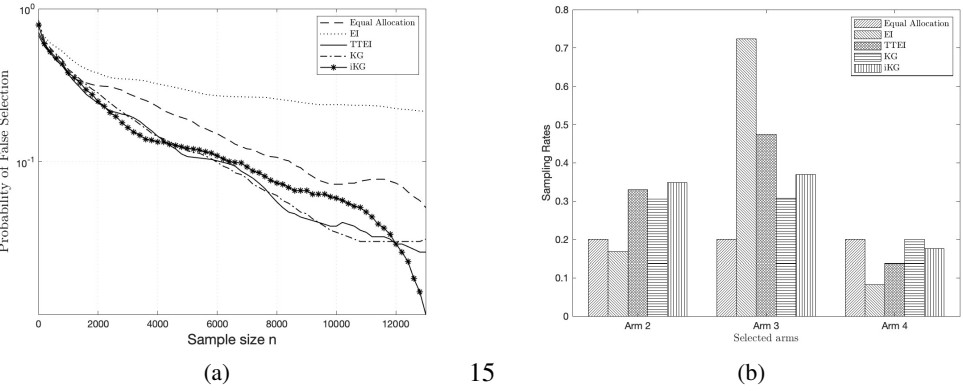

(a)                        15                (b)

Figure 4: PFS and sampling rates of selected arms for the best arm identification (Dose-Finding Problem)

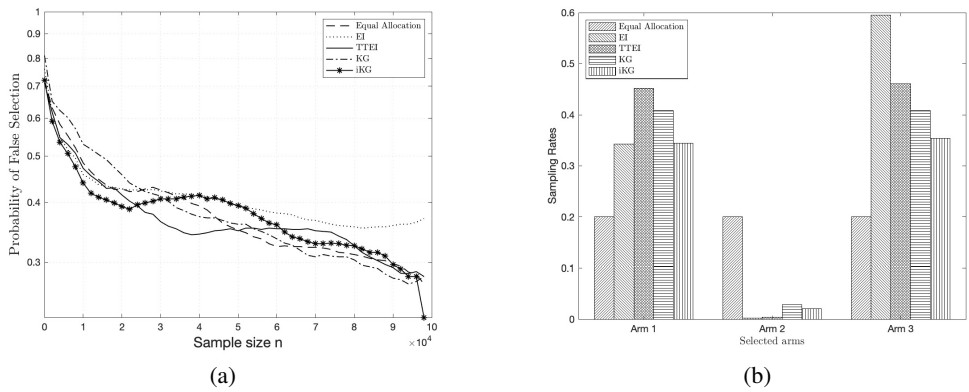

(a)                                              (b)

Figure 5: PFS and sampling rates of selected arms for the best arm identification (Drug Selection Problem)

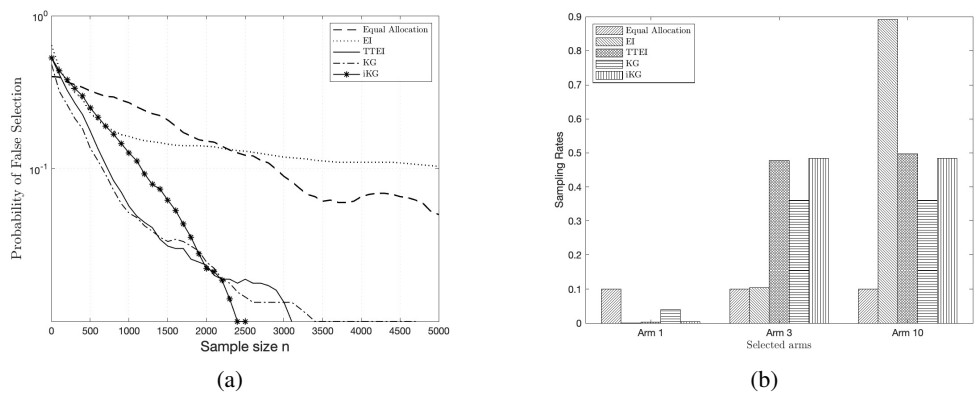

(a)                                              (b)

Figure 6: PFS and sampling rates of selected arms for the best arm identification (Caption 853)

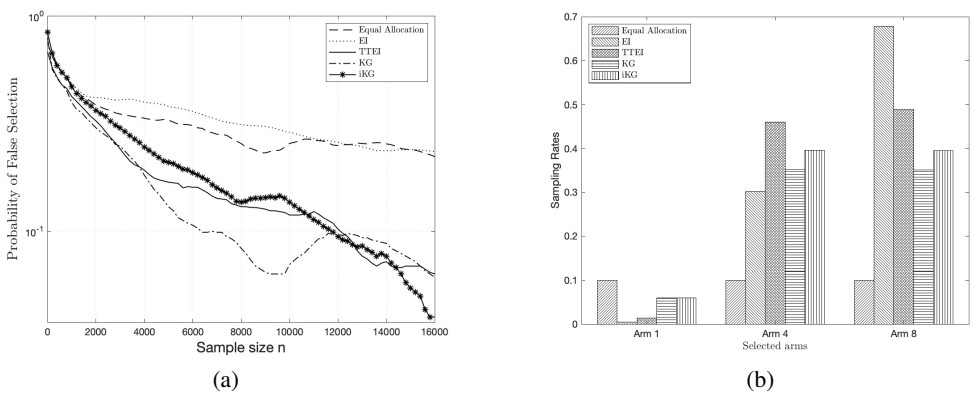

(a)                                              (b)

Figure 7: PFS and sampling rates of selected arms for the best arm identification (Caption 854)

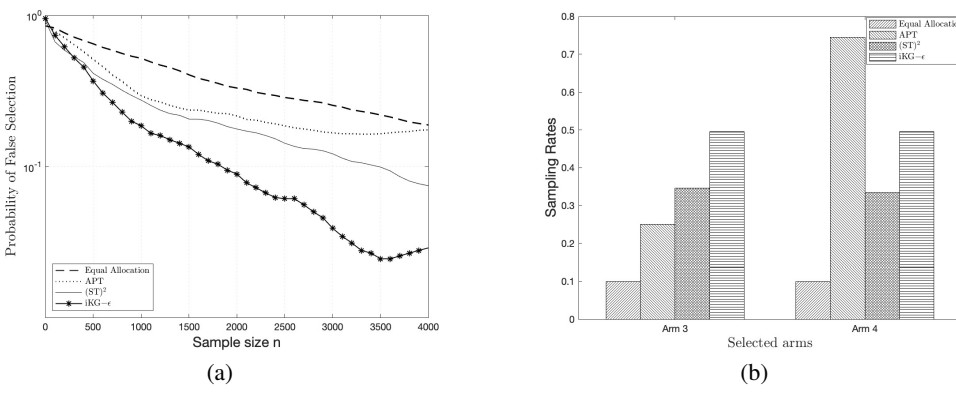

Figure 8: PFS and sampling rates of selected arms for the $\epsilon$-good arm identification (Example 1)

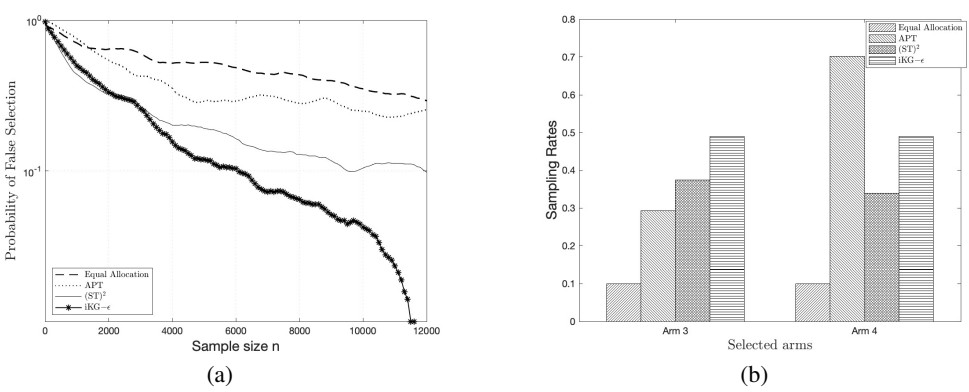

Figure 9: PFS and sampling rates of selected arms for the $\epsilon$-good arm identification (Example 2)

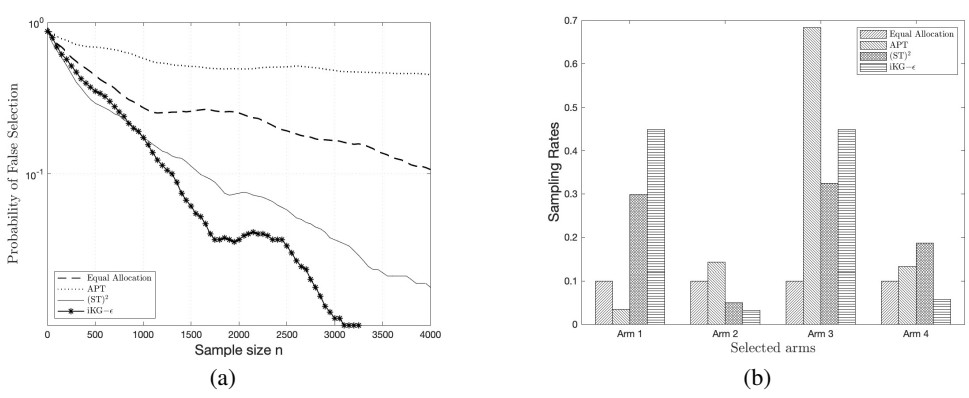

Figure 10: PFS and sampling rates of selected arms for the $\epsilon$-good arm identification (Example 3)

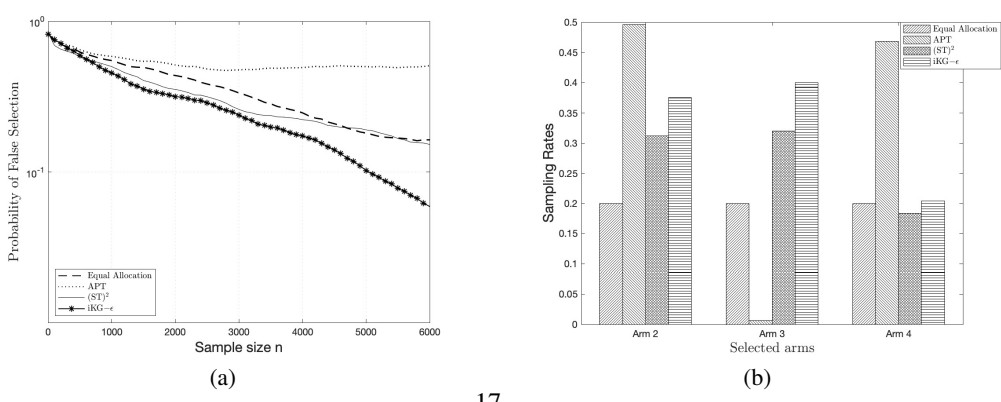

17

Figure 11: PFS and sampling rates of selected arms for the $\epsilon$-good arm identification (Dose-Finding Problem)

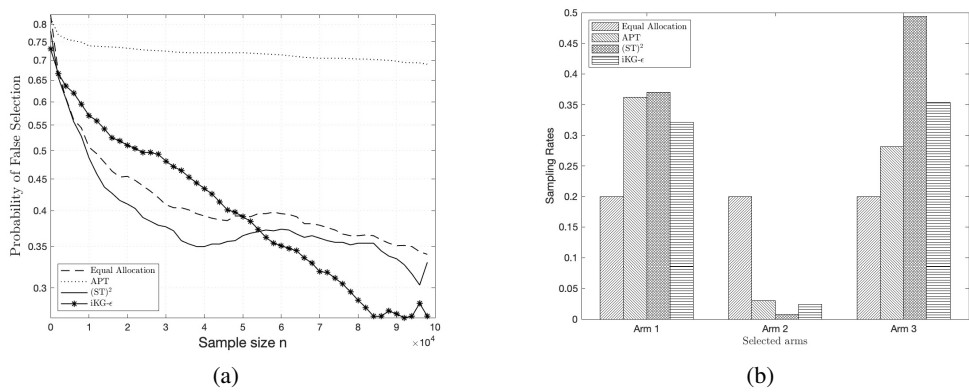

(a)                                         (b)

Figure 12: PFS and sampling rates of selected arms for the $\epsilon$-good arm identification (Drug Selection Problem)

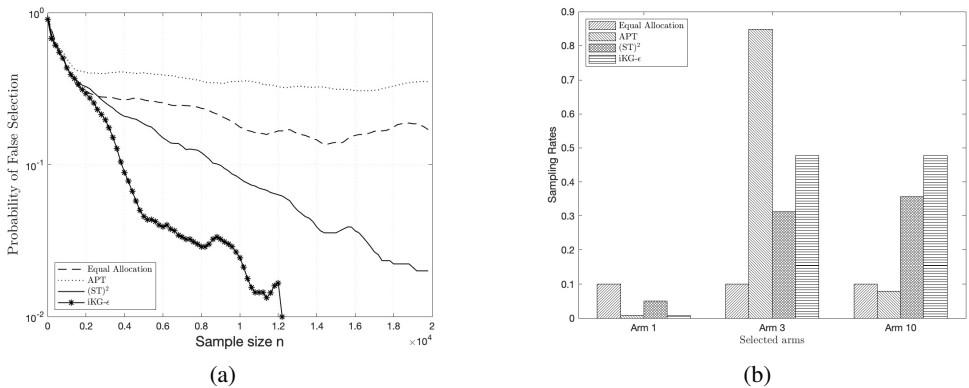

(a)                                         (b)

Figure 13: PFS and sampling rates of selected arms for the $\epsilon$-good arm identification (Caption 853)

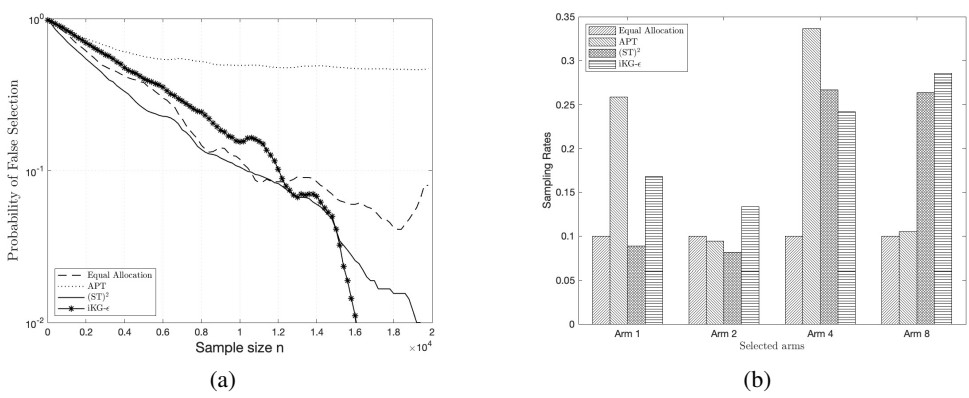

(a)                                         (b)

Figure 14: PFS and sampling rates of selected arms for the $\epsilon$-good arm identification (Caption 854)

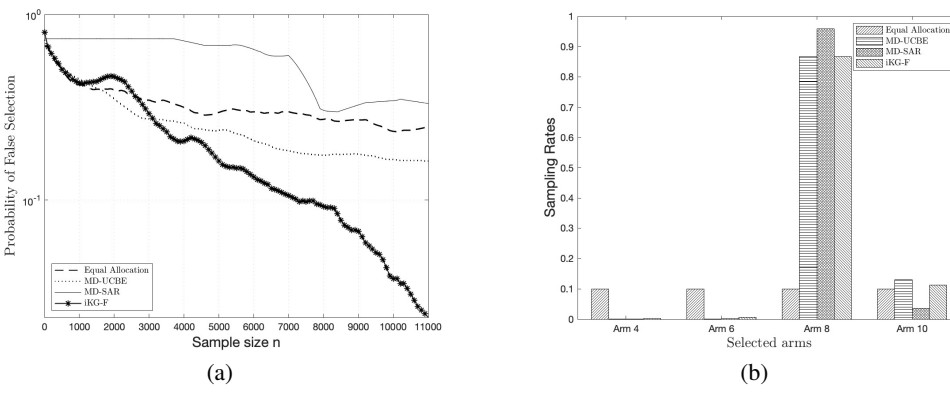

(a)          (b)

Figure 15: PFS and sampling rates of selected arms for the feasible arm identification (Example 1)

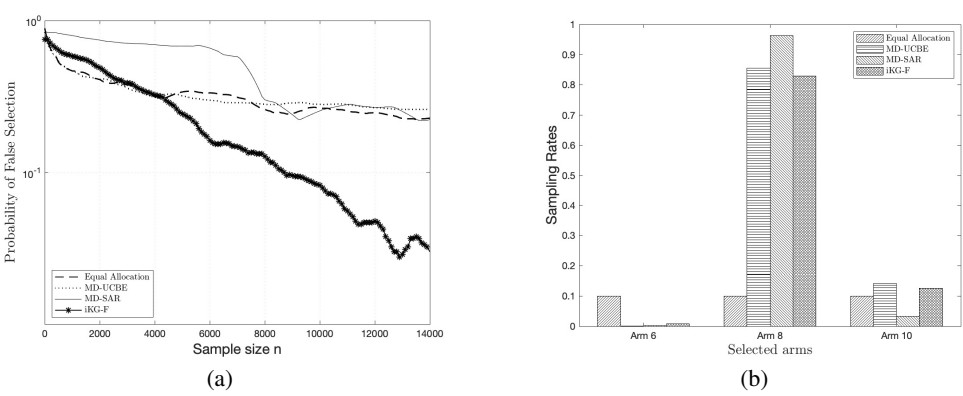

(a)          (b)

Figure 16: PFS and sampling rates of selected arms for the feasible arm identification (Example 2)

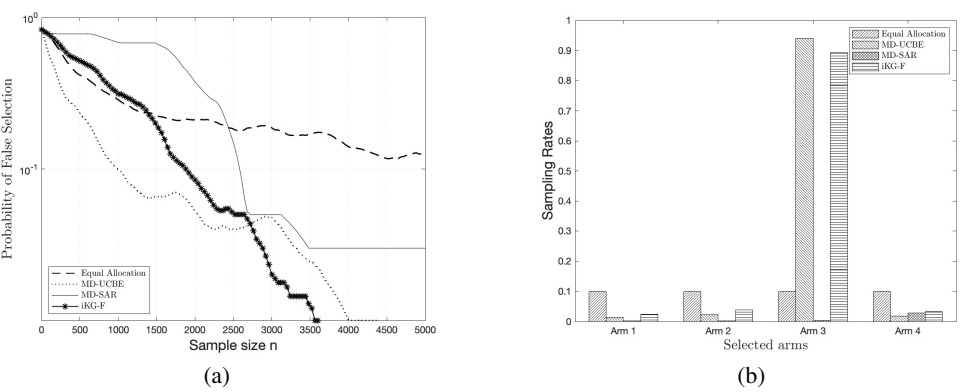

(a)          (b)

Figure 17: PFS and sampling rates of selected arms for the feasible arm identification (Example 3)

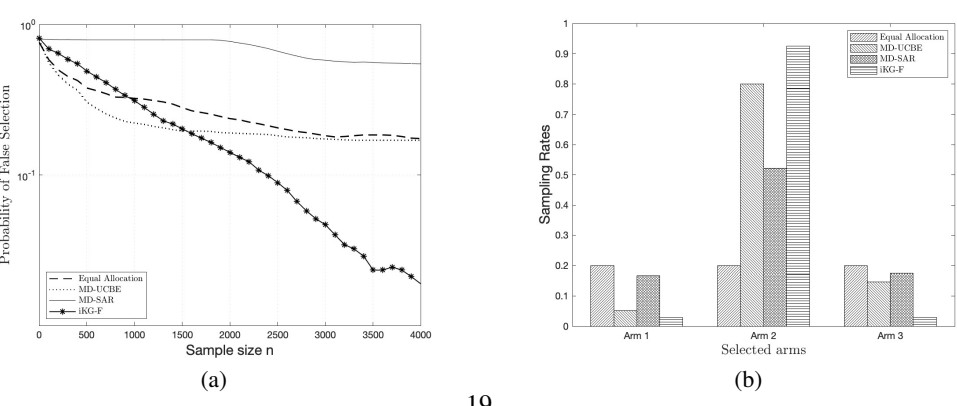

(a)          (b)

Figure 18: PFS and sampling rates of selected arms for the feasible arm identification (Dose-Finding Problem)

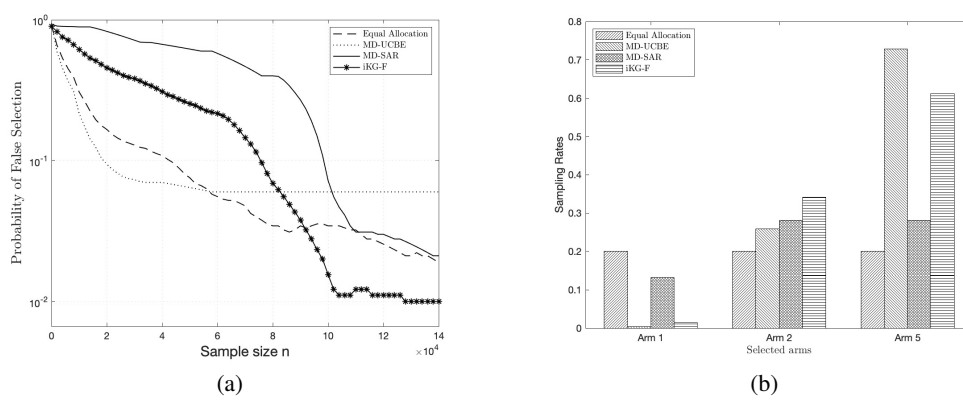

(a)                                                          (b)

Figure 19: PFS and sampling rates of selected arms for the feasible arm identification (Drug Selection Problem)

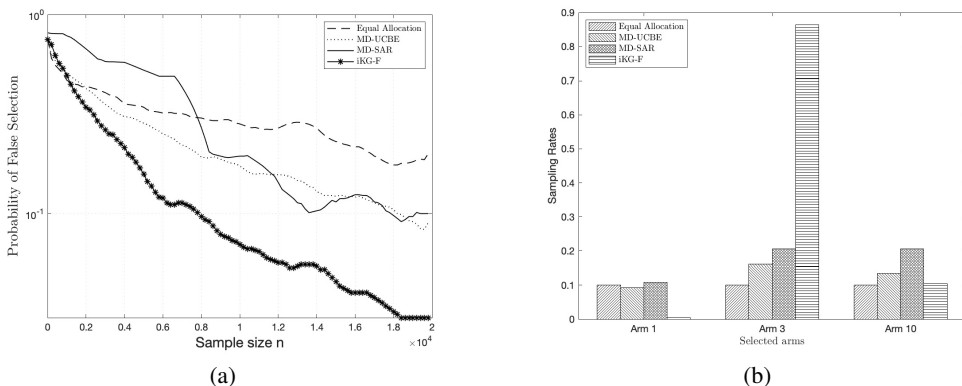

(a)                                                          (b)

Figure 20: PFS and sampling rates of selected arms for the feasible arm identification (Caption 853)

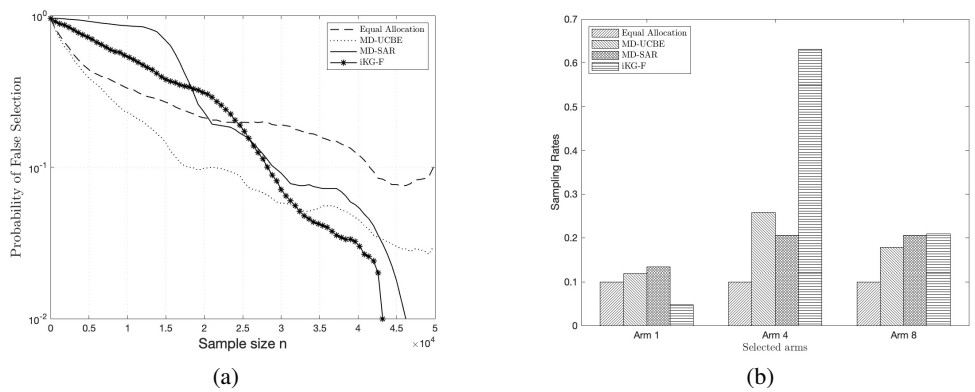

(a)                                                          (b)

Figure 21: PFS and sampling rates of selected arms for the feasible arm identification (Caption 854)