# OpenReview forum: "Improving the Knowledge Gradient Algorithm"
_NeurIPS.cc/2023/Conference — NeurIPS 2023 poster_

### Official Review · Reviewer_dyGQ · 2023-06-12

**Soundness:** 3 good
**Presentation:** 2 fair
**Contribution:** 3 good
**Rating:** 7
**Confidence:** 3

**Summary:**

The paper tackles the problem of best arm identification (BAI). The paper introduces "improved knowledge gradient (iKG)" that can be shown to be asymptotically optimal. Some numerical results are also provided

**Strengths:**

The contribution of the paper has an interesting scope of showing limitations of an existing algorithm for best-arm identification (BAI), providing a variant along with theoretical analysis and numerical experiments. The paper also discussed variant problems of BAI making the scope quite large within the multi-arm bandit setting.
The paper seem to me scientifically sound and is overall well-written.

**Weaknesses:**

(minor) Some elements in the formalization, some notations and minor typos could possibly be improved.

Disclaimer: I can't fully assess the novelty as I'm not an expert in the field of the best arm identification problem and I didn't check in details all the proofs.

**Questions:**

- In the beginning of Section 2, $i$ refers to the arms and then it is mentioned that $\sigma_i$ are known variances but then in Equation 1, the sigma is estimated. Can you clarify this?

Minor additional comment:
- There are few typos/english mistakes: e.g. "this policy has limitations, causing the algorithm not asymptotically optimal", "These algorithms are not optimal in general except there is only one round left".
- I know that it is a usual notation to say something like "$\mu_1 > \mu_2 \ge . . . \ge \mu_k$". But this feels strange as it would (naively seem to) imply that one already knows that the arm i=1 is the best. Is it possible to introduce another notation or do I miss something?
- Line 70: $\theta_i$ is defined as "the random variable following *its* posterior distribution". Is it the $\mu$ only or is it for the whole state $S_t$.
- Line 74: The transition function that depends on two states and one action and deterministically transition to a new state is not entirely clear to me from a formal perspective. $\theta_{t,i}$ is defined as a random variable following some (posterior) distribution but it is then apparently also a "state"? Isn't it a probability distribution over the state space?


**Limitations:**

The limitations are adequately discussed.

---

> ### Author Rebuttal · Authors · 2023-08-09
>
> We would like to thank the reviewer for the careful and detailed review and insightful comments. In the following, we provide point-by-point detailed responses.
>
> **Q:** (minor) Some elements in the formalization, some notations and minor typos could possibly be improved.
>
> **A:** Thanks for the comment. We will conduct a thorough round of editorial check to improve the readability of the manuscript.
>
> **Q:** In the beginning of Section 2, $i$ refers to the arms and then it is mentioned that $\sigma_{i}^2$ are known variances but then in Equation 1, the sigma is estimated. Can you clarify this?
>
> **A:** In this research, $\sigma_{i}^{2}$ is the sampling variance of arm $i$ and is assumed to be known. In the Bayesian model, the unknown mean $\mu_{i}$ is treated as random, and $\theta_{i}$ is denoted as the random variable following the posterior distribution of $\mu_{i}$. The term $\sigma_{t, i}^{2}$ (with subscript $t$, different from the $\sigma_{i}^{2}$ above) refers to the variance of the random variable $\theta_{i}$ in round $t$, which needs to be estimated by Eqn. (1). We will make it clear in the manuscript.
>
> **Q:** There are a few typos/english mistakes: e.g. "this policy has limitations, causing the algorithm not asymptotically optimal", "These algorithms are not optimal in general except there is only one round left".
>
> **A:** Thanks for pointing them out. We will fix them and conduct a thorough round of editorial check and improvement for the manuscript.
>
> **Q:** I know that it is a usual notation to say something like "$\mu_{1}>\mu_{2}\geq\ldots\geq\mu_{k}$". But this feels strange as it would (naively seem to) imply that one already knows that the arm i=1 is the best. Is it possible to introduce another notation or do I miss something?
>
> **A:** We agree with the reviewer. In the updated manuscript, we will replace it by "$\mu_{\langle 1 \rangle}>\mu_{\langle 2 \rangle}\geq\ldots\geq\mu_{\langle k \rangle}$", with arm $\langle 1 \rangle$ being the best arm.
>
> **Q:** Line 70: $\theta_{i}$ is defined as "the random variable following its posterior distribution". Is it the $\mu$ only or is it for the whole state $S_{t}$.
>
> **A:** The term $\theta_{i}$ is the random variable for the unknown mean of arm $i$, which follows $\mathcal{N}(\mu_{t,i}, \sigma_{t,i}^{2})$. The  state $S_{t}$ consists of the posterior means and variances of all the $k$ arms in round $t$. We will make it clear in the updated manuscript.
>
> **Q:** Line 74: The transition function that depends on two states and one action and deterministically transition to a new state is not entirely clear to me from a formal perspective. $\theta_{t,i}$ is defined as a random variable following some (posterior) distribution but it is then apparently also a "state"? Isn't it a probability distribution over the state space?
>
> **A:** The agent chooses the measurement that yields the greatest one-step improvement in the probability of selecting the best arm. However, we do not know state $S_{t+1}$ for calculating the probability of selecting the best arm in round $t+1$, so we construct a predictive model to calculate state $S_{t+1}$. We use $\theta_{t,i}$ (following the normal distribution with posterior mean $\mu_{t,i}$ and variance $\sigma_{i}^2$) as the sample of arm $i$ in round $t$. State $S_{t+1}$ is obtained by pulling arm $i$ and generating $\theta_{t,i}$, and then $\mathcal{T}(S_t,i,\theta_{t,i})$ can be computed by Eqn. (1). Hence $\theta_{t,i}$ is not a "state" and is not a probability distribution over the state space either. We will make it clear in the updated manuscript.

---

> > ### Comment · Reviewer_dyGQ · 2023-08-14
> > **Thanks for the clarifications**
> >
> > Thanks for the clarifications. I keep my score of 7 unchanged.

---

### Official Review · Reviewer_eJBQ · 2023-07-04

**Soundness:** 4 excellent
**Presentation:** 4 excellent
**Contribution:** 2 fair
**Rating:** 5
**Confidence:** 5

**Summary:**

The paper focuses on the fixed-budget best arm identification (BAI) problem, in which an agent aims to correctly identify the best arm (one with the highest mean) among k given arms within a known fixed number of samples. An improved knowledge gradient (iKG) policy is proposed, which prioritizes the arm that maximizes the one-step improvement in the probability of selecting the best arm. The algorithm yields an asymptotically optimal solution, overcoming the limitations of the knowledge gradient (KG) policy that is not rate optimal. The paper also extends iKG to the problem of ε-good arm identification and feasible arm identification examples.

**Strengths:**

The idea of considering probability instead of expectation is new in this context (even though it has already been used in other contexts). The analyses of extensions is also an added benefit.

**Weaknesses:**

The computational experiments are weak, in particular the choice (more precisely, creation) of datasets bags for more. In short, the experiments are mostly based on synthetic data and some of them come across as very far fetched.

The concepts are very similar to TTEI which makes the contributions marginal.

**Questions:**

Why not comparing iKG with KG, for example in the 'one average' sense? I do understand that one is asymptotically optimal and the other not, but perhaps the cases when KG is not are very rare and thus the choice between iKG and KG would be a wash.

The datasets are really contrived. Why not getting more real-world data and then tweaking it to fit the purpose?

---

> ### Author Rebuttal · Authors · 2023-08-09
>
> We would like to thank the reviewer for the careful and detailed review and insightful comments. In the following, we provide point-by-point detailed responses.
>
> **Q:** The computational experiments are weak, in particular the choice (more precisely, creation) of datasets bags for more. In short, the experiments are mostly based on synthetic data and some of them come across as very far fetched.
>
> **A:** Thanks for the comment. We agree that to better illustrate the empirical performances of the proposed algorithms, more examples should be tested. During the rebuttal period, we have conducted numerical tests on three additional real examples from two new datasets. Please see our global response for detailed numerical results. We will add these results to the updated manuscript.
>
> **Q:** The concepts are very similar to TTEI which makes the contributions marginal.
>
> **A:** We agree that in terms of the objective, both TTEI and this paper aim to modify an existing popular BAI algorithm to asymptotic optimality. In fact, we believe that developing (asymptotic) optimal algorithms, either based on new ideas or by improving existing suboptimal algorithms (like in TTEI and this paper), should be the same goal of all BAI (and other) research.
>
> More importantly, we would like to point out that, how we improve KG to asymptotic optimality is entirely different from that for EI. The iKG algorithm is based on the concept of selecting the measurement that results in the greatest one-step improvement in the probability of choosing the best arm. It gives rise to a completely different acquisition function from KG. It is not just a new idea that can be used to improve KG. It is also an algorithm design framework that can be used for different BAI variant problems, as is shown in the examples of $\epsilon$-good arm identification and feasible arm identification. TTEI only changes the sampling rate of the best arm of EI. This level of modification to EI is minor compared to the way we modify KG.
>
> **Q:** Why not comparing iKG with KG, for example in the "one average'' sense? I do understand that one is asymptotically optimal and the other not, but perhaps the cases when KG is not are very rare and thus the choice between iKG and KG would be a wash.
>
> **A:** Now the empirical performances of KG and iKG are based on the average of multiple replications of the algorithms. It is the typical way of reporting and studying the algorithm performances. In the meantime, we appreciate the suggestion of the reviewer. In each replication of the algorithm, due to the uncertainty of the sample path, the relative performances of KG and iKG might deviate from the average pattern. It provides another direction for numerically comparing KG and iKG. From our experience of doing numerical experiments, based on one replication of the two algorithms, if they perform differently, it is highly likely the iKG correctly selects the best arm while KG does not. It is rare to observe the other way. In the updated manuscript, we will report the percentage of the replications when KG correctly selects the best arm while iKG does not, as suggested by the reviewer. We will also discuss the result, seeking to obtain some guidance on choosing between KG and iKG.
>
> On a related note, as mentioned in our global response, we will strengthen the numerical part of this paper by adding three real examples in the numerical experiments. These additional examples can help us better understand the relative performances of KG and iKG.

---

> > ### Comment · Reviewer_eJBQ · 2023-08-17
> >
> > I appreciate answering my concerns. My opinion hasn't changed.

---

### Official Review · Reviewer_2tPM · 2023-07-05

**Soundness:** 4 excellent
**Presentation:** 4 excellent
**Contribution:** 3 good
**Rating:** 7
**Confidence:** 4

**Summary:**

The paper is concerned with the best arm identification (BAI) problem. The main contribution of the paper is the proposal of an improved version of the knowledge gradient algorithm which is also shown to be asymptotically optimal. Moreover, the proposed algorithm is extended to the variant of BAI problem and illustrated on certain applications.

**Strengths:**

- The main message and contributions of the paper are clearly stated and demonstrated.
- The advantage of the proposed algorithm is theoretically proven, by showing that it is asymptotically optimal.
- By addressing several variants of the BAI problem, the versatility of the approach is shown, supported by theoretical results.
- The impact of the proposed method on several applications are shown.


**Weaknesses:**

- The limitations of the proposed method is not discussed
- The range of the problems the proposed approach can solve might be narrow.


**Questions:**

- Can you describe the limitations of the proposed approach, and possibly directions for future research?

---

> ### Author Rebuttal · Authors · 2023-08-09
>
> We would like to thank the reviewer for the careful and detailed review and insightful comments. In the following, we provide point-by-point detailed responses.
>
> **Q:** The limitation of the proposed method is not discussed.
>
> **A:** The proposed iKG algorithm improves the KG algorithm to asymptotic optimality. A potential limitation of this research is that, our theoretical development (especially the superiority of iKG over KG) has been established in the asymptotic regime. It is not clear if this result can be extended to the finite-time environment. In fact, it is a common challenge faced by most fixed-budget BAI algorithms, that the finite-time performances are difficult to characterize.
> We believe that one way to address this issue is to use the numerical test. In our last submission, we compared iKG and KG on three synthetic and one real datasets, and iKG has demonstrated better finite-time performance than KG on all these four examples. In the updated manuscript, we will strengthen the comparison results. Specifically, we will compare iKG and KG on three more real examples: one based on the Drug Review Dataset (https://doi.org/10.24432/C5SK5S) for selecting effective contraceptives, and the other two based on the New Yorker Cartoon Caption Contest Dataset (https://nextml.github.io/caption-contest-data/) for selecting good captions. Please see our global response for more details. These additional real examples will help us better understand the relative performances of iKG and KG.
>
> **Q:** The range of the problems the proposed approach can solve might be narrow.
>
> **A:** We would like to point out that, the setting of this research is common and general. The range of problems that can be solved by the proposed iKG algorithm is actually quite broad. The problems that can be solved by typical BAI algorithms such as KG (Frazier, 2008), EI (Ryzhov, 2016), TTEI (Qin et al., 2017), TTTS (Russo, 2020), etc. can all be solved by the iKG algorithm.
>
> We believe that one possible reason leading to this concern of the reviewer is that, in our numerical experiments, we only tested the algorithms on one real dataset. In the updated manuscript, we will add three more real examples to the test. Please see our global response for more details. We hope that the new experiments and results can dispel the reviewer's concern about the range of applicability of our algorithms.
>
> **References:**
> P. I. Frazier, W. B. Powell, and S. Dayanik. A knowledge-gradient policy for sequential information collection. *SIAM Journal on Control and Optimization*, 47(5):2410–2439, 2008.
> I. O. Ryzhov. On the convergence rates of expected improvement methods. *Operations Research* 64.6 (2016): 1515-1528.
> C. Qin, D. Klabjan, and D. Russo. Improving the expected improvement algorithm. In *Proceedings of the 31st International Conference on Neural Information Processing Systems*, pages 5387–5397, 2017.
> D. Russo. Simple Bayesian algorithms for best arm identification. *Operations Research*, 68(6):1625–1647, 2020.

---

> > ### Comment · Reviewer_2tPM · 2023-08-15
> >
> > Thanks for the response to my questions. My review remains positive, although I hoped the authors would give a better discussion on the limitations that leads to directions for future research. The current response discusses a limitation of the analysis that does not capture the strength of the algorithm.

---

### Official Review · Reviewer_h3kD · 2023-07-07

**Soundness:** 3 good
**Presentation:** 2 fair
**Contribution:** 3 good
**Rating:** 6
**Confidence:** 2

**Summary:**

The paper considers knowledge gradient (KG) algorithm and its limitations in the best arm identification (BAI) problem. The authors first demonstrate that the KG policy is not asymptotically optimal, highlighting its shortcomings, and then provide the improved knowledge gradient (iKG) policy. The key idea is to select measurements that maximize the one-step improvement in the probability of selecting the best arm, rather than the mean improvement. The paper establishes that iKG is asymptotically optimal, offering a remedy to the limitations of KG. The empirical evidence for the effectiveness of the proposed method is also provided.

**Strengths:**

1. The theoretical analysis on the KG algorithm on its sub-optimality gap
2. This work proposed a novel improved KG algorithm with optimality guarantees
3. Discussion on the variant problems of BAI shows that the proposed iKG method is general and can be used in various scenarios.

**Weaknesses:**

1. Some notations are rather unclear, e.g., the definition of subscript $i$ in line 61 is not introduced until line 64, definition of $I_t$
2. What is the computational complexity comparing with KG? It is beneficial to discuss its relationship with the scale of the problem.

**Questions:**

1. How does the last inequality in Eqn. F.2 hold?

**Limitations:**

The author has addressed the limitations well.

---

> ### Author Rebuttal · Authors · 2023-08-09
>
> We would like to thank the reviewer for the careful and detailed review and insightful comments. In the following, we provide point-by-point detailed responses.
>
> **Q:** Some notations are rather unclear, e.g., the definition of subscript $i$ in line 61 is not introduced until line 64, definition of $I_{t}$.
>
> **A:** Thanks for pointing it out. The notation $i$ is the index for the $i$-th arm in the $k$ arms. The notation $I_{t}$ refers to the arm the agent pulls in round $t$. We will explain them when they first appear. We will also improve the notation system and conduct a thorough round of editorial check to improve the readability of the manuscript.
>
> **Q:** What is the computational complexity comparing with KG? It is beneficial to discuss its relationship with the scale of the problem.
>
> **A:** The computational complexities of the iKG and KG algorithms are $O(kn)$ and $O(k^2n)$ respectively, where $k$ is the total number of arms and $n$ is the total number of rounds. Although the two algorithms are based on similar frameworks,
> the computation of $KG_{t,i}$ is $O(k)$ due to the term $\max\{\mathcal{T}(\mu_{t,i},i,\theta_{t,i}), \max_{i'\neq i}\mu_{t,i'}\}$, greater than that of $iKG_{t,i}$ which is $O(1)$. Therefore, the proposed iKG is actually computationally more efficient than KG and has better scalability for large-scale problems. We will add this discussion to the updated manuscript.
>
> **Q:** How does the last inequality in Eqn. F.2 hold?
>
> **A:** Before Eqn. F.2, with the results in Lines 169 and 177 of the Supplement, we have shown that $|\mu_{n,i}-\mu_{i}|<\delta_{15}$ for any $\delta_{15}>0$ and $i=1,2,\ldots,k$. We can find a sufficiently large positive integer $n'$ such that when $n>n'$, $S_{n}^{1}=S^{1}$, $S_{n}^{2}=S^{2}$, $\mathcal{E}\_{n,i}^{1}=\mathcal{E}\_{i}^{1}$ and $\mathcal{E}\_{n,i}^{2}=\mathcal{E}\_{i}^{2}$, where $i=1,2,\ldots,k$ and $j=1,2,\ldots,m$. Note that $S_{n}^{1}\cap S_{2}^{2}=\emptyset$ and $S^{1}\cap S^{2}=\emptyset$. If $i\in S_{n}^{1}=S^{1}$,
> $|iKG_{n,i}^{F}-iKG_{n,i'}^{F}|\leq2m|\exp(-w_{i}\min_{j\in \mathcal{E}\_{n,i}^{1}}\frac{(\gamma_{j}-\mu_{n,ij})^{2}}{2\sigma_{ij}^{2}})-\exp(-w_{i'}\min_{j\in \mathcal{E}\_{n,i'}^{1}}\frac{(\gamma_{j}-\mu_{n,i'j})^{2}}{2\sigma_{i'j}^{2}})|$$\leq 2m|\exp(-w_{i}\min_{j\in \mathcal{E}\_{i}^{1}}\frac{(\gamma_{j}-\mu_{ij}-\delta_{15})^{2}}{2\sigma_{ij}^{2}})-\exp(-w_{i'}\min_{j\in \mathcal{E}\_{i'}^{1}}\frac{(\gamma_{j}-\mu_{i'j}+\delta_{15})^{2}}{2\sigma_{i'j}^{2}})|$. We have shown
> that $|iKG_{n,i}^{F}-iKG_{n,i'}^{F}|<\delta_{18}$ in Line 181. Hence $|w_{i}\min\limits_{j\in\mathcal{E}\_{i}^{1}}\frac{(\gamma_{j}-\mu_{ij})^{2}}{2\sigma_{ij}^{2}}-w_{i'}\min\limits_{j\in\mathcal{E}\_{i'}^{1}}\frac{(\gamma_{j}-\mu_{i'j})^{2}}{2\sigma_{i'j}^{2}}|<\delta_{19}$ by the continuity of the function $\exp(\cdot)$. Similarly, Eqn. F.2 holds if $i\in S_{n}^{2}=S^{2}$. We will add this explanation to the updated supplement.

---

### Official Review · Reviewer_Ye7U · 2023-07-07

**Soundness:** 2 fair
**Presentation:** 3 good
**Contribution:** 3 good
**Rating:** 4
**Confidence:** 2

**Summary:**

The researchers tackle the problem of fixed-budget Best Arm Identification (BAI). Specifically, they observed that typical Knowledge Gradient approaches based on the policy of choosing the step with the best expected one step improvement are not asymptotically optimal. They define a new policy, improved Knowledge Gradient (iKG) that, instead, seeks to improve to create the best one-step improvement in the probability of selecting the best arm.

**Strengths:**

iKG has more applicability than standard KG. They specifically show applications to epsilon-good arm identification and feasible arm identification, which KG cannot handle. They show the application of iKG on synthetic and real datasets and the iKG variants perform the best overall.

**Weaknesses:**

Although I can appreciate the complexity of the numerical examples and the breadth of comparison algorithms for iKG-e and iKG-F, I do not believe there are enough different datasets to talk about the performance generally. The authors should have chosen several datasets of varying natures to show general applicability. It is hard for me to gain a sense of whether these datasets are just lucky examples.

**Questions:**

Can you show comparison algorithms for several other datasets that can duplicate the type of performance seen here? Are there datasets where performance of iKG is worse or not as effective, and can you explain why?

**Limitations:**

The authors discuss the limitations of KG, but not of their techniques specifically. I think this could be accomplished by showing some contrary datasets or examples.

---

> ### Author Rebuttal · Authors · 2023-08-09
>
> We would like to thank the reviewer for the careful and detailed review and insightful comments. In the following, we provide point-by-point detailed responses.
>
> **Q:** Although I can appreciate the complexity of the numerical examples and the breadth of comparison algorithms for iKG-$\epsilon$ and iKG-F, I do not believe there are enough different datasets to talk about the performance generally. The authors should have chosen several datasets of varying natures to show general applicability. It is hard for me to gain a sense of whether these datasets are just lucky examples.
>
> **A:** Thanks for the comment. We agree that to better illustrate the empirical performances of iKG-$\epsilon$ and iKG-F (and iKG), more examples should be tested. During the rebuttal period, we have conducted numerical tests on three additional real examples from two new datasets. Please see our global response for detailed numerical results. We will add these new examples and results to the updated manuscript.
>
> **Q:** Can you show comparison algorithms for several other datasets that can duplicate the type of performance seen here? Are there datasets where the performance of iKG is worse or not as effective, and can you explain why?
>
> **A:** In our first submission, we tested three synthetic datasets and one real dataset, and in all our experiments, iKG performs the best in the compared algorithms. It is probably because iKG is asymptotically optimal while the compared algorithms are not, and this advantage can be inherited by iKG when the number of samples is finite (and small).
>
> As discussed in our global response, we will compare KG and iKG on three additional real examples in the updated manuscript. It can help us better understand the relative performances of these two algorithms.
>
> **Q:** The authors discuss the limitations of KG, but not of their techniques specifically. I think this could be accomplished by showing some contrary datasets or examples.
>
> **A:** As suggested by the reviewer, in the updated manuscript, we will add three real examples in the numerical experiments. It can help us better understand the relative performances of KG and iKG.

---

### Author Rebuttal · Authors · 2023-08-09

Multiple reviewers pointed out that the numerical part of this research should be strengthened. During the rebuttal period, we have tested the empirical performances of the proposed iKG, iKG-$\epsilon$ and iKG-F algorithms on three additional real examples from two new datasets. The new test examples and results will be added to the updated manuscript.

The first example is about selecting effective contraceptives based on the Drug Review Dataset (https://doi.org/10.24432/C5SK5S). We consider five contraceptives: Ethinyl estradiol / levonorgest, Ethinyl estradiol / norethindro, Ethinyl estradiol / norgestimat, Etonogestrel and Nexplanon, which can be treated as five arms. The dataset provides user reviews on the five drugs along with related conditions and a $10$ star user rating reflecting overall user satisfaction. We set the means of the five arms as $\mu_{1}=5.8676$, $\mu_{2}=5.6469$, $\mu_{3}=5.8765$, $\mu_{4}=5.8298$ and $\mu_{5}=5.6332$, and the variances of the five arms as $\sigma_{1}^{2}=3.2756$, $\sigma_{2}^{2}=3.4171$, $\sigma_{3}^{2}=3.2727$, $\sigma_{4}^{2}=3.3198$ and $\sigma_{5}^{2}=3.3251$, all calculated by the data. When this example is used for the best arm identification and $\epsilon$-good arm identification, the best arm (with the highest user satisfaction) and 0.003-good arm are both arm 3 (Ethinyl estradiol / norgestimat). When this example is used for feasible arm identification, we will select the drugs whose ratings are over $5.8$, and the feasible arms are arm $1$ (Ethinyl estradiol / levonorgest), arm $3$ (Ethinyl estradiol / norgestimat) and arm $4$ (Etonogestrel).

The second and third examples are about selecting good captions based on the New Yorker Cartoon Caption Contest Dataset (https://nextml.github.io/caption-contest-data/). In the contests, each caption can be treated as an arm. The dataset provides the mean and variance of each arm, which can be used to set up our experiments. Specifically, we will test contests 853 (example two) and 854 (example three).

In example two, we randomly select ten captions as arms. We set the means of the ten arms as $\mu_{1}=1.1400$, $\mu_{2}=1.0779$, $\mu_{3}=1.4160$, $\mu_{4}=1.0779$, $\mu_{5}=1.1081$, $\mu_{6}=1.1467$, $\mu_{7}=1.1333$, $\mu_{8}=1.1075$, $\mu_{9}=1.1026$ and $\mu_{10}=1.4900$, and the variances of the arms as $\sigma_{1}^{2}=0.1418$, $\sigma_{2}^{2}=0.0991$, $\sigma_{3}^{2}=0.4871$, $\sigma_{4}^{2}=0.0728$, $\sigma_{5}^{2}=0.0977$, $\sigma_{6}^{2}=0.1809$, $\sigma_{7}^{2}=0.1843$, $\sigma_{8}^{2}=0.0970$, $\sigma_{9}^{2}=0.0932$ and $\sigma_{10}^{2}=0.4843$, which are all calculated by the data. When this example is used for the best arm identification, the best arm (with the highest funniness score) is arm 10. When this example is used for $\epsilon$-good arm identification, the 0.1-good arms are arms 3 and 10. When this example is used for feasible arm identification, we will select the captions whose funniness scores are over 1.4, and the feasible arms are arms 3 and 10.

In example three, we also randomly select ten captions as arms. We set the means of the ten arms as $\mu_{1}=1.1986$, $\mu_{2}=1.1890$, $\mu_{3}=1.1400$, $\mu_{4}=1.2621$, $\mu_{5}=1.1544$, $\mu_{6}=1.0339$, $\mu_{7}=1.1349$, $\mu_{8}=1.2786$, $\mu_{9}=1.1765$ and $\mu_{10}=1.1367$, and the variances of the arms as $\sigma_{1}^{2}=0.1879$, $\sigma_{2}^{2}=0.2279$, $\sigma_{3}^{2}=0.1346$, $\sigma_{4}^{2}=0.3186$, $\sigma_{5}^{2}=0.1314$, $\sigma_{6}^{2}=0.0330$, $\sigma_{7}^{2}=0.1337$, $\sigma_{8}^{2}=0.3167$, $\sigma_{9}^{2}=0.1858$ and $\sigma_{10}^{2}=0.1478$, all calculated by the data. When this example is used for the best arm identification, the best arm is arm 8. When this example is used for $\epsilon$-good arm identification, the 0.05-good arms are arms 4 and 8. When this example is used for feasible arm identification, we will select the captions whose funniness scores are over 1.25, and the feasible arms are arms 4 and 8.

The three new examples are applied to the three BAI and variant problems for testing, except that Example 1 is not tested for the feasible arm identification due to time limitation. Here we briefly report the results (please see the attachment for more details).

Figures 1(a)-1(c) show how the probabilities of false selection (PFS) of the compared algorithms change with the sample sizes for the best arm identification problem in Examples 1-3. It can be observed that the proposed iKG algorithm performs the best, followed by KG, EI and the equal allocation. On the log scale, the PFS values of the iKG algorithm demonstrate linear patterns, indicating the potentially exponential convergence rates. For both EI and KG, the rates of posterior convergence are not optimal, which might compromise their empirical performances. The equal allocation performs the worst in general.

Figures 2(a)-2(c) show how the probabilities of false selection of the compared algorithms change with the sample sizes for the $\epsilon$-good arm identification in Examples 1-3. It can be observed that the proposed iKG-${\epsilon}$ algorithm performs the best and demonstrates a linear pattern on the log scale. (ST)$^2$ and APT are inferior, and the equal allocation performs the worst.

Figures 3(a)-3(b) show how the probabilities of false selection of the compared algorithms change with the sample sizes for the feasible arm identification in Examples 2 and 3. The results are similar to those in Figures 1(a)-2(c). The proposed iKG-F algorithm has the best performance, followed by the compared MD-UCBE, equal allocation and MD-SAR.

---

### Decision · Program_Chairs · 2023-09-21

**Decision:**

Accept (poster)

**Comment:**

This paper studies best arm identification in multi-armed bandits and proposes an improved version of the knowledge gradient algorithm. It proves asymptotic converge rates that are optimal and which are a strict improvement of the rate of the vanilla knowledge gradient algorithm. The paper further empirically compares the proposed method against baselines on several benchmark problems.

There is consensus that the paper provides a good contribution to the Best Arm Identification problem and the reviewers appreciated the improved convergence rate shown for the proposed iKG variant. Reviewer eJBQ was concerned that the extension in this paper is similar to the TTEI extension for EI. The authors addressed this concern by pointing out the differences. The main concern raised in several reviews is the scope of the experimental evaluation. The authors promised to add 3 additional benchmarks and presented a summary of the results in their rebuttal.

Based on the ACs own reading of the paper and the reviewer discussion, there are however additional issues with the empirical evaluation:
* The results do not show CIs. The authors state that their reported results are averages over 100 replications so the conclusions are probably valid, but uncertainty measures have to be included. The plotted averages in the appendix look somewhat noisy which may suggest that they are not yet converged to the population limit and the CIs may be fairly large, but without actual CIs, that is all speculation.  Statistical confidence quantification (through error bars / confidence regions / p-values) is a must for a NeurIPS paper in 2023. In fact, the authors did not fill out that part of the checklist.
* Given that TTEI with $\beta = 1/2$ has been shown by https://arxiv.org/abs/1705.10033 to perform much better than EI and KG, the authors should include a comparison against it as well. The asymptotic rates suggest that iKG performs as well as TTEI for any $\beta$ so finite-sample empirical evaluations would be very interesting here.
* It is unclear how the sample-sizes were picked that were shown in Table 1 and 2. There was concern that these may be cherry-picked. Looking at the full graphs in the appendix, the AC believes that iKG overall performs on par or better than alternatives on most benchmarks, but definitely understands these concerns and they show that the way the results are presented raise doubts.  To avoid any such concerns, authors should include the full graphs including confidence intervals instead of tables..

Besides these issues related to the empirical evaluation there are also other concerns related to the discussion and presentation:
* As pointed out by reviewer 2tPM, the paper needs a more comprehensive discussion on the limitations of the algorithm and results.
* Up to the direct comparison to TTEI through the way the results are stated, there is very little discussion comparing the rates to those shown for other algorithms.

The active reviewer discussion could not reach a consensus on whether to accept or reject the paper. While the empirical results leave a lot of things to be desired, the main contribution of this paper is of theoretical nature and the convergence rate bounds are indeed interesting. Overall, this makes the paper very much borderline. Since the primary contributions of this paper are the theoretical analysis and algorithm design, which are interesting and significant on their own, this paper is recommended to be accepted. However, the authors do need to address the issues in the empirical evaluation and improve the discussion of the relation to other works in the camera-ready version.